# On the Margin Theory of Feedforward Neural Networks

## Abstract

Past works have shown that, somewhat surprisingly, over-parametrization can help generalization in neural networks. Towards explaining this phenomenon, we adopt a margin-based perspective. We establish: 1) for multi-layer feedforward relu networks, the global minimizer of a weakly-regularized cross-entropy loss has the maximum normalized margin among all networks, 2) as a result, increasing the over-parametrization improves the normalized margin and generalization error bounds for deep networks. In the case of two-layer networks, an infinite-width neural network enjoys the best generalization guarantees. The typical infinite feature methods are kernel methods; we compare the neural net margin with that of kernel methods and construct natural instances where kernel methods have much weaker generalization guarantees. We validate this gap between the two approaches empirically. Finally, this infinite-neuron viewpoint is also fruitful for analyzing optimization. We show that a perturbed gradient flow on infinite-size networks finds a global optimizer in polynomial time.

## 1 Introduction

In deep learning, over-parametrization refers to the widely-adopted technique of using more parameters than necessary (Krizhevsky et al., 2012; Livni et al., 2014). Both computationally and statistically, over-parametrization is crucial for learning neural nets. Controlled experiments demonstrate that over-parametrization eases optimization by smoothing the non-convex loss surface (Livni et al., 2014; Sagun et al., 2017). Statistically, increasing model size without any regularization still *improves* generalization even after the model interpolates the data perfectly (Neyshabur et al., 2017b). This is surprising given the conventional wisdom on the trade-off between model capacity and generalization.

In the absence of an explicit regularizer, algorithmic regularization is likely the key contributor to good generalization. Recent works have shown that gradient descent finds the minimum norm solution fitting the data for problems including logistic regression, linearized neural networks, and matrix factorization (Soudry et al., 2018; Gunasekar et al., 2018b; Li et al., 2018; Gunasekar et al., 2018a; Ji & Telgarsky, 2018). Many of these proofs require a delicate analysis of the algorithm's dynamics, and some are not fully rigorous due to assumptions on the iterates. To the best of our knowledge, it is an open question to prove analogous results for even two-layer relu networks. (For example, the technique of Li et al. (2018) on two-layer neural nets with quadratic activations still falls within the realm of linear algebraic tools, which apparently do not suffice for other activations.)

We propose a different route towards understanding generalization: *making the regularization explicit*. The motivations are: 1) with an explicit regularizer, we can analyze generalization without fully understanding optimization; 2) it is unknown whether gradient descent provides additional implicit regularization beyond what $\ell_2$ regularization already offers; 3) on the other hand, with a sufficiently weak $\ell_2$ regularizer, we can prove stronger results that apply to multi-layer relu networks. Additionally, explicit regularization is perhaps more relevant because $\ell_2$ regularization is typically used in practice.

Concretely, we add a norm-based regularizer to the cross entropy loss of a multi-layer feedforward neural network with relu activations. We show that the global minimizer of the regularized objective achieves the maximum normalized margin among all the models with the same architecture, if the regularizer is sufficiently weak (Theorem 2.1). Informally, for models with norm 1 that perfectly classify the data, the margin is the smallest difference across all datapoints between the classifier score for the true label and the next best score. We are interested in normalized margin because its

inverse bounds the generalization error (see recent work (Bartlett et al., 2017; Neyshabur et al., 2017a; 2018; Golowich et al., 2017) or Proposition 3.1). Our work explains why optimizing the training loss can lead to parameters with a large margin and thus, better generalization error (see Corollary 3.2). We further note that the maximum possible margin is non-decreasing in the width of the architecture, and therefore the generalization bound of Corollary 3.2 can only improve as the size of the network grows (see Theorem 3.3). Thus, even if the dataset is already separable, it could still be useful to increase the width to achieve larger margin and better generalization.

At a first glance, it might seem counterintuitive that *decreasing* the regularizer is the right approach. At a high level, we show that the regularizer only serves as a tiebreaker to steer the model towards choosing the largest normalized margin. Our proofs are simple, oblivious to the optimization procedure, and apply to any norm-based regularizer. We also show that an exact global minimum is unnecessary: if we approximate the minimum loss within a constant factor, we obtain the max-margin within a constant factor (Theorem 2.2).

To better understand the neural network max-margin, in Section 4 we compare the max-margin two-layer network obtained by optimizing both layers jointly to kernel methods corresponding to fixing random weights for the hidden layer and solving a 2-norm max-margin on the top layer. We design a simple data distribution (Figure 1) where neural net margin is large but the kernel margin is small. This translates to an $\Omega(\sqrt{d})$ factor gap between the generalization error bounds for the two approaches and demonstrates the power of neural nets compared to kernel methods. We experimentally confirm that a gap does indeed exist.

In the setting of two-layer networks, we also study how over-parametrization helps optimization. Prior works (Mei et al., 2018; Chizat & Bach, 2018; Sirignano & Spiliopoulos, 2018; Rotskoff & Vanden-Eijnden, 2018) show that gradient descent on two-layer networks becomes Wasserstein gradient flow over parameter distributions in the limit of infinite neurons. For this setting, we prove that perturbed Wasserstein gradient flow finds a global optimizer in polynomial time.

Finally, we empirically validate several claims made in this paper. First, we confirm that neural networks do generalize better than kernel methods. Second, we show that for two-layer networks, the test error decreases and margin increases as the hidden layer grows, as predicted by our theory.

## 1.1 Additional Related Work

Zhang et al. (2016) and Neyshabur et al. (2017b) show that neural network generalization defies conventional explanations and requires new ones. Neyshabur et al. (2014) initiate the search for the "inductive bias" of neural networks towards solutions with good generalization. Recent papers (Hardt et al., 2015; Brutzkus et al., 2017; Chaudhari et al., 2016) study inductive bias through training time and sharpness of local minima. Neyshabur et al. (2015a) propose a new steepest descent algorithm in a geometry invariant to weight rescaling and show that this improves generalization. Morcos et al. (2018) relate generalization in deep nets to the number of "directions" in the neurons. Other papers (Gunasekar et al., 2017; Soudry et al., 2018; Nacson et al., 2018; Gunasekar et al., 2018b; Li et al., 2018; Gunasekar et al., 2018a) study implicit regularization towards a specific solution. Ma et al. (2017) show that implicit regularization can help gradient descent avoid overshooting optima. Rosset et al. (2004a;b) study logistic regression with a weak regularization and show convergence to the max margin solution. We adopt their techniques and extend their results.

A line of work initiated by Neyshabur et al. (2015b) has focused on deriving tighter norm-based Rademacher complexity bounds for deep neural networks (Bartlett et al., 2017; Neyshabur et al., 2017a; Golowich et al., 2017) and new compression based generalization properties (Arora et al., 2018b). Dziugaite & Roy (2017) manage to compute non-vacuous generalization bounds from PAC-Bayes bounds. Neyshabur et al. (2018) investigate the Rademacher complexity of two-layer networks and propose a bound that is decreasing with the distance to initialization. Liang & Rakhlin (2018) and Belkin et al. (2018) study the generalization of kernel methods.

On the optimization side, Soudry & Carmon (2016) explain why over-parametrization can remove bad local minima. Safran & Shamir (2016) show that over-parametrization can improve the quality of the random initialization. Haeffele & Vidal (2015), Nguyen & Hein (2017), and Venturi et al. (2018) show that for sufficiently overparametrized networks, all local minima are global, but do not show how to find these minima via gradient descent. Du & Lee (2018) show that for two-layer networks

with quadratic activations, all second-order stationary points are global minimizers. Arora et al. (2018a) interpret over-parametrization as a means of implicit acceleration during optimization. Mei et al. (2018), Chizat & Bach (2018), and Sirignano & Spiliopoulos (2018) take a distributional view of over-parametrized networks. Chizat & Bach (2018) show that Wasserstein gradient flow converges to global optimizers under structural assumptions. We extend this to a polynomial-time result.

## 1.2 NOTATION

Let $\mathbb{R}$ denote the set of real numbers. We will use $\|\cdot\|$ to indicate a general norm, with $\|\cdot\|_1, \|\cdot\|_2, \|\cdot\|_\infty$ denoting the $\ell_1, \ell_2, \ell_\infty$ norms on finite dimensional vectors, respectively, and $\|\cdot\|_F$ denoting the Frobenius norm on a matrix. In general, we use $^-$ on top of a symbol to denote a unit vector: when applicable, $\bar{u} \triangleq u/\|u\|$, where the norm $\|\cdot\|$ will be clear from context. Let $\mathbb{S}^{d-1} \triangleq \{\bar{u} \in \mathbb{R}^d : \|\bar{u}\|_2 = 1\}$ be the unit sphere in $d$ dimensions. Let $\mathcal{L}^p(\mathbb{S}^{d-1})$ be the space of functions on $\mathbb{S}^{d-1}$ for which the $p$-th power of the absolute value is Lebesgue integrable. For $\alpha \in \mathcal{L}^p(\mathbb{S}^{d-1})$, we overload notation and write $\|\alpha\|_p \triangleq \left(\int_{\mathbb{S}^{d-1}} |\alpha(\bar{u})|^p d\bar{u}\right)^{1/p}$. Additionally, for $\alpha_1 \in \mathcal{L}^1(\mathbb{S}^{d-1})$ and $\alpha_2 \in \mathcal{L}^\infty(\mathbb{S}^{d-1})$ or $\alpha_1, \alpha_2 \in \mathcal{L}^2(\mathbb{S}^{d-1})$, we can define $\langle \alpha_1, \alpha_2 \rangle \triangleq \int_{\mathbb{S}^{d-1}} \alpha_1(\bar{u})\alpha_2(\bar{u})d\bar{u} < \infty$. Furthermore, we will use $\mathrm{Vol}(\mathbb{S}^{d-1}) \triangleq \int_{\mathbb{S}^{d-1}} 1 d\bar{u}$. Throughout this paper, we reserve the symbol $X = [x_1, \ldots, x_n]$ to denote the collection of datapoints (as a matrix), and $Y = [y_1, \ldots, y_n]$ to denote labels. We use $d$ to denote the dimension of our data. We often use $\Theta$ to denote the parameters of a prediction function $f$, and $f(\Theta; x)$ to denote the prediction of $f$ on datapoint $x$. We will use the notation $\lesssim, \gtrsim$ to mean less than or greater than up to a universal constant, respectively. Unless stated otherwise, $O(\cdot), \Omega(\cdot)$ denote some universal constant in upper and lower bounds, respectively. The notation poly denotes a universal constant-degree polynomial in the arguments.

## 2 WEAK REGULARIZER GUARANTEES MAX MARGIN SOLUTIONS

In this section, we will show that when we add a weak regularizer to cross-entropy loss with a positive-homogeneous prediction function, the normalized margin of the optimum converges to some max-margin solution. As a concrete example, feedforward relu networks are positive-homogeneous.

Let $l$ be the number of labels, so the $i$-th example has label $y_i \in [l]$. We work with a family $\mathcal{F}$ of prediction functions $f(\Theta; \cdot) : \mathbb{R}^d \to \mathbb{R}^l$ that are $a$-positive-homogeneous in their parameters for some $a > 0$: $f(c\Theta; x) = c^a f(\Theta; x), \forall c > 0$. We additionally require that $f$ is continuous in $\Theta$. For some general norm $\|\cdot\|$, we study the $\lambda$-regularized cross-entropy loss $L_\lambda$, defined as

$$L_\lambda(\Theta) \triangleq \sum_{i=1}^n -\log \frac{\exp(f_{y_i}(\Theta; x_i))}{\sum_{j=1}^l \exp(f_j(\Theta; x_i))} + \lambda\|\Theta\|^r \tag{2.1}$$

for fixed $r > 0$. Let $\Theta_\lambda \in \arg\min L_\lambda(\Theta)$.[1] We define the normalized margin of $\Theta_\lambda$ as:

$$\gamma_\lambda \triangleq \min_i \left( f_{y_i}(\bar{\Theta}_\lambda; x_i) - \max_{j \neq y_i} f_j(\bar{\Theta}_\lambda; x_i) \right) \tag{2.2}$$

Define the $\|\cdot\|$-max normalized margin as

$$\gamma^\star \triangleq \max_{\|\Theta\| \leq 1} \left[ \min_i \left( f_{y_i}(\Theta; x_i) - \max_{j \neq y_i} f_j(\Theta; x_i) \right) \right]$$

and let $\Theta^\star$ be a parameter achieving this maximum. We show that with sufficiently small regularization level $\lambda$, the normalized margin $\gamma_\lambda$ approaches the maximum margin $\gamma^\star$. Our theorem and proof are inspired by the result of Rosset et al. (2004a;b), who analyze the special case when $f$ is a linear predictor. In contrast, our result can be applied to non-linear $f$ as long as $f$ is homogeneous.

**Theorem 2.1.** *Assume the training data is separable by a network $f(\Theta^\star; \cdot) \in \mathcal{F}$ with an optimal normalized margin $\gamma^\star > 0$. Then, the normalized margin of the global optimum of the weakly-regularized objective (equation 2.1) converges to $\gamma^\star$ as the strength of the regularizer goes to zero. Mathematically, let $\gamma_\lambda$ be defined in equation 2.2. Then*

$$\gamma_\lambda \to \gamma^\star \text{ as } \lambda \to 0$$

---

[1] We formally show that $L_\lambda$ has a minimizer in Claim A.1 of Section A.

An intuitive explanation for our result is as follows: because of the homogeneity, the loss $L(\Theta_\lambda)$ roughly satisfies the following (for small $\lambda$, and ignoring problem parameters such as $n$):

$$L_\lambda(\Theta_\lambda) \approx \exp(-\|\Theta_\lambda\|^a \gamma_\lambda) + \lambda\|\Theta_\lambda\|^r$$

Thus, the loss selects parameters with larger margin, while the regularization favors parameters with a smaller norm. The full proof of the theorem is deferred to Section A.1.

Theorem 2.1 applies to feedforward relu networks and states that global minimizers of the weakly-regularized loss will obtain a maximum margin among all networks of the given architecture. By considering global minimizers, Theorem 2.1 provides a framework for directly analyzing generalization properties of the solution without considering details of the optimization algorithm. In Section 3 we leverage this framework and existing generalization bounds (Golowich et al., 2017) to provide a clean argument that over-parameterization can improve generalization.

We can also provide an analogue of Theorem 2.1 for the binary classification setting. For this setting, our prediction is now a single real output and we train using logistic loss. We provide formal definitions and results in Section A.2. Our study of the generalization properties of the max-margin (see Section 3 and Section 4) is based in this setting.

## 2.1 OPTIMIZATION ACCURACY

Since $L_\lambda$ is typically hard to optimize exactly for neural nets, we study how accurately we need to optimize $L_\lambda$ to obtain a margin that approximates $\gamma^\star$ up to a constant. The following theorem shows that it suffices to find $\Theta'$ achieving a constant factor multiplicative approximation of $L_\lambda(\Theta_\lambda)$, where $\lambda$ is some sufficiently small polynomial in $n, l, \gamma^\star$. Though our theorem is stated for the general multi-class setting, it also applies for binary classification. We provide the proof in Section A.3.

**Theorem 2.2.** *In the setting of Theorem 2.1, suppose that we choose*

$$\lambda = \exp(-(2^{r/a} - 1)^{-a/r}) \frac{(\gamma^\star)^{r/a}}{n^c (l-1)^c}$$

*for sufficiently large $c$ (that only depends on $r/a$). For $\beta \leq 2$, let $\Theta'$ denote a $\beta$-approximate minimizer of $L_\lambda$, so $L_\lambda(\Theta') \leq \beta L_\lambda(\Theta_\lambda)$. Denote the normalized margin of $\Theta'$ by $\gamma'$. Then*

$$\gamma' \geq \frac{\gamma^\star}{10 \cdot \beta^{a/r}}.$$

## 3 GENERALIZATION PROPERTIES OF A MAXIMUM MARGIN NEURAL NETWORK

In Section 2 we showed that optimizing a weakly-regularized logistic loss leads to the maximum normalized margin. We now study the direct implications of this result on the generalization properties of the solution. Specifically, we use existing Rademacher complexity bounds of Golowich et al. (2017) to present a generalization bound that depends on the network architecture only through the inverse $\ell_2$-normalized margin and depth of the network (see Proposition 3.1). Next, we combine this bound with Theorem 2.1 to conclude that parameters obtained by optimizing logistic loss with weak $\ell_2$-regularization will have a generalization bound that scales with the inverse of the maximum possible margin and depth. Finally, we note that the maximum possible margin can only increase as the size of the network grows, which suggests that increasing the size of the network improves the generalization of the solution (see Theorem 3.3).

We consider depth-$K$ neural networks with 1-Lipschitz, 1-positive-homogeneous activation $\phi$ for $K \geq 2$. Suppose that the collection of parameters $\Theta$ is given by matrices $W_1, \ldots, W_K$. The $K$-layer network will compute a real-valued score

$$f(\Theta; x) \triangleq W_K \phi(W_{K-1}\phi(\cdots \phi(W_1 x) \cdots)) \tag{3.1}$$

where we overload notation to let $\phi(\cdot)$ denote the element-wise application of the activation $\phi$. Let $m_i$ denote the size of the $i$-th hidden layer, so $W_1 \in \mathbb{R}^{m_1 \times d}, W_2 \in \mathbb{R}^{m_2 \times m_1}, \cdots, W_K \in \mathbb{R}^{1 \times m_{K-1}}$. We will let $\mathcal{M} \triangleq (m_1, \ldots, m_{K-1})$ denote the sequence of hidden layer sizes. We will focus on

$\ell_2$-regularized loss. The weakly-regularized logistic loss of the depth-$K$ architecture with hidden layer sizes $\mathcal{M}$ is therefore

$$L_{\lambda,\mathcal{M}}(\Theta) \triangleq \frac{1}{n} \sum_{i=1}^{n} \log(1 + \exp(-y_i f(\Theta; x_i))) + \lambda \|\Theta\|_F^2 \tag{3.2}$$

We note that $f$ is $K$-homogeneous in $\Theta$, so the results of Section 2 apply to $L_{\lambda,\mathcal{M}}$.[2] Following our conventions from Section 2, we denote the optimizer of $L_{\lambda,\mathcal{M}}$ by $\Theta_{\lambda,\mathcal{M}}$, the normalized margin of $\Theta_{\lambda,\mathcal{M}}$ by $\gamma_{\lambda,\mathcal{M}}$, the max-margin solution by $\Theta^{\star,\mathcal{M}}$, and the max-margin by $\gamma^{\star,\mathcal{M}}$. Our notation emphasizes the architecture of the network. Since the classifier $f$ now predicts a single real value, we need to redefine

$$\gamma_{\lambda,\mathcal{M}} \triangleq \min_i y_i f(\bar{\Theta}_{\lambda,\mathcal{M}}; x_i)$$

$$\gamma^{\star,\mathcal{M}} \triangleq \max_{\|\Theta\|_2 \le 1} \min_i y_i f(\Theta; x_i)$$

When the data is not separable by a neural network with architecture $\mathcal{M}$, we define $\gamma^{\star,\mathcal{M}}$ to be zero.

Recall that $X = [x_1, \ldots, x_n]$ denotes the matrix with all the data points as columns, and $Y = [y_1, \ldots, y_n]$ denotes the labels. We sample $X$ and $Y$ i.i.d. from the data generating distribution $p_{\text{data}}$, which is supported on $\mathcal{X} \times \{-1, +1\}$. We can define the population 0-1 loss and training 0-1 loss of the network parametrized by $\Theta$ by

$$L(\Theta) = \Pr_{(x,y) \sim p_{\text{data}}} [y f(\Theta; x) \le 0]$$

Let $C \triangleq \sup_{x \in \mathcal{X}} \|x\|_2$ be an upper bound on the norm of a single datapoint. Proposition 3.1 shows that the generalization error only depends on the parameters through the inverse of the margin on the training data. We obtain Proposition 3.1 by applying Theorem 1 of Golowich et al. (2017) with the standard technique of using margin loss to bound classification error. There exist other generalization bounds which depend on the margin and some normalization (Neyshabur et al., 2015b; 2017a; Bartlett et al., 2017; Neyshabur et al., 2018); we choose the bounds of Golowich et al. (2017) because they fit well with $\ell_2$ normalization. In the two-layer case $K = 2$, the bound below also follows from Neyshabur et al. (2015b).

**Proposition 3.1.** *[Straightforward consequence of Golowich et al. (2017, Theorem 1)] Suppose $\phi$ is 1-Lipschitz and 1-positive-homogeneous. For any depth-$K$ network $f(\Theta; \cdot)$ separating the data with normalized margin $\gamma \triangleq \min_i y_i f(\bar{\Theta}; x_i) > 0$, with probability at least $1 - \delta$ over the draw of $X, Y$,*

$$L(\Theta) \lesssim \frac{C}{\gamma K^{(K-1)/2} \sqrt{n}} + \epsilon(\gamma) \tag{3.3}$$

*where $\epsilon(\gamma) \triangleq \sqrt{\frac{\log \log_2 \frac{4C}{\gamma}}{n}} + \sqrt{\frac{\log(1/\delta)}{n}}$. Note that $\epsilon(\gamma)$ is typically small, and thus the above bound mainly scales with $\frac{C}{\gamma K^{(K-1)/2} \sqrt{n}}$.[3]*

For completeness, we state the proof in Section C.1. By combining this bound with our Theorem 2.1 we can conclude that optimizing weakly-regularized logistic loss gives us generalization error bounds that depend on the maximum possible margin of a network with the given architecture.

**Corollary 3.2.** *In the setting of Proposition 3.1, with probability $1 - \delta$,*

$$\limsup_{\lambda \to 0} L(\Theta_{\lambda,\mathcal{M}}) \lesssim \frac{C}{\gamma^{\star,\mathcal{M}} K^{(K-1)/2} \sqrt{n}} + \epsilon(\gamma^{\star,\mathcal{M}}) \tag{3.4}$$

*where $\epsilon(\gamma)$ is defined as in Proposition 3.1. Above we implicitly assume $\gamma^{\star,\mathcal{M}} > 0$, since otherwise the right hand side of the bound is vacuous.*

---

[2]Although Theorem 2.1 is written in the language of multi-class prediction where the classifier outputs $l \ge 2$ scores, the results translate to single-output binary classification. See Section A.2.

[3]Although the $\frac{1}{K^{(K-1)/2}}$ factor of equation 3.3 decreases with depth $K$, the margin $\gamma$ will also tend to decrease as the constraint $\|\bar{\Theta}\|_F \le 1$ becomes more stringent.

By applying Theorem 2.2 with Proposition 3.1, we can also conclude that optimizing $L_{\lambda,\mathcal{M}}$ within a constant factor gives a margin, and therefore generalization bound, approximating the best possible.

One consequence of Corollary 3.2 is that optimizing weakly-regularized logistic loss results in the best possible generalization bound out of all models with the given architecture. This indicates that the widely used algorithm of optimizing deep networks with $\ell_2$-regularized logistic loss has an implicit bias towards solutions with good generalization.

Next, we observe that the maximum normalized margin is non-decreasing with the size of the architecture. Formally, for two depth-$K$ architectures $\mathcal{M} = (m_1, \ldots, m_{K-1})$ and $\mathcal{M}' = (m'_1, \ldots, m'_{K-1})$, we say $\mathcal{M} \leq \mathcal{M}'$ if $m_i \leq m'_i \ \forall i = 1, \ldots K - 1$. Theorem 3.3 states that if $\mathcal{M} \leq \mathcal{M}'$, then the max-margin over networks with architecture $\mathcal{M}'$ is at least the max-margin over networks with architecture $\mathcal{M}$.

**Theorem 3.3.** *Recall that $\gamma^{\star,\mathcal{M}}$ denotes the maximum normalized margin of a network with architecture $\mathcal{M}$. If $\mathcal{M} \leq \mathcal{M}'$, we have $\gamma^{\star,\mathcal{M}} \leq \gamma^{\star,\mathcal{M}'}$. As a important consequence,*

*the generalization error bound of Corollary 3.2 for $\mathcal{M}'$ is at least as good as that for $\mathcal{M}$.*

This theorem is simple to prove and follows because we can directly implement any network of architecture $\mathcal{M}$ using one of architecture $\mathcal{M}'$, if $\mathcal{M} \leq \mathcal{M}'$. This can explain why additional over-parameterization has been empirically observed to improve generalization in two-layer networks (Neyshabur et al., 2017b): the margin does not decrease with a larger network size, and therefore Corollary 3.2 gives a better generalization bound. In Section 6, we provide empirical evidence that the test error decreases with larger network size while the margin is non-decreasing.

The phenomenon in Theorem 3.3 contrasts with standard $\ell_2$-normalized linear prediction. In this setting, adding more features increases the norm of the data, and therefore the generalization error bounds could also increase. On the other hand, Theorem 3.3 shows that adding more neurons (which can be viewed as learned features) can only improve the generalization of the max-margin solution.

# 4 NEURAL NET MAX-MARGIN VS. KERNEL METHODS

We will continue our study of the max-margin neural network via comparison against kernel methods, a context in which margins have already been extensively studied. We show that two-layer networks can obtain a larger margin, and therefore better generalization guarantees, than kernel methods. Our comparison between the two methods is motivated by an equivalence between the $\ell_2$ max-margin of an infinite-width two-layer network and the $\ell_1$-SVM (Zhu et al., 2004) over the lifted feature space defined by the activation function applied to all possible hidden units (Neyshabur et al., 2014; Rosset et al., 2007; Bengio et al., 2006). The kernel method corresponds to the $\ell_2$-SVM in this same feature space, and is equivalent to fixing random hidden layer weights and solving an $\ell_2$-SVM over the top layer. In Theorem 4.3, we construct a distribution for which the generalization upper bounds for the $\ell_1$-SVM on this feature space are smaller than those for the $\ell_2$-SVM by a $\Omega(\sqrt{d})$ factor. Our work provides evidence that optimizing all layers of a network can be beneficial for generalization.

There have been works that compare $\ell_1$ and $\ell_2$-regularized solutions in the context of feature selection and construct a feature space for which a generalization gap exists (e.g., see Ng (2004)). In contrast, we work in the fixed feature space of relu activations, which makes our construction particularly challenging.

We will use $m$ to denote the width of the single hidden layer of the network. Following the convention from Section 3, we will use $\gamma^{\star,m}$ to denote the maximum possible normalized margin of a two-layer network with hidden layer size $m$ (note the emphasis on the size of the single hidden layer). The depth $K = 2$ case of Corollary 3.2 immediately implies that optimizing weakly-regularized $\ell_2$ loss over width-$m$ two-layer networks gives parameters whose generalization upper bounds depend on the hidden layer size only through $1/\gamma^{\star,m}$. Furthermore, from Theorem 3.3 it immediately follows that

$$\gamma^{\star,1} \leq \gamma^{\star,2} \leq \cdots \leq \gamma^{\star,\infty}$$

The work of Neyshabur et al. (2014) links $\gamma^{\star,m}$ to the $\ell_1$ SVM over a lifted space. Formally, we define a lifting function $\varphi : \mathbb{R}^d \to \mathcal{L}^\infty(\mathbb{S}^{d-1})$ mapping data to an infinite feature vector:

$$x \in \mathbb{R}^d \to \varphi(x) \in \mathcal{L}^\infty(\mathbb{S}^{d-1}) \text{ satisfying } \varphi(x)[\bar{u}] = \phi(\bar{u}^\top x) \tag{4.1}$$

where $\phi$ is the activation of Section 3. We look at the margin of linear functionals corresponding to $\alpha \in \mathcal{L}^1(\mathbb{S}^{d-1})$. The 1-norm SVM (Zhu et al., 2004) over the lifted feature $\varphi(x)$ solves for the maximum margin:

$$\gamma_{\ell_1} \triangleq \max_{\alpha} \min_{i \in [n]} y_i \langle \alpha, \varphi(x_i) \rangle$$

$$\text{subject to } \|\alpha\|_1 \leq 1 \tag{4.2}$$

where we rely on the inner product and 1-norm defined in Section 1.2. This formulation is equivalent to a hard-margin optimization on "convex neural networks" (Bengio et al., 2006). Bach (2017) also study optimization and generalization of convex neural networks. Using results from Rosset et al. (2007); Neyshabur et al. (2014); Bengio et al. (2006), our Theorem 2.1 implies that optimizing weakly-regularized logistic loss over two-layer networks is equivalent to solving equation 4.2 when the size of the hidden layer is at least $n + 1$, where $n$ is the number of training examples. Proposition 4.1 essentially restates this with the minor improvement that this equivalence[4] also holds when the size of the hidden layer is $n$.

**Proposition 4.1.** *Let $\gamma_{\ell_1}$ be defined in equation 4.2. Then $\frac{\gamma_{\ell_1}}{2} = \gamma^{\star,n} = \cdots = \gamma^{\star,\infty}$.*

For completeness, we prove Proposition 4.1 in Section B, relying on the work of Tibshirani (2013) and Rosset et al. (2004a).

Importantly, the $\ell_1$-max margin on the lifted feature space is obtainable by optimizing a finite neural network. We compare this to the $\ell_2$ margin attainable via kernel methods. Following the setup of equation 4.2, we define the kernel problem over $\alpha \in \mathcal{L}^2(\mathbb{S}^{d-1})$:

$$\gamma_{\ell_2} \triangleq \max_{\alpha} \min_{i \in [n]} y_i \langle \alpha, \varphi(x_i) \rangle$$

$$\text{subject to } \sqrt{\kappa} \|\alpha\|_2 \leq 1 \tag{4.3}$$

Figure 1: A visualization of 60 sampled points from $\mathcal{D}$ in 3 dimensions. Red points denote negative examples and blue points denote positive examples.

where $\kappa \triangleq \mathrm{Vol}(\mathbb{S}^{d-1})$. (We scale $\|\alpha\|_2$ by $\sqrt{\kappa}$ to make the lemma statement below cleaner.) First, $\gamma_{\ell_2}$ can be used to obtain a standard upper bound on the generalization error of the kernel SVM. Following the notation of Section 3, we will let $L_{\ell_2\text{-svm}}$ denote the 0-1 population classification error for the optimizer of equation 4.3.

**Lemma 4.2.** *In the setting of Proposition 3.1, with probability at least $1-\delta$, the generalization error of the standard kernel SVM with* relu *feature (defined in equation 4.3) is bounded by*

$$L_{\ell_2\text{-svm}} \lesssim \frac{C}{\gamma_{\ell_2}\sqrt{dn}} + \epsilon_{\ell_2} \tag{4.4}$$

*where $\epsilon_{\ell_2} \triangleq \sqrt{\dfrac{\log \max\left\{\log_2 \frac{C}{\sqrt{d}\gamma_{\ell_2}}, 2\right\}}{n}} + \sqrt{\dfrac{\log(1/\delta)}{n}}$ is typically a lower-order term.*

The bound above follows from standard techniques (Bartlett & Mendelson, 2002), and we provide a full proof in Section C.2. We construct a data distribution for which this lemma does not give a good bound for kernel methods, but Corollary 3.2 does imply good generalization for two-layer networks.

**Theorem 4.3.** *There exists a data distribution $p_{\text{data}}$ such that the $\ell_1$ SVM with* relu *features has a good margin: $\gamma_{\ell_1} \gtrsim 1$ and with probability $1 - \delta$ over the choice of i.i.d. samples from $p_{\text{data}}$, obtains generalization error*

$$L_{\ell_1\text{-svm}} \lesssim \sqrt{\frac{d \log n}{n}} + \epsilon_{\ell_1}$$

*where $\epsilon_{\ell_1} \triangleq \sqrt{\dfrac{\log(1/\delta)}{n}}$ is typically a lower order term. Meanwhile, with high probability the $\ell_2$ SVM has a small margin: $\gamma_{\ell_2} \lesssim \max\left\{\sqrt{\dfrac{\log n}{n}}, 1/d\right\}$ and therefore the generalization upper bound from*

---

[4]The factor of $\frac{1}{2}$ is due the the relation that every unit-norm parameter $\Theta$ corresponds to an $\alpha$ in the lifted space with $\|\alpha\| = 2$.

*Lemma 4.2 is at least*

$$\Omega\left(\min\left\{1, d\sqrt{\frac{\log n}{n}}\right\}\right)$$

*In particular, the $\ell_2$ bound is larger than the $\ell_1$ bound by a $\Omega(\sqrt{d})$ factor.*

Although Theorem 4.3 compares upper bounds, our construction highlights properties of distributions which result in better neural network generalization than kernel method generalization. Furthermore, in Section 6 we empirically validate the gap in generalization between the two methods.

We briefly overview the construction of $p_{\text{data}}$ here. The full proof is in Section D.1.

**Proof sketch for Theorem 4.3.** We base $p_{\text{data}}$ on the distribution $\mathcal{D}$ of examples $(x, y)$ described below. Here $e_i$ is the i-th standard basis vector and we use $x^\top e_i$ to represent the $i$-coordinate of $x$ (since the subscript is reserved to index training examples).

$$\begin{bmatrix} e_3^\top x \\ \vdots \\ e_d^\top x \end{bmatrix} \sim \mathcal{N}(0, I_{d-2}), \text{ and } \begin{cases} y = +1, & x^\top e_1 = +1, & x^\top e_2 = +1 & \text{w/ prob. } 1/4 \\ y = +1, & x^\top e_1 = -1, & x^\top e_2 = -1 & \text{w/ prob. } 1/4 \\ y = -1, & x^\top e_1 = +1, & x^\top e_2 = -1 & \text{w/ prob. } 1/4 \\ y = -1, & x^\top e_1 = -1, & x^\top e_2 = +1 & \text{w/ prob. } 1/4 \end{cases}$$

Figure 1 shows samples from $\mathcal{D}$ when there are 3 dimensions. From the visualization, it is clear that there is no linear separator for $\mathcal{D}$. As Lemma D.1 shows, a relu network with four neurons can fit this relatively complicated decision boundary. On the other hand, for kernel methods, we prove that the symmetries in $\mathcal{D}$ induce cancellation in feature space. As a result, the features are less predictive of the true label and the margin will therefore be small. We formalize this argument in Section D.1.

**Gap in regression setting:** We are able to prove an even larger $\Omega(\sqrt{n/d})$ gap between neural networks and kernel methods in the regression setting where we wish to interpolate continuous labels. Analogously to the classification setting, optimizing a regularized squared error loss on neural networks is equivalent to solving a minimum 1-norm regression problem (see Theorem D.5). Furthermore, kernel methods correspond to a minimum 2-norm problem. We construct distributions $p_{\text{data}}$ where the 1-norm solution will have a generalization error bound of $O(\sqrt{d/n})$, whereas the 2-norm solution will have a generalization error bound that is $\Omega(1)$ and thus vacuous. In Section D.2, we define the 1-norm and 2-norm regression problems. In Theorem D.10 we formalize our construction.

## 5 PERTURBED WASSERSTEIN GRADIENT FLOW FINDS GLOBAL OPTIMIZERS IN POLYNOMIAL TIME

In the prior section, we studied the limiting behavior of the generalization of a two-layer network as its width goes to infinity. In this section, we will now study the limiting behavior of the optimization algorithm, gradient descent. Prior work (Mei et al., 2018; Chizat & Bach, 2018) has shown that as the hidden layer size grows to infinity, gradient descent for a finite neural network approaches the Wasserstein gradient flow over distributions of hidden units (defined in equation 5.1). Chizat & Bach (2018) assume the gradient flow converges, a non-trivial assumption since the space of distributions is infinite-dimensional, and given the assumption prove that Wasserstein gradient flow converges to a global optimizer in this setting, but do not specify a convergence rate. Mei et al. (2018) show global convergence for the infinite-neuron limit of stochastic Langevin dynamics, but also do not provide a convergence rate.

We show that a perturbed version of Wasserstein gradient flow converges in *polynomial time*. The informal take-away of this section is that a perturbed version of gradient descent converges in polynomial time on infinite-size neural networks (for the right notion of infinite-size.)

Formally, we optimize the following functional over distributions $\rho$ on $\mathbb{R}^{d+1}$:

$$L[\rho] \triangleq R\left(\int \Phi d\rho\right) + \int V d\rho$$

where $\Phi : \mathbb{R}^{d+1} \to \mathbb{R}^k$, $R : \mathbb{R}^k \to \mathbb{R}$, and $V : \mathbb{R}^{d+1} \to \mathbb{R}$. In this work, we consider 2-homogeneous $\Phi$ and $V$. We will additionally require that $R$ is convex and nonnegative and $V$ is positive on the unit sphere. Finally, we need standard regularity assumptions on $R$, $\Phi$, and $V$:

**Assumption 5.1** (Regularity conditions on $\Phi$, $R$, $V$). $\Phi$ and $V$ are differentiable as well as upper bounded and Lipschitz on the unit sphere. $R$ is Lipschitz and its Hessian has bounded operator norm.

We provide more details on the specific parameters (for boundedness, Lipschitzness, etc.) in Section E.1. We note that relu networks satisfy every condition but differentiability of $\Phi$.[5] We can fit a neural network under our framework as follows:

**Example 5.2** (Logistic loss for neural networks). We interpret $\rho$ as a distribution over the parameters of the network. Let $k \triangleq n$ and $\Phi_i(\theta) \triangleq w\phi(u^\top x_i)$ for $\theta = (w, u)$. In this case, $\int \Phi d\rho$ is a distributional neural network that computes an output for each of the $n$ training examples (like a standard neural network, it also computes a weighted sum over hidden units). We can compute the distributional version of the regularized logistic loss in equation 3.2 by setting $V(\theta) \triangleq \lambda\|\theta\|_2^2$ and $R(a_1, \ldots, a_n) \triangleq \sum_{i=1}^n \log(1 + \exp(-y_i a_i))$.

We will define $L'[\rho] : \mathbb{R}^{d+1} \to \mathbb{R}$ with $L'[\rho](\theta) \triangleq \langle R'(\int \Phi d\rho), \Phi(\theta)\rangle + V(\theta)$ and $v[\rho](\theta) \triangleq -\nabla_\theta L'[\rho](\theta)$. Informally, $L'[\rho]$ is the gradient of $L$ with respect to $\rho$, and $v$ is the induced velocity field. For the standard Wasserstein gradient flow dynamics, $\rho_t$ evolves according to

$$\frac{d}{dt}\rho_t = -\nabla \cdot (v[\rho_t]\rho_t) \tag{5.1}$$

where $\nabla\cdot$ denotes the divergence of a vector field. For neural networks, these dynamics formally define continuous-time gradient descent when the hidden layer has infinite size (see Theorem 2.6 of Chizat & Bach (2018), for instance).

We propose the following modification of the Wasserstein gradient flow dynamics:

$$\frac{d}{dt}\rho_t = -\sigma\rho_t + \sigma U^d - \nabla \cdot (v[\rho_t]\rho_t) \tag{5.2}$$

where $U^d$ is the uniform distribution on $\mathbb{S}^d$. In our perturbed dynamics, we add very small uniform noise over $U^d$, which ensures that at all time-steps, there is sufficient mass in a descent direction for the algorithm to decrease the objective. For infinite-size neural networks, one can informally interpret this as re-initializing a very small fraction of the neurons at every step of gradient descent. We prove convergence to a global optimizer in time polynomial in $1/\epsilon$, $d$, and the regularity parameters.

**Theorem 5.3** (Theorem E.4 with regularity parameters omitted). *Suppose that $\Phi$ and $V$ are 2-homogeneous and the regularity conditions of Assumption 5.1 are satisfied. Also assume that from starting distribution $\rho_0$, a solution to the dynamics in equation 5.2 exists. Define $L^\star \triangleq \inf_\rho L[\rho]$. Let $\epsilon > 0$ be a desired error threshold and choose $\sigma \triangleq \exp(-d\log(1/\epsilon)\text{poly}(k, L[\rho_0] - L^\star))$ and $t_\epsilon \triangleq \frac{d^2}{\epsilon^4}\text{poly}(\log(1/\epsilon), k, L[\rho_0] - L^\star)$, where the regularity parameters for $\Phi$, $V$, and $R$ are hidden in the $\text{poly}(\cdot)$. Then, perturbed Wasserstein gradient flow converges to an $\epsilon$-approximate global minimum in $t_\epsilon$ time:*

$$\min_{0 \le t \le t_\epsilon} L[\rho_t] - L^\star \le \epsilon.$$

We provide a theorem statement that includes regularity parameters in Section E.1. We prove the theorem in Section E.2.

As a technical detail, Theorem 5.3 requires that a solution to the dynamics exists. We can remove this assumption by analyzing a discrete-time version of equation 5.2:

$$\rho_{t+1} \triangleq \rho_t + \eta(-\sigma\rho_t + \sigma U^d - \nabla \cdot (v[\rho_t]\rho_t))$$

and additionally assuming $\Phi$ and $V$ have Lipschitz gradients. In this setting, a polynomial time convergence result also holds. We state the result in Section E.3.

An implication of our Theorem 5.3 is that for infinite networks, we can optimize the weakly-regularized logistic loss in time polynomial in the problem parameters and $\lambda^{-1}$. By Theorem 2.2, we only require $\lambda^{-1} = \text{poly}(n)$ to approximate the maximum margin within a constant factor. Thus, for infinite networks, we can approximate the max margin within a constant factor in polynomial time.

---

[5]The relu activation is non-differentiable at 0 and hence the gradient flow is not well-defined. Chizat & Bach (2018) acknowledge this same difficulty with relu.

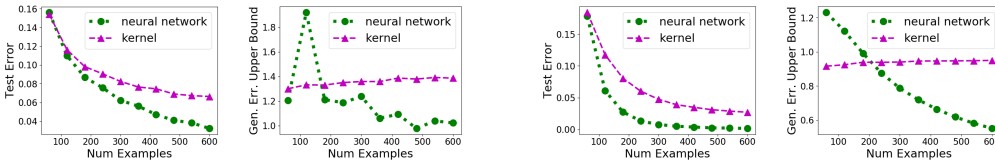

Figure 2: Comparing neural networks and kernel methods. **Left:** Classification. **Right:** Regression.

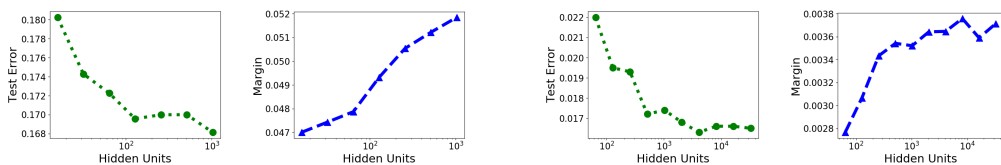

Figure 3: Dependence of margin and test error on hidden layer size. **Left:** Synthetic. **Right:** MNIST.

## 6 SIMULATIONS

We first compare the generalization of neural networks and kernel methods for classification and regression. In Figure 2 we plot the generalization error and predicted generalization upper bounds[6] of a trained neural network against a $\ell_2$ kernel method with relu features as we vary $n$. Our data comes from a synthetic distribution generated by a neural network with 6 hidden units; we provide a detailed setup in Section F.1. For classification we plot 0-1 error, whereas for regression we plot squared error. The variance in the neural network generalization bound for classification likely occured because we did not tune learning rate and training time, so the optimization failed to find the best margin. The plots show that two-layer networks clearly outperform kernel methods in test error as $n$ grows. However, there seems to be looseness in the bounds: the kernel generalization bound appears to stay constant with $n$ (as predicted by our theory for regression), but the test error decreases.

We also plot the dependence of the test error and margin on the hidden layer size in Figure 3 for synthetic data generated from a ground truth network with 10 hidden units and also MNIST. The plots indicate that test error is decreasing in hidden layer size while margin is increasing, as Theorem 3.3 predicts. We provide more details on the experimental setup in Section F.2.

In Section F.3, we verify the convergence of a simple neural network to the max-margin solution as regularization decreases. In Section F.4, we train modified WideResNet architectures on CIFAR10 and CIFAR100. Although ResNet is not homogeneous, we still report improvements in generalization from annealing the weight decay during training, versus staying at a fixed decay rate.

## 7 CONCLUSION

We have made the case that maximizing margin is one of the inductive biases of relu networks obtained from optimizing weakly-regularized cross-entropy loss. Our framework allows us to directly analyze generalization properties of the network without considering the optimization algorithm used to obtain it. Using this perspective, we provide a simple explanation for why over-parametrization can improve generalization. It is a fascinating question for future work to characterize other generalization properties of the max-margin solution. On the optimization side, we make progress towards understanding over-parametrized gradient descent by analyzing infinite-size neural networks. A natural direction for future work is to apply our theory to optimize the margin of finite-sized neural networks.

---

[6]We compute the leading term that is linear in the norm or inverse margin from the bounds in Proposition 3.1 and Lemmas 4.2, D.8, and D.9.

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

## A  MISSING PROOFS IN SECTION 2

We first show that $L_\lambda$ does indeed have a global minimizer.

**Claim A.1.** *In the setting of Theorems 2.1 and A.3,* $\arg\min_\Theta L_\lambda(\Theta)$ *exists.*

*Proof.* We will argue in the setting of Theorem 2.1 where $L_\lambda$ is the multi-class cross entropy loss, because the logistic loss case is analogous. We first note that $L_\lambda$ is continuous in $\Theta$ because $f$ is continuous in $\Theta$ and the term inside the logarithm is always positive. Next, define $b \triangleq \inf_\Theta L_\lambda(\Theta) > 0$. Then we note that for $\|\Theta\| > (b/\lambda)^{1/r} \triangleq M$, we must have $L_\lambda(\Theta) > b$. It follows that $\inf_{\|\Theta\| \leq M} L_\lambda(\Theta) = \inf_\Theta L_\lambda(\Theta)$. However, there must be a value $\Theta_\lambda$ which attains $\inf_{\|\Theta\| \leq M} L_\lambda(\Theta)$, because $\{\Theta : \|\Theta\| \leq M\}$ is a compact set and $L_\lambda$ is continuous. Thus, $\inf_\Theta L_\lambda(\Theta)$ is attained by some $\Theta_\lambda$. □

## A.1 Missing Proofs for Multi-class Setting

Towards proving Theorem 2.1, we first show as we decrease $\lambda$, the norm of the solution $\|\Theta_\lambda\|$ grows.

**Lemma A.2.** *In the setting of Theorem 2.1, as $\lambda \to 0$, we have $\|\Theta_\lambda\| \to \infty$.*

To prove Theorem 2.1, we rely on the exponential scaling of the cross entropy: $L_\lambda$ can be lower bounded roughly by $\exp(-\|\Theta_\lambda\|\gamma_\lambda)$, but also has an upper bound that scales with $\exp(-\|\Theta_\lambda\|\gamma^\star)$. By Lemma A.2, we can take large $\|\Theta_\lambda\|$ so the gap $\gamma^\star - \gamma_\lambda$ vanishes. This proof technique is inspired by that of Rosset et al. (2004a).

*Proof of Theorem 2.1.* For any $M > 0$ and $\Theta$ with $\gamma_\Theta \triangleq \min_i \left( f(\bar{\Theta}; x_i) - \max_{j \neq y_i} f(\bar{\Theta}; x_i) \right)$,

$$L_\lambda(M\Theta) = \frac{1}{n} \sum_{i=1}^n - \log \frac{\exp(M^a f_{y_i}(\Theta; x_i))}{\sum_{j=1}^l \exp(M^a f_j(\Theta; x_i))} + \lambda M^r \|\Theta\|^r \quad \text{(by the homogeneity of } f)$$

$$= \frac{1}{n} \sum_{i=1}^n - \log \frac{1}{1 + \sum_{j \neq y_i} \exp(M^a(f_j(\Theta; x_i) - f_{y_i}(\Theta; x_i)))} + \lambda M^r \|\Theta\|^r \quad \text{(A.1)}$$

$$\leq \log(1 + (l-1)\exp(-M^a \gamma_\Theta)) + \lambda M^r \|\Theta\|^r \quad \text{(A.2)}$$

We can also apply $\sum_{j \neq y_i} \exp(M^a(f_j(\Theta; x_i) - f_{y_i}(\Theta; x_i))) \geq \max \exp(M^a(f_j(\Theta; x_i) - f_{y_i}(\Theta; x_i))) = \exp \gamma_\Theta$ in order to lower bound equation A.1 and obtain

$$L_\lambda(M\Theta) \geq \frac{1}{n} \log(1 + \exp(-M^a \gamma_\Theta)) + \lambda M^r \|\Theta\|^r \quad \text{(A.3)}$$

Applying equation A.2 with $M = \|\Theta_\lambda\|$ and $\Theta = \Theta^\star$, noting that $\|\Theta^\star\| \leq 1$, we have:

$$L_\lambda(\Theta^\star \|\Theta_\lambda\|) \leq \log(1 + (l-1)\exp(-\|\Theta_\lambda\|^a \gamma^\star)) + \lambda \|\Theta_\lambda\|^r \quad \text{(A.4)}$$

Next we lower bound $L_\lambda(\Theta_\lambda)$ by applying equation A.3,

$$L_\lambda(\Theta_\lambda) \geq \frac{1}{n} \log(1 + \exp(-\|\Theta_\lambda\|^a \gamma_\lambda)) + \lambda \|\Theta_\lambda\|^r \quad \text{(A.5)}$$

Combining equation A.4 and equation A.5 with the fact that $L_\lambda(\Theta_\lambda) \leq L_\lambda(\Theta^\star \|\Theta_\lambda\|)$ (by the global optimality of $\Theta_\lambda$), we have

$$\forall \lambda > 0, n \log(1 + (l-1)\exp(-\|\Theta_\lambda\|^a \gamma^\star)) \geq \log(1 + \exp(-\|\Theta_\lambda\|^a \gamma_\lambda))$$

Recall that by Lemma A.2, as $\lambda \to 0$, we have $\|\Theta_\lambda\| \to \infty$. Therefore, $\exp(-\|\Theta_\lambda\|^a \gamma^\star), \exp(-\|\Theta_\lambda\|^a \gamma_\lambda) \to 0$. Thus, we can apply Taylor expansion to the equation above with respect to $\exp(-\|\Theta_\lambda\|^a \gamma^\star)$ and $\exp(-\|\Theta_\lambda\|^a \gamma_\lambda)$. If $\max\{\exp(-\|\Theta_\lambda\|^a \gamma^\star), \exp(-\|\Theta_\lambda\|^a \gamma_\lambda)\} < 1$, then we obtain

$$n(l-1)\exp(-\|\Theta_\lambda\|^a \gamma^\star) \geq \exp(-\|\Theta_\lambda\|^a \gamma_\lambda) - O(\max\{\exp(-\|\Theta_\lambda\|^a \gamma^\star)^2, \exp(-\|\Theta_\lambda\|^a \gamma_\lambda)^2\})$$

We claim this implies that $\gamma^\star \leq \liminf_{\lambda \to 0} \gamma_\lambda$. If not, we have $\liminf_{\lambda \to 0} \gamma_\lambda < \gamma^\star$, which implies that the equation above is violated with sufficiently large $\|\Theta_\lambda\|$ ($\|\Theta_\lambda\| \gg \log(2(\ell-1)n)^{1/a}$ would suffice). By Lemma A.2, $\|\Theta_\lambda\| \to \infty$ as $\lambda \to 0$ and therefore we get a contradiction.

Finally, we have $\gamma_\lambda \leq \gamma^\star$ by definition of $\gamma^\star$. Hence, $\lim_{\lambda \to 0} \gamma_\lambda$ exists and equals $\gamma^\star$. $\qquad \square$

Now we fill in the proof of Lemma A.2.

*Proof of Lemma A.2.* For the sake of contradiction, we assume that $\exists C > 0$ such that for any $\lambda_0 > 0$, there exists $0 < \lambda < \lambda_0$ with $\|\Theta_\lambda\| \leq C$. We will determine the choice of $\lambda_0$ later and pick $\lambda$ such that $\|\Theta_\lambda\| \leq C$. Then the logits (the prediction $f_j(\Theta, x_i)$ before softmax) are bounded in absolute value by some constant (that depends on $C$), and therefore the loss function $- \log \frac{\exp(f_{y_i}(\Theta; x_i))}{\sum_{j=1}^l \exp(f_j(\Theta; x_i))}$ for every example is bounded from below by some constant $D > 0$ (depending on $C$ but not $\lambda$.)

Let $M = \lambda^{-1/(r+1)}$, we have that

$$0 < D \leq L_\lambda(\Theta_\lambda) \leq L_\lambda(M\Theta^\star) \qquad \text{(by the optimality of } \Theta_\lambda)$$

$$\leq -\log \frac{1}{1 + (l-1)\exp(-M^a\gamma^\star)} + \lambda M^r \qquad \text{(by equation A.2)}$$

$$= \log(1 + (l-1)\exp(-\lambda^{-a/(r+1)}\gamma^\star)) + \lambda^{1/(r+1)}$$

$$\leq \log(1 + (l-1)\exp(-\lambda_0^{-a/(r+1)}\gamma^\star)) + \lambda_0^{1/(r+1)}$$

Taking a sufficiently small $\lambda_0$, we obtain a contradiction and complete the proof. $\qquad \square$

## A.2 FULL BINARY CLASSIFICATION SETTING

For completeness, we state and prove our max-margin results for the setting where we fit binary labels $y_i \in \{-1, +1\}$ (as opposed to indices in $[l]$) and redefining $f(\Theta; \cdot)$ to assign a single real-valued score (as opposed to a score for each label). This lets us work with the simpler $\lambda$-regularized logistic loss:

$$L_\lambda(\Theta) \triangleq \frac{1}{n}\sum_{i=1}^n \log(1 + \exp(-y_i f(\Theta; x_i))) + \lambda\|\Theta\|^r$$

As before, let $\Theta_\lambda \in \arg\min L_\lambda(\Theta)$, and define the normalized margin $\gamma_\lambda$ by $\gamma_\lambda \triangleq \min_i y_i f(\bar\Theta_\lambda; x_i)$. Define the maximum possible normalized margin

$$\gamma^\star \triangleq \max_{\|\Theta\| \leq 1} \min_i y_i f(\Theta; x_i) \qquad (A.6)$$

**Theorem A.3.** *Assume $\gamma^\star > 0$ in the binary classification setting with logistic loss. Then as $\lambda \to 0$, $\gamma_\lambda \to \gamma^\star$.*

The proof follows via simple reduction to the multi-class case.

*Proof of Theorem A.3.* We prove this theorem via reduction to the multi-class case with $l = 2$. Construct $\tilde f : \mathbb{R}^d \to \mathbb{R}^2$ with $\tilde f_1(\Theta; x_i) = -\frac{1}{2}f(\Theta; x_i)$ and $\tilde f_2(\Theta; x_i) = \frac{1}{2}f(\Theta; x_i)$. Define new labels $\tilde y_i = 1$ if $y_i = -1$ and $\tilde y_i = 2$ if $y_i = 1$. Now note that $\tilde f_{\tilde y_i}(\Theta; x_i) - \tilde f_{j \neq \tilde y_i}(\Theta; x_i) = y_i f(\Theta; x_i)$, so the multi-class margin for $\Theta$ under $\tilde f$ is the same as binary margin for $\Theta$ under $f$. Furthermore, defining

$$\tilde L_\lambda(\Theta) \triangleq \frac{1}{n}\sum_{i=1}^n -\log \frac{\exp(\tilde f_{\tilde y_i}(\Theta; x_i))}{\sum_{j=1}^2 \exp(\tilde f_j(\Theta; x_i))} + \lambda\|\Theta\|^r$$

we get that $\tilde L_\lambda(\Theta) = L_\lambda(\Theta)$, and in particular, $\tilde L_\lambda$ and $L_\lambda$ have the same set of minimizers. Therefore we can apply Theorem 2.1 for the multi-class setting and conclude $\gamma_\lambda \to \gamma^\star$ in the binary classification setting. $\qquad \square$

## A.3 MISSING PROOF FOR OPTIMIZATION ACCURACY

*Proof of Theorem 2.2.* Choose $B \triangleq \left(\frac{1}{\gamma^\star}\log\frac{(l-1)(\gamma^\star)^{r/a}}{\lambda}\right)^{1/a}$. We can upper bound $L_\lambda(\Theta')$ by computing

$$L_\lambda(\Theta') \leq \beta L\lambda(\Theta_\lambda) \leq \beta L_\lambda(B\Theta^\star)$$

$$\leq \beta \log(1 + (l-1)\exp(-B^a\gamma^\star)) + \beta\lambda B^r \qquad \text{(by equation A.2)}$$

$$\leq \beta(l-1)\exp(-B^a\gamma^\star) + \beta\lambda B^r \qquad \text{(using } \log(1+x) \leq x)$$

$$\leq \beta\frac{\lambda}{(\gamma^\star)^{r/a}} + \beta\lambda\left(\frac{1}{\gamma^\star}\log\frac{(l-1)(\gamma^\star)^{r/a}}{\lambda}\right)^{r/a}$$

$$\leq \beta\frac{\lambda}{(\gamma^\star)^{r/a}}\left(1 + \left(\log\frac{(l-1)(\gamma^\star)^{r/a}}{\lambda}\right)^{r/a}\right) \triangleq L^{(UB)}$$

Furthermore, it holds that $\|\Theta'\|^r \leq \frac{L^{(UB)}}{\lambda}$. Now we note that

$$L_\lambda(\Theta') \leq L^{(UB)} \leq 2\beta \frac{\lambda}{(\gamma^\star)^{r/a}} \left( \log \frac{(l-1)(\gamma^\star)^{r/a}}{\lambda} \right)^{r/a} \leq \frac{1}{2n}$$

for sufficiently large $c$ depending only on $a/r$. Now using the fact that $\log(x) \geq \frac{x}{1+x} \ \forall x \geq -1$, we additionally have the lower bound $L_\lambda(\Theta') \geq \frac{1}{n} \log(1 + \exp(-\gamma'\|\Theta'\|^a)) \geq \frac{1}{n} \frac{\exp(-\gamma'\|\Theta'\|^a)}{1+\exp(-\gamma'\|\Theta'\|^a)}$. Since $L^{(UB)} \leq 1$, we can rearrange to get

$$\gamma' \geq \frac{-\log \frac{nL_\lambda(\Theta')}{1-nL_\lambda(\Theta')}}{\|\Theta'\|^a} \geq \frac{-\log \frac{nL^{(UB)}}{1-nL^{(UB)}}}{\|\Theta'\|^a} \geq \frac{-\log(2nL^{(UB)})}{\|\Theta'\|^a}$$

The middle inequality followed because $\frac{x}{1-x}$ is increasing in $x$ for $0 \leq x < 1$, and the last because $L^{(UB)} \leq \frac{1}{2n}$. Since $-\log 2nL^{(UB)} > 0$ we can also apply the bound $\|\Theta'\|^r \leq \frac{L^{(UB)}}{\lambda}$ to get

$$\gamma' \geq \frac{-\lambda^{a/r} \log 2nL^{(UB)}}{(L^{(UB)})^{a/r}}$$

$$= \frac{-\log \left( 2n\beta \frac{\lambda}{(\gamma^\star)^{r/a}} \left( 1 + \left( \log \frac{(l-1)(\gamma^\star)^{r/a}}{\lambda} \right)^{r/a} \right) \right)}{\frac{\beta^{a/r}}{\gamma^\star} \left( 1 + \left( \log \frac{(l-1)(\gamma^\star)^{r/a}}{\lambda} \right)^{r/a} \right)^{a/r}} \qquad \text{(by definition of } L^{(UB)}\text{)}$$

$$\geq \frac{\gamma^\star}{\beta^{a/r}} \left( \underbrace{\frac{\log(\frac{(\gamma^\star)^{r/a}}{2\beta n\lambda})}{\left( 1 + \left( \log \frac{(l-1)(\gamma^\star)^{r/a}}{\lambda} \right)^{r/a} \right)^{a/r}}}_{\clubsuit} - \underbrace{\frac{\log \left( 1 + \left( \log \frac{(l-1)(\gamma^\star)^{r/a}}{\lambda} \right)^{r/a} \right)}{\left( 1 + \left( \log \frac{(l-1)(\gamma^\star)^{r/a}}{\lambda} \right)^{r/a} \right)^{a/r}}}_{\heartsuit} \right)$$

We will first bound $\clubsuit$. First note that

$$\frac{\log(\frac{(\gamma^\star)^{r/a}}{2\beta n\lambda})}{\log \frac{(l-1)(\gamma^\star)^{r/a}}{\lambda}} = \frac{\log \frac{(\gamma^\star)^{r/a}}{\lambda} - \log 2\beta n}{\log \frac{(\gamma^\star)^{r/a}}{\lambda} + \log(l-1)} \geq \frac{\log \frac{(\gamma^\star)^{r/a}}{\lambda} - \log 2\beta n(l-1)}{\log \frac{(\gamma^\star)^{r/a}}{\lambda}} \geq \frac{c-3}{c} \qquad \text{(A.7)}$$

where the last inequality follows from the fact that $\frac{(\gamma^\star)^{r/a}}{\lambda} \geq n^c(l-1)^c$ and $\beta \leq 2$. Next, using the fact that $\log \frac{(\gamma^\star)^{r/a}}{\lambda} \geq \frac{1}{(2^{r/a}-1)^{a/r}}$, we note that

$$\left( 1 + \left( \log \frac{(l-1)(\gamma^\star)^{r/a}}{\lambda} \right)^{-r/a} \right)^{a/r} \leq \left( 1 + \left( \frac{1}{(2^{r/a}-1)^{a/r}} \right)^{-r/a} \right)^{a/r} \leq 2 \qquad \text{(A.8)}$$

Combining equation A.7 and equation A.8, we can conclude that

$$\clubsuit = \frac{\log(\frac{(\gamma^\star)^{r/a}}{2\beta n\lambda})}{\log \frac{(l-1)(\gamma^\star)^{r/a}}{\lambda}} \left( 1 + \left( \log \frac{(l-1)(\gamma^\star)^{r/a}}{\lambda} \right)^{-r/a} \right)^{-a/r} \geq \frac{c-3}{2c}$$

Finally, we note that if $1 + \left( \log \frac{(l-1)(\gamma^\star)^{r/a}}{\lambda} \right)^{r/a}$ is a sufficiently large constant that depends only on $a/r$ (which can be achieved by choosing $c$ sufficiently large), it will follow that $\heartsuit \leq \frac{1}{10}$. Thus, if $c \geq 5$, we can combine our bounds on $\clubsuit$ and $\heartsuit$ to get that

$$\gamma' \geq \frac{\gamma^\star}{10\beta^{a/r}}$$

$\square$

# B    MISSING PROOF OF PROPOSITION 4.1

Proposition 4.1 follows simply from applying Corollary 1 of Neyshabur et al. (2014) to a hard-margin SVM problem. For completeness, we provide another proof here. The proof of Proposition 4.1 will consist of two steps: first, show that equation 4.2 has an optimal solution with sparsity $n$, and second, show that sparse solutions to equation 4.2 can be mapped to a neural network with the same margin, and vice versa. The following lemma and proof are based on Lemma 14 of Tibshirani (2013).

**Lemma B.1.** *Let $supp(\alpha) \triangleq \{\bar{u} : |\alpha(\bar{u})| > 0\}$. There exists an optimal solution $\alpha^\star$ to equation 4.2 with $|supp(\alpha^\star)| \leq n$.*

For the proof of this lemma, we find it convenient to work with a minimum norm formulation which we show is equivalent to equation 4.2:

$$\min_{\alpha} \|\alpha\|_1$$
$$\text{subject to } y_i \langle \alpha, \varphi(x_i) \rangle \geq 1 \; \forall i \tag{B.1}$$

**Claim B.2.** *Let $S \subset \mathcal{L}^1(\mathbb{S}^{d-1})$ be the set of optimizers for equation 4.2, and let $S' \subset \mathcal{L}^1(\mathbb{S}^{d-1})$ be the set of optimizers for equation B.1. If equation B.1 is feasible, for any $\alpha \in S$, $\frac{\alpha}{\gamma_{\ell_1}} \in S'$, and for any $\alpha' \in S'$, $\frac{\alpha'}{\|\alpha'\|_1} \in S$.*

*Proof.* Let $\mathrm{opt}'$ denote the optimal objective for equation B.1. We note that $\frac{\alpha'}{\|\alpha'\|_1}$ is feasible for equation 4.2 with objective $\frac{1}{\mathrm{opt}'}$, and therefore $\gamma_{\ell_1} \geq \frac{1}{\mathrm{opt}'}$. Furthermore, $\frac{1}{2\gamma_{\ell_1}} y_i \int_{\bar{u} \in \mathbb{S}^{d-1}} \alpha(\bar{u}) \phi(\bar{u}^\top x_i) d\bar{u} \geq 1 \; \forall i$, and so $\frac{\alpha}{\gamma_{\ell_1}}$ is feasible for equation B.1 with objective $\frac{1}{\gamma_{\ell_1}}$. Therefore, $\mathrm{opt}' \leq \frac{1}{\gamma_{\ell_1}}$. As a result, it must hold that $\mathrm{opt}' = \frac{1}{\gamma_{\ell_1}}$, which means that $\frac{\alpha'}{\|\alpha'\|_1}$ is optimal for equation 4.2, and $\frac{\alpha}{\gamma_{\ell_1}}$ is optimal for equation B.1, as desired.    □

First, note that if equation B.1 is not feasible, then $\gamma_{\ell_1} = 0$ and equation 4.2 has a trivial sparse solution, the all zeros function. Thus, it suffices to show that an optimal solution to equation B.1 exists that is $n$-sparse, since by Lemma B.2 equation B.1 and equation 4.2 have equivalent solutions up to a scaling. We begin by taking the dual of equation B.1.

**Claim B.3.** *The dual of equation B.1 has form*

$$\max_{\lambda \in \mathbb{R}^n} \lambda^\top \vec{1}$$
$$\text{subject to } \left| \sum_{i=1}^n \lambda_i y_i \phi(\bar{u}^\top x_i) \right| \leq 1 \; \forall \bar{u} \in \mathbb{S}^{d-1}$$
$$\lambda_i \geq 0$$

*For any primal optimal solution $\alpha^\star$ and dual optimal solution $\lambda^\star$, it must hold that*

$$\sum_{i=1}^n \lambda_i^\star y_i \phi(\bar{u}^\top x_i) = \mathrm{sign}(\alpha^\star(\bar{u})) \iff \alpha^\star(\bar{u}) \neq 0 \tag{B.2}$$

*Proof.* The dual form can be solved for by computation. By strong duality, equation B.2 must follow from the KKT conditions.    □

Now define the mapping $v : \mathbb{S}^{d-1} \to \mathbb{R}^n$ with $v_i(\bar{u}) \triangleq y_i \phi(\bar{u}^\top x_i)$. We will show a general result about linearly dependent $v(\bar{u})$ for $\bar{u} \in supp(\alpha^\star)$, after which we can reduce directly to the proof of Tibshirani (2013).

**Claim B.4.** *Let $\alpha^\star$ be any optimal solution. Suppose that there exists $S \subseteq supp(\alpha^\star)$ such that $\{v(\bar{u}) : \bar{u} \in S\}$ forms a linearly dependent set, i.e.*

$$\sum_{\bar{u} \in S} c_{\bar{u}} v(\bar{u}) = \vec{0} \tag{B.3}$$

*for coefficients c. Then $\sum_{\bar{u} \in S} c_{\bar{u}} \, \mathrm{sign}(\alpha^\star(\bar{u})) = 0$.*

*Proof.* Let $\lambda^\star$ be any dual optimal solution, then $\lambda^{\star\top} v(\bar{u}) = \text{sign}(\alpha^\star(\bar{u})) \; \forall \bar{u} \in \text{supp}(\alpha^\star)$ by Claim B.3. Thus, we apply $\lambda^{\star\top}$ to both sides of equation B.3 to get the desired statement. $\qquad \square$

*Proof of Lemma B.1.* The rest of the proof follows Lemma 14 in Tibshirani (2013). The lemma argues that if the conclusion of Claim B.4 holds and an optimal solution $\alpha^\star$ has $S \subseteq \text{supp}(\alpha^\star)$ with $\{v(\bar{u}) : \bar{u} \in S\}$ linearly dependent, we can construct a new $\alpha'$ with $\|\alpha'\|_1 = \|\alpha^\star\|_1$ and $\text{supp}(\alpha') \subset \text{supp}(\alpha^\star)$ (where the inclusion is strict). Thus, if we consider an optimal $\alpha^\star$ with minimal support, it must follow that $\{v(\bar{u}) : \bar{u} \in \text{supp}(\alpha^\star)\}$ is a linearly independent set, and therefore $|\text{supp}(\alpha^\star)| \leq n$. $\qquad \square$

We can now complete the proof of Proposition 4.1.

*Proof of Proposition 4.1.* For ease of notation, we will parametrize a two-layer network with $m$ units by top layer weights $w_1, \ldots, w_m \in \mathbb{R}$ and bottom layer weights $u_1, \ldots, u_m \in \mathbb{R}^d$. As before, we use $\Theta$ to refer to the collection of parameters, so the network computes the real-valued function

$$f(\Theta; x) = \sum_{j=1}^{m} w_j \phi(u_j^\top x)$$

Note that we simply renamed the variables from the parametrization of equation 3.1.

We first apply Lemma B.1 to conclude that equation 4.2 admits a $n$-sparse optimal solution $\alpha^\star$. Because of sparsity, we can now abuse notation and treat $\alpha^\star$ as a real-valued function such that $\sum_{\bar{u} \in \text{supp}(\alpha^\star)} |\alpha^\star(\bar{u})| \leq 1$. We construct $\Theta$ corresponding to a two-layer network with $m \geq n$ hidden units and normalized margin at least $\frac{\gamma_{\ell_1}}{2}$. For clarity, we let $W$ correspond to the top layer weights and $U$ correspond to the bottom layer weights. For every $\bar{u} \in \text{supp}(\alpha)$, we let $\Theta$ have a corresponding hidden unit $j$ with $(w_j, u_j) = \left( \text{sign}(\alpha^\star(\bar{u}))\sqrt{\frac{|\alpha^\star(\bar{u})|}{2}}, \sqrt{\frac{|\alpha^\star(\bar{u})|}{2}}\bar{u} \right)$, and set the remaining hidden units to $\vec{0}$. This is possible because $m \geq n$. Now

$$f(\Theta; x) = \sum_{j=1}^{m} w_j \phi(u_j^\top x) = \frac{1}{2} \sum_{\bar{u} \in \text{supp}(\alpha^\star)} \alpha^\star(\bar{u})\phi(\bar{u}^\top x)$$

Furthermore,

$$\|\Theta\|_2^2 = \sum_{j=1}^{m} w_j^2 + \|u_j\|_2^2 = \sum_{\bar{u} \in \text{supp}(\alpha)} \frac{|\alpha^\star(\bar{u})|}{2} + \frac{|\alpha^\star(\bar{u})|}{2}\|\bar{u}\|_2^2 = \sum_{\bar{u} \in \text{supp}(\alpha)} |\alpha^\star(\bar{u})| \leq 1$$

Thus it follows that $\Theta$ has normalized margin at least $\gamma_{\ell_1}/2$, so $\gamma^{\star,m} \geq \gamma_{\ell_1}/2$.

To conclude, we show that $\gamma^{\star,m} \leq \gamma_{\ell_1}/2$. Let $\Theta^{\star,m}$ denote the parameters obtaining optimal $m$-unit margin $\gamma^{\star,m}$ with hidden units $(w_j^{\star,m}, u_j^{\star,m})$ for $j \in [m]$. We can construct $\alpha$ to put a scaled delta mass of $2w_j^{\star,m}\|u_j^{\star,m}\|_2$ on $\bar{u}_j^{\star,m}$ for $j \in [m]$. It follows that

$$\|\alpha\|_1 = \sum_{j=1}^{m} 2|w_j^{\star,m}|\|u_j^{\star,m}\|_2 \leq \sum_{j=1}^{m} w_j^{\star,m2} + \|u_j^{\star,m}\|_2^2 = \|\Theta^{\star,m}\|_2^2 \leq 1$$

Furthermore,

$$\int_{\mathbb{S}^{d-1}} \alpha(\bar{u})\phi(\bar{u}^\top x) = 2\sum_{j=1}^{m} w_j^{\star,m}\|u_j^{\star,m}\|_2 \phi((\bar{u}_j^{\star,m})^\top x)$$

$$= 2\sum_{j=1}^{m} w_j^{\star,m}\phi(u_j^{\star,m\top} x) = 2f(\Theta^{\star,m}; x)$$

Thus, $\alpha$ is a feasible solution to equation 4.2 with objective value at least $2\gamma^{\star,m}$. Therefore, $\gamma_{\ell_1} \geq 2\gamma^{\star,m}$, so $\gamma^{\star,m} = \gamma_{\ell_1}/2$. $\qquad \square$

## C    RADEMACHER COMPLEXITY AND GENERALIZATION ERROR

We prove the generalization error bounds stated in Proposition 3.1 and Lemma 4.2 via Rademacher complexity and margin theory.

Assume that our data $X, Y$ are drawn i.i.d. from ground truth distribution $p_{\text{data}}$ supported on $\mathcal{X} \times \mathcal{Y}$. For some hypothesis class $\mathcal{F}$ of real-valued functions, we define the empirical Rademacher complexity $\hat{\mathfrak{R}}(\mathcal{F})$ as follows:

$$\hat{\mathfrak{R}}(\mathcal{F}) \triangleq \frac{1}{n} \mathbb{E}_{\epsilon_i} \left[ \sup_{f \in \mathcal{F}} \sum_{i=1}^{n} \epsilon_i f(x_i) \right]$$

where $\epsilon_i$ are independent Rademacher random variables.

For a classifier $f$, following the notation of Section 3 we will use $L(f) \triangleq \Pr_{(x,y) \sim p_{\text{data}}}(y f(x) \leq 0)$ to denote the population 0-1 loss of the classifier $f$. The following classical theorem (Koltchinskii et al., 2002), (Kakade et al., 2009) bounds generalization error in terms of the Rademacher complexity and margin loss.

**Theorem C.1** (Theorem 2 of Kakade et al. (2009))**.** *Let $(x_i, y_i)_{i=1}^{n}$ be drawn iid from $p_{\text{data}}$. We work in the binary classification setting, so $\mathcal{Y} = \{-1, 1\}$. Assume that for all $f \in \mathcal{F}$, we have $\sup_{x \in \mathcal{X}} f(x) \leq C$. Then with probability at least $1 - \delta$ over the random draws of the data, for every $\gamma > 0$ and $f \in \mathcal{F}$,*

$$L(f) \leq \frac{1}{n} \sum_{i=1}^{n} \mathbf{1}(y_i f(x_i) < \gamma) + \frac{4\hat{\mathfrak{R}}(\mathcal{F})}{\gamma} + \sqrt{\frac{\log \log_2 \frac{4C}{\gamma}}{n}} + \sqrt{\frac{\log(1/\delta)}{2n}}$$

### C.1    PROOF OF PROPOSITION 3.1

We will prove Proposition 3.1 by applying the Rademacher complexity bounds of Golowich et al. (2017) with Theorem C.1.

First, we show the following lemma bounding the generalization of neural networks whose weight matrices have bounded Frobenius norms.

**Lemma C.2.** *Define the hypothesis class $\mathcal{F}_K$ over depth-$K$ neural networks by*

$$\mathcal{F}_K = \left\{ f(\Theta; \cdot) : \|W_j\|_F \leq \frac{1}{\sqrt{K}} \; \forall j \right\}$$

*Let $C \triangleq \sup_{x \in \mathcal{X}} \|x\|_2$. Recall that $L(\Theta)$ denotes the 0-1 population loss $L(f(\Theta; \cdot))$. Then for any $f(\Theta; \cdot) \in \mathcal{F}_K$ classifying the training data correctly with unnormalized margin $\gamma_\Theta \triangleq \min_i y_i f(\Theta; x_i) > 0$, with probability at least $1 - \delta$,*

$$L(\Theta) \lesssim \frac{C}{\gamma_\Theta K^{(K-1)/2} \sqrt{n}} + \sqrt{\frac{\log \log_2 \frac{4C}{\gamma_\Theta}}{n}} + \sqrt{\frac{\log(1/\delta)}{n}} \tag{C.1}$$

*Note the dependence on the unnormalized margin rather than the normalized margin.*

*Proof.* We first claim that $\sup_{f(\Theta; \cdot) \in \mathcal{F}_K} \sup_{x \in \mathcal{X}} f(\Theta; x) \leq C$. To see this, for any $f(\Theta; \cdot) \in \mathcal{F}_K$,

$$\begin{aligned}
f(\Theta; x) &= W_K \phi(\cdots \phi(W_1 x) \cdots) \\
&\leq \|W_K\|_F \|\phi(W_{K-1} \phi(\cdots \phi(W_1 x) \cdots))\|_2 \\
&\leq \|W_K\|_F \|W_{K-1} \phi(\cdots \phi(W_1 x) \cdots)\|_2 \\
&\qquad \text{(since } \phi \text{ is 1-Lipschitz and } \phi(0) = 0 \text{, so } \phi \text{ performs a contraction)} \\
&< \|x\|_2 \leq C \qquad \text{(repeatedly applying this argument and using } \|W_j\|_F < 1\text{)}
\end{aligned}$$

Furthermore, by Theorem 1 of Golowich et al. (2017), $\hat{\mathfrak{R}}(\mathcal{F}_K)$ has upper bound

$$\hat{\mathfrak{R}}(\mathcal{F}_K) \lesssim \frac{C}{K^{(K-1)/2} \sqrt{n}}$$

Thus, we can apply Theorem C.1 to conclude that for all $f(\Theta; \cdot) \in \mathcal{F}_K$ and all $\gamma > 0$, with probability $1 - \delta$,

$$L(\Theta) \lesssim \frac{1}{n} \sum_{i=1}^{n} \mathbf{1}(y_i f(\Theta; x_i) < \gamma) + \frac{C}{\gamma K^{(K-1)/2} \sqrt{n}} + \sqrt{\frac{\log \log_2 \frac{4C}{\gamma}}{n}} + \sqrt{\frac{\log(1/\delta)}{n}}$$

In particular, by definition choosing $\gamma = \gamma_\Theta$ makes the first term on the LHS vanish and gives the statement of the lemma. □

*Proof of Proposition 3.1.* Given parameters $\Theta = (W_1, \ldots, W_K)$, we first construct parameters $\tilde{\Theta} = (\tilde{W}_1, \ldots, \tilde{W}_K)$ such that $f(\bar{\Theta}; \cdot)$ and $f(\tilde{\Theta}; \cdot)$ compute the same function, and $\|\tilde{W}_1\|_F^2 = \|\tilde{W}_2\|_F^2 = \cdots = \|\tilde{W}_K\|_F^2 \leq \frac{1}{K}$. To do this, we set

$$\tilde{W}_j = \frac{(\prod_{k=1}^{K} \|W_k\|_F)^{1/k}}{\|W_j\|_F \|\Theta\|_F} W_j$$

By construction

$$\begin{aligned} \|\tilde{W}_j\|_F^2 &= \frac{(\prod_{k=1}^{K} \|W_k\|_F^2)^{1/k}}{\|\Theta\|_F^2} \\ &= \frac{(\prod_{k=1}^{K} \|W_k\|_F^2)^{1/k}}{\sum_{k=1}^{K} \|W_k\|_F^2} \\ &\leq \frac{1}{k} \qquad \text{(by the AM-GM inequality)} \end{aligned}$$

Furthermore, we also have

$$\begin{aligned} f(\tilde{\Theta}; x) &= \tilde{W}_K \phi(\cdots \phi(\tilde{W}_1 x) \cdots) \\ &= \prod_{j=1}^{K} \frac{(\prod_{k=1}^{K} \|W_k\|_F)^{1/k}}{\|W_j\|_F \|\Theta\|_F} W_K \phi(\cdots \phi(W_1 x) \cdots) \qquad \text{(by the homogeneity of } \phi) \\ &= \frac{1}{\|\Theta\|_F^K} f(\Theta; x) \\ &= f\left(\frac{\Theta}{\|\Theta\|_F}; x\right) \qquad \text{(since } f \text{ is } K\text{-homogeneous in } \Theta) \\ &= f(\bar{\Theta}; x) \end{aligned}$$

Now we note that by construction, $L(\Theta) = L(\tilde{\Theta})$. Now $f(\tilde{\Theta}; \cdot)$ must also classify the training data perfectly, has unnormalized margin $\gamma$, and furthermore $f(\tilde{\Theta}; \cdot) \in \mathcal{F}_K$. As a result, Lemma C.2 allows us to conclude the desired statement. □

To conclude Corollary 3.2, we apply the above on $\Theta_{\lambda, \mathcal{M}}$ and use Theorem A.3.

## C.2 PROOF OF KERNEL GENERALIZATION BOUNDS

Let $\mathcal{F}_B^{2,\phi}$ denote the class of $\ell_2$-bounded linear functionals in lifted feature space: $\mathcal{F}_B^{2,\phi} \triangleq \{x \mapsto \langle \alpha, \varphi(x) \rangle : \alpha \in \mathcal{L}^2(\mathbb{S}^{d-1}), \|\alpha\|_2 \leq B\}$. We abuse notation and write $\alpha \in \mathcal{F}_B^{2,\phi}$ to indicate a linear functional from $\mathcal{F}_B^{2,\phi}$. As before, we will use $L(\alpha)$ to indicate the 0-1 population loss of the classifier $x \mapsto \langle \alpha, \varphi(x) \rangle$ and let $C \triangleq \sup_{x \in \mathcal{X}} \|x\|_2$ be an upper bound on the norm of the data. We focus on analyzing the Rademacher complexity $\hat{\mathfrak{R}}(\mathcal{F}_B^{2,\phi})$, mirroring derivations done in the past (Bartlett & Mendelson, 2002). We include our derivations here for completeness.

**Lemma C.3.** $\hat{\mathfrak{R}}(\mathcal{F}_B^{2,\phi}) \leq \frac{1}{n} B \sqrt{\sum_{i=1}^{n} \|\varphi(x_i)\|_2^2}$.

*Proof.* We write

$$
\begin{aligned}
\hat{\mathfrak{R}}(\mathcal{F}_B^{2,\phi}) &= \frac{1}{n}\mathbb{E}_{\epsilon_i}\left[\sup_{\alpha\in\mathcal{F}_B^{2,\phi}}\langle\alpha,\sum_{i=1}^n\epsilon_i\varphi(x_i)\rangle\right] \\
&\leq \frac{1}{n}\mathbb{E}_{\epsilon_i}\left[\sup_{\alpha\in\mathcal{F}_B^{2,\phi}}\|\alpha\|_2\left\|\sum_{i=1}^n\epsilon_i\varphi(x_i)\right\|_2\right] \\
&\leq \frac{1}{n}B\cdot\mathbb{E}_{\epsilon_i}\left[\left\|\sum_{i=1}^n\epsilon_i\varphi(x_i)\right\|_2\right] \\
&\leq \frac{1}{n}B\sqrt{\mathbb{E}_{\epsilon_i}\left[\left\|\sum_{i=1}^n\epsilon_i\varphi(x_i)\right\|_2^2\right]} \qquad\text{(via Jensen's inequality)} \\
&\leq \frac{1}{n}B\sqrt{\mathbb{E}_{\epsilon_i}\left[\sum_{i=1}^n\sum_{j=1}^n\epsilon_i\epsilon_j\langle\varphi(x_i),\varphi(x_i)\rangle\right]} \\
&\leq \frac{1}{n}B\sqrt{\sum_{i=1}^n\|\varphi(x_i)\|_2^2} \qquad\text{(terms where } i\neq j \text{ cancel out)}
\end{aligned}
$$

$\square$

As an example, we can apply this bound to relu features:

**Corollary C.4.** *Suppose that $\phi$ is the* relu *activation. Let $\kappa \triangleq \text{Vol}(\mathbb{S}^{d-1})$. Then $\hat{\mathfrak{R}}(\mathcal{F}_B^{2,\phi}) \lesssim \frac{B\|X\|_F\sqrt{\kappa}}{n\sqrt{d}} \leq \frac{BC\sqrt{\kappa}}{\sqrt{dn}}$.*

*Proof.* We first show that $\|\varphi(x_i)\|_2^2 = \Theta\left(\frac{\kappa}{d}\|x_i\|_2^2\right)$. We can compute

$$
\begin{aligned}
\|\varphi(x_i)\|_2^2 &= \text{Vol}(\mathbb{S}^{d-1})\mathbb{E}_{\bar{u}\sim\mathbb{S}^{d-1}}[\text{relu}(\bar{u}^\top x_i)^2] \\
&= \frac{\kappa}{d}\mathbb{E}_{\bar{u}\sim\mathbb{S}^{d-1}}[\text{relu}(\sqrt{d}\bar{u}^\top x_i)^2] \\
&= \frac{\kappa}{d}\frac{1}{M_2}\mathbb{E}_{u\sim\mathcal{N}(0,I_{d\times d})}[\text{relu}(u^T x_i)^2] \qquad (M_2 \text{ is the second moment of } \mathcal{N}(0,1)) \\
&= \Theta\left(\frac{\kappa}{d}\|x_i\|_2^2\right) \qquad\qquad\qquad\qquad\qquad\qquad\qquad\qquad (C.2)
\end{aligned}
$$

where the last line uses the computation provided in Lemma A.1 by Du et al. (2017). Now we plug this into Lemma C.3 to get the desired bound. $\square$

We will now prove Lemma 4.2.

*Proof of Lemma 4.2.* From equation C.2, we first obtain $\sup_{x\in\mathcal{X}}\|\varphi(x)\|_2 \lesssim C\sqrt{\frac{\kappa}{d}}$. Denote the optimizer for equation 4.3 by $\alpha_{\ell_2}$. Note that $\sqrt{\kappa}\alpha_{\ell_2} \in \mathcal{F}_1^{2,\phi}$, and furthermore $L(\alpha_{\ell_2}) = L(\sqrt{\kappa}\alpha_{\ell_2})$. Since $\sqrt{\kappa}\alpha_{\ell_2}$ has unnormalized margin $\sqrt{\kappa}\gamma_{\ell_2}$, we apply Theorem C.1 on margin $\sqrt{\kappa}\gamma_{\ell_2}$ and hypothesis class $\mathcal{F}_1^{2,\phi}$ to get with probability $1-\delta$,

$$
\begin{aligned}
L_{\ell_2\text{-svm}} = L(\sqrt{\kappa}\alpha_{\ell_2}) &\leq \frac{4\hat{\mathfrak{R}}(\mathcal{F}_1^{2,\phi})}{\sqrt{\kappa}\gamma_{\ell_2}} + \sqrt{\frac{\log\log_2\frac{4\sup_{x\in\mathcal{X}}\|\varphi(x)\|_2}{\sqrt{\kappa}\gamma_{\ell_2}}}{n}} + \sqrt{\frac{\log(1/\delta)}{2n}} \\
&\lesssim \frac{C}{\gamma_{\ell_2}\sqrt{dn}} + \sqrt{\frac{\log\max\left\{\log_2\frac{C}{\sqrt{d}\gamma_{\ell_2}},2\right\}}{n}} + \sqrt{\frac{\log(1/\delta)}{n}} \\
&\qquad\qquad\qquad\qquad\qquad\qquad\qquad\qquad\text{(applying Corollary C.4)}
\end{aligned}
$$

$\square$

## D MISSING PROOFS FOR COMPARISON TO KERNEL METHODS

### D.1 CLASSIFICATION

In this section we will complete a proof of Theorem 4.3. Recall the construction of the distribution $\mathcal{D}$ provided in Section 4. We first provide a classifier of this data with small $\ell_1$ norm.

**Lemma D.1.** *In the setting of Theorem 4.3, we have that*

$$\gamma_{\ell_1} \geq \frac{\sqrt{2}}{4}.$$

*Proof.* Consider the network $f(x) = \frac{1}{4} \Big( (x^\top (e_1 + e_2)/\sqrt{2})_+ + (x^\top (-e_1 - e_2)/\sqrt{2})_+ - (x^\top (-e_1 + e_2)/\sqrt{2})_+ - (x^\top (e_1 - e_2)/\sqrt{2})_+ \Big)$. The attained margin $\gamma = \frac{\sqrt{2}}{4}$, so $\gamma_{\ell_1} \geq \frac{\sqrt{2}}{4}$. □

Now we will upper bound the margin attainable by the $\ell_2$ SVM.

**Lemma D.2** (Margin upper bound tool). *In the setting of Theorem 4.3, we have*

$$\gamma_{\ell_2} \leq \frac{1}{\sqrt{\kappa}} \cdot \left\| \frac{1}{n} \sum_{i=1}^{n} \varphi(x_i) y_i \right\|_2$$

*Proof.* By the definition of $\gamma_{\ell_2}$, we have that for any $\alpha$ with $\sqrt{\kappa} \|\alpha\|_2 \leq 1$, we have

$$\gamma_{\ell_2} \leq \max_{\sqrt{\kappa}\|\alpha\|_2 \leq 1} \frac{1}{n} \sum_{i=1}^{n} \langle \alpha, y_i \varphi(x_i) \rangle$$

Setting $\alpha = \frac{1}{\sqrt{\kappa}} \frac{1}{n} \sum_{i=1}^{n} \varphi(x_i) y_i / \|\frac{1}{n} \sum_{i=1}^{n} \varphi(x_i) y_i\|_2$ completes the proof. (Attentive readers may realize that this is equivalent to setting the dual variable of the convex program 4.3 to all 1's function.) □

**Lemma D.3.** *In the setting of Theorem 4.3, let $(x_i, y_i)_{i=1}^{n}$ be $n$ i.i.d samples and corresponding labels from $\mathcal{D}$. Let $\varphi$ be defined in equation 4.1 with $\phi = \text{relu}$. With high probability (at least $1 - dn^{-10}$), we have*

$$\left\| \frac{1}{n} \sum_{i=1}^{n} \varphi(x_i) y_i \right\|_2 \lesssim \sqrt{\kappa/n} \log n + \sqrt{\kappa}/d$$

*Proof.* Let $W_i = \varphi(x_i) y_i$. We will bound several quantities regarding $W_i$'s. In the rest of the proof, we will condition on the event $E$ that $\forall i, \|x_i\|_2^2 \lesssim d \log n$. Note that $E$ is a high probability event and conditioned on $E$, $x_i$'s are still independent. We omit the condition on $E$ in the rest of the proof for simplicity.

We first show that assuming the following three inequalities that the conclusion of the Lemma follows.

1. $\forall i, \|W_i\|_2^2 \lesssim \kappa \log n$ .

2. $\sigma^2 \triangleq \text{Var}[\sum_i W_i] \triangleq \sum_{i=1}^{n} \mathbb{E}[\|W_i - \mathbb{E} W_i\|_2^2] \lesssim n\kappa \log n$

3. $\|\mathbb{E}[\sum W_i]\|_2 \lesssim \sqrt{\kappa} n/d.$

By bullets 1, 2, and Bernstein inequality, we have that with probability at least $1 - dn^{-10}$ over the randomness of the data $(X, Y)$,

$$\left\| \sum_{i=1}^{n} W_i - \mathbb{E}\left[ \sum_{i=1}^{n} W_i \right] \right\|_2 \lesssim \sqrt{\kappa} \log^{1.5} n + \sqrt{n\kappa \log^2 n} \lesssim \sqrt{n\kappa \log^2 n}$$

By bullet 3 and equation above, we complete the proof with triangle inequality:

$$\left\|\sum_{i=1}^{n} W_i\right\|_2 \leq \left\|\mathbb{E}\left[\sum_{i=1}^{n} W_i\right]\right\|_2 + \sqrt{n\kappa \log^2 n} \lesssim \sqrt{n\kappa \log^2 n} + \sqrt{\kappa}n/d$$

Therefore, it suffices to prove bullets 1, 2 and 3. Note that 2 is a direct corollary of 1 so we will only prove 1 and 3. We start with 3:

By the definition of the $\ell_2$ norm in $\mathcal{L}^2(\mathbb{S}^{d-1})$ and the independence of $(x_i, y_i)$'s, we can rewrite

$$\left\|\mathbb{E}\left[\sum_{i=1}^{n} W_i\right]\right\|_2^2 = \kappa \cdot n^2 \mathop{\mathbb{E}}_{\bar{u}\sim\mathbb{S}^{d-1}}\left[\mathop{\mathbb{E}}_{(x,y)\sim\mathcal{D}} \varphi(x)[\bar{u}] \cdot y\right]^2 \tag{D.1}$$

Let $\bar{u} = (\bar{u}_1, \ldots, \bar{u}_d)$ and $\bar{u}_{-2} = (\bar{u}_3, \ldots, \bar{u}_d) \in \mathbb{R}^{d-2}$, and define $\tau \triangleq \|\bar{u}_{-2}\|_2$. Let $x_{-2} = (x^\top e_3, \ldots, x^\top e_d)$. Note that $\varphi(x)[\bar{u}]y = y[\bar{u}_1 \cdot x^\top e_1 + \bar{u}_2 \cdot x^\top e_2 + \bar{u}_{-2}^\top x_{-2}]$ and $\bar{u}_{-2}^\top x_{-2}$ has distribution $\|\bar{u}_{-2}\|_2 \cdot \mathcal{N}(0,1) = \tau \cdot \mathcal{N}(0,1)$. Let $z = \bar{u}_{-2}^\top x_{-2}/\tau$, and therefore $z$ has standard normal distribution. With this change of the variables, by the definition of the distribution $\mathcal{D}$, we have

$$\mathop{\mathbb{E}}_{(x,y)\sim\mathcal{D}} \varphi(x)[\bar{u}] \cdot y = \frac{1}{4} \mathop{\mathbb{E}}_{z\sim N(0,1)}[(\bar{u}_1 + \bar{u}_2 + \tau z)_+] + \frac{1}{4} \mathop{\mathbb{E}}_{z\sim N(0,1)}[(-\bar{u}_1 - \bar{u}_2 + \tau z)_+]$$
$$- \frac{1}{4} \mathop{\mathbb{E}}_{z\sim N(0,1)}[(\bar{u}_1 - \bar{u}_2 + \tau z)_+] - \frac{1}{4} \mathop{\mathbb{E}}_{z\sim N(0,1)}[(+\bar{u}_1 - \bar{u}_2 + \tau z)_+]$$

By claim D.4, and the 1-homogeneity of relu, we can simplify the above equation to

$$\mathop{\mathbb{E}}_{(x,y)\sim\mathcal{D}} \varphi(x)[\bar{u}] \cdot y = \frac{1}{4}\tau \cdot \left(2c_1 + O(\min\{|\bar{u}_1 + \bar{u}_2|/\tau, |\bar{u}_1 + \bar{u}_2|^2/\tau^2\})\right)$$
$$- \frac{\tau}{4}\left(2c_1 - O(\min\{|\bar{u}_1 - \bar{u}_2|/\tau, |\bar{u}_1 - \bar{u}_2|^2/\tau^2\}))\right)$$
$$\lesssim \min\{|\bar{u}_1| + |\bar{u}_2|, (|\bar{u}_1| + |\bar{u}_2|)^2/\tau\}$$

It follows that

$$\mathop{\mathbb{E}}_{\bar{u}\sim\mathbb{S}^{d-1}}\left[\mathop{\mathbb{E}}_{(x,y)\sim\mathcal{D}} \varphi(x)[\bar{u}] \cdot y\right]^2 \lesssim \mathbb{E}_{\bar{u}}\left[\min\{(|\bar{u}_1| + |\bar{u}_2|)^2, (|\bar{u}_1| + |\bar{u}_2|)^4/\|\bar{u}_{-2}\|_2^2\}\right]$$
$$\lesssim \mathbb{E}_{\bar{u}}\left[(|\bar{u}_1| + |\bar{u}_2|)^2 \cdot \mathbf{1}[\|\bar{u}_{-2}\|_2 \leq 1/2]\right] + \mathbb{E}_{\bar{u}}\left[(|\bar{u}_1| + |\bar{u}_2|)^4/\|\bar{u}_{-2}\|_2^2 \cdot \mathbf{1}[\|\bar{u}_{-2}\|_2 \geq 1/2]\right]$$
$$\lesssim \exp(-\sqrt{d}) + \mathbb{E}_{\bar{u}}\left[(|\bar{u}_1| + |\bar{u}_2|)^4\right] \lesssim 1/d^2 \tag{D.2}$$

Combining equation D.1 and equation D.2 we complete the proof of bullet 3. Next we prove bullet 1. Note that $\varphi(x)[\bar{u}]y$ is bounded by $|\bar{u}_1| + |\bar{u}_2| + \|\bar{u}_{-2}^\top x_{-2}\|_2$. Therefore, conditioned on $\|x_i\|_2 \lesssim d\log n$

$$\|W_i\|_2^2 \leq \mathop{\mathbb{E}}_{\bar{u}\sim\mathbb{S}^{d-1}}\left[(|\bar{u}_1| + |\bar{u}_2| + \|\bar{u}_{-2}^\top x_{-2}\|_2)^2\right]$$
$$\lesssim \mathop{\mathbb{E}}_{\bar{u}\sim\mathbb{S}^{d-1}}\left[|\bar{u}_1|^2\right] + \mathop{\mathbb{E}}_{\bar{u}\sim\mathbb{S}^{d-1}}\left[|\bar{u}_2|^2\right] + \mathop{\mathbb{E}}_{\bar{u}\sim\mathbb{S}^{d-1}}\left[\|\bar{u}_{-2}^\top x_{-2}\|_2^2\right]$$
$$\lesssim 1/d + \|x_{-2}\|_2^2/d \lesssim \log n$$

Hence we complete the proof. □

**Claim D.4.** *Let $Z \sim \mathcal{N}(0,1)$ and $a \in \mathbb{R}$. Then, there exists a universal constant $c_1$ and $c_2$ such that*
$$|\mathbb{E}[(a + Z)_+ + (-a + Z)_+] - 2c_1| \leq c_2 \min\{|a|, a^2\}.$$

*Proof.* Without loss of generality we can assume $a \geq 0$. Then,

$$\mathbb{E}[(a + Z)_+ + (-a + Z)_+] = \mathbb{E}[(a + Z)\mathbf{1}[Z \geq -a]] + \mathbb{E}[(Z - a)\mathbf{1}[Z \geq a]]$$
$$= \mathbb{E}[a \cdot \mathbf{1}[Z \geq -a]] + \mathbb{E}[Z \cdot \mathbf{1}[Z \geq -a]] - \mathbb{E}[a \cdot \mathbf{1}[Z \geq a]] + \mathbb{E}[Z \cdot \mathbf{1}[Z \geq a]]$$
$$= \mathbb{E}[a\mathbf{1}[-a \leq Z \leq a]] + 2\mathbb{E}[Z \cdot \mathbf{1}[Z \geq a]] \qquad (\text{by } \mathbb{E}[Z \cdot \mathbf{1}[-a \leq Z \leq a]] = 0)$$
$$= \mathbb{E}[a\mathbf{1}[-a \leq Z \leq a]] + 2\mathbb{E}[Z \cdot \mathbf{1}[Z \geq 0]] - 2\mathbb{E}[Z \cdot \mathbf{1}[a \geq Z \geq 0]]$$
$$= 2c_1 + O(\min\{a, a^2\})$$

where the last equality uses the fact that $c_1 \triangleq \mathbb{E}\left[Z \cdot \mathbf{1}[Z \geq 0]\right]$ and $\mathbb{E}\left[a\mathbf{1}[-a \leq Z \leq a]\right] \leq a\mathbb{E}\left[\mathbf{1}[-a \leq Z \leq a]\right] \lesssim a\min\{1, a\}$. □

Now we will prove Theorem 4.3.

*Proof of Theorem 4.3.* To circumvent the technical issue of bounded support in Proposition 3.1 and Lemma 4.2, we construct $p_{\text{data}}$ to be a slightly modified version of $\mathcal{D}$: perform rejection sampling of $(x, y) \sim \mathcal{D}$ until we obtain a sample with $\|x\|_2^2 \lesssim d\log n$. Since this occurs with very high probability, the high probability result of Lemma D.3 still translates to $p_{\text{data}}$. Now apply Lemma D.2 to conclude that $\gamma_{\ell_2} \lesssim \frac{\log n}{\sqrt{n}} + \frac{1}{d}$. Furthermore, Lemma D.1 allows us to conclude that $\gamma_{\ell_1} \gtrsim 1$.

We can therefore apply Proposition 3.1, and conclude that with probability $1 - \delta$,

$$L_{\ell_1\text{-svm}} \lesssim \sqrt{\frac{d\log n}{n}} + \sqrt{\frac{\log\log(d\log n)}{n}} + \sqrt{\frac{\log(1/\delta)}{n}}$$

Furthermore, plugging $\gamma_{\ell_2}$ into the bound of Lemma 4.2 gives us

$$\min\left\{1, d\sqrt{\frac{\log n}{n}}\right\} + \sqrt{\frac{\log\log(dn)}{n}} + \sqrt{\frac{\log(1/\delta)}{n}}$$

□

## D.2 REGRESSION

We will first define the 1-norm and 2-norm regression problems. The regression equivalent of equation 4.2 for $\alpha \in \mathcal{L}^1(\mathbb{S}^{d-1})$ is as follows:

$$\alpha_{\ell_1} \in \arg\min_\alpha \|\alpha\|_1$$
$$\text{subject to } \langle \alpha, \varphi(x_i) \rangle = y_i \tag{D.3}$$

Next we define the regression version of equation 4.3:

$$\alpha_{\ell_2} \in \arg\min_\alpha \|\alpha\|_2$$
$$\text{subject to } \langle \alpha, \varphi(x_i) \rangle = y_i \tag{D.4}$$

where $\alpha \in \mathcal{L}^2(\mathbb{S}^{d-1})$.

We will briefly motivate our study of the regression setting by connecting the minimum 1-norm solution to neural networks. To compare, in the classification setting, optimizing the weakly regularized loss over neural networks is equivalent to solving the $\ell_1$ SVM. In the regression setting, solving the weakly regularized squared error loss is equivalent is equivalent to finding the minimum 1-norm solution that fits the datapoints exactly.

**Theorem D.5.** *Let $f(\Theta; \cdot)$ be some two-layer neural network with $m \geq n$ hidden units parametrized by $\Theta$, as in Section 4. Define the $\lambda$-regularized squared error loss*

$$L_{\lambda,m}(\Theta) \triangleq \frac{1}{n}\sum_{i=1}^n (f(\Theta; x_i) - y_i)^2 + \lambda\|\Theta\|_2^2$$

*with $\Theta_{\lambda,m} \in \arg\min_\Theta L_{\lambda,m}(\Theta)$. Suppose that equation D.3 is feasible with optimal solution $\alpha_{\ell_1}$. Then as $\lambda \to 0$, $L_{\lambda,m}(\Theta_{\lambda,m}) \to 0$ and $\|\Theta_{\lambda,m}\|_2^2 \to 2\|\alpha_{\ell_1}\|_1$.*

*Proof.* We can see that equation D.3 will have a $n$-sparse solution $\alpha^\star$ using the same reasoning as the proof of Lemma B.1. Furthermore, following the proof of Proposition 4.1, the function $x \mapsto \langle \alpha^\star, \varphi(x) \rangle$ is implementable by a neural network $\Theta^{\star,m}$ with $\|\Theta^{\star,m}\|_2^2 = 2\|\alpha^\star\|_1 = 2\|\alpha_{\ell_1}\|_1$. Following the same reasoning as before, we can also conclude that $\Theta^{\star,m}$ is an optimal solution for:

$$\min_\Theta \|\Theta\|_2^2$$
$$\text{subject to } f(\Theta; x_i) = y_i \tag{D.5}$$

Now we note that $\lambda\|\Theta_{\lambda,m}\|_2^2 \le L_{\lambda,m}(\Theta_{\lambda,m}) \le L_{\lambda,m}(\Theta^{\star,m}) = \lambda\|\Theta^{\star,m}\|_2^2$, so as $\lambda \to 0$, and also $\|\Theta_{\lambda,m}\|_2 \le \|\Theta^{\star,m}\|_2$. Now assume for the sake of contradiction that $\exists B$ with $\|\Theta_{\lambda,m}\|_2 \le B < \|\Theta^{\star,m}\|_2$ for arbitrarily small $\lambda$. We define

$$r^\star \triangleq \min_{\Theta} \frac{1}{n} \sum_{i=1}^{n} (f(\Theta; x_i) - y_i)^2$$
$$\text{subject to } \|\Theta\|_2 \le B$$

Note that $r^\star > 0$ since $\Theta^{\star,m}$ is optimal for equation D.5. However, $L_{\lambda,m} \ge r^\star$ for arbitrarily small $\lambda$, a contradiction. Thus, $\lim_{\lambda\to 0} \|\Theta^{\star,m}\|_2^2 = \|\Theta^{\star,m}\|_2^2$. $\qquad\square$

We proceed to provide similar generalization bounds as the classification setting. This time, our bounds depend on the norms of the solution rather than the margin. Let $f^\phi(\alpha; \cdot) \triangleq x \mapsto \langle \alpha, \varphi(x) \rangle$ (for $\varphi(x)$ defined in equation 4.1). Following the convention in Section C.2, define hypothesis class $\mathcal{F}_B^{1,\phi} \triangleq \{f^\phi(\alpha; \cdot) : \alpha \in \mathcal{L}^1(\mathbb{S}^{d-1}), \|\alpha\|_1 \le B\}$ of linear functionals bounded by $B$ in 1-norm. As before, define $\mathcal{F}_B^{2,\phi}$ and let $\hat{\mathfrak{R}}(\mathcal{F})$ denote the empirical Rademacher complexity of hypothesis class $\mathcal{F}$.

We will first derive a Rademacher complexity bound for $\mathcal{F}_B^{1,\phi}$.

**Claim D.6.** $\hat{\mathfrak{R}}(\mathcal{F}_B^{1,\phi}) \le \frac{1}{n} B \mathbb{E}_{\epsilon_i} \left[ \|\sum_{i=1}^{n} \epsilon_i \varphi(x_i)\|_\infty \right].$

*Proof.* We write

$$\hat{\mathfrak{R}}(\mathcal{F}_B^{1,\phi}) = \frac{1}{n} \mathbb{E}_{\epsilon_i} \left[ \sup_{\alpha \in \mathcal{F}_B^{1,\phi}} \langle \alpha, \sum_{i=1}^{n} \epsilon_i \varphi(x_i) \rangle \right]$$
$$\le \frac{1}{n} \mathbb{E}_{\epsilon_i} \left[ \sup_{\alpha \in \mathcal{F}_B^{1,\phi}} \|\alpha\|_1 \left\| \sum_{i=1}^{n} \epsilon_i \varphi(x_i) \right\|_\infty \right]$$
$$\le \frac{1}{n} B \cdot \mathbb{E}_{\epsilon_i} \left[ \left\| \sum_{i=1}^{n} \epsilon_i \varphi(x_i) \right\|_\infty \right]$$

$\qquad\square$

We will now complete the bound on $\hat{\mathfrak{R}}(\mathcal{F}_B^{1,\phi})$ for Lipschitz activations $\phi$ with $\phi(0) = 0$.

**Claim D.7.** *Suppose that our activation $\phi$ is $M$-Lipschitz and $\phi(0) = 0$. Then*

$$\hat{\mathfrak{R}}(\mathcal{F}_B^{1,\phi}) \le \frac{3BM\sqrt{\sum_{i=1}^{n} \|x_i\|_2^2}}{n}$$

*Proof.* We will show that

$$\mathbb{E}_{\epsilon_i} \left[ \left\| \sum_{i=1}^{n} \epsilon_i \varphi(x_i) \right\|_\infty \right] \le 3M \sqrt{\sum_{i=1}^{n} \|x_i\|_2^2}$$

from which the statement of the lemma follows via Claim D.6. Fix any $\bar{u}' \in \mathbb{S}^{d-1}$. Then we get the decomposition

$$\mathbb{E}_{\epsilon_i} \left[ \sup_{\bar{u} \in \mathbb{S}^{d-1}} \left| \sum_{i=1}^{n} \epsilon_i \phi(\bar{u}^\top x_i) \right| \right] \le \mathbb{E}_{\epsilon_i} \left[ \left| \sum_{i=1}^{n} \epsilon_i \phi(\bar{u}'^\top x_i) \right| \right] +$$
$$\mathbb{E}_{\epsilon_i} \left[ \sup_{\bar{u} \in \mathbb{S}^{d-1}} \left| \sum_{i=1}^{n} \epsilon_i \phi(\bar{u}^\top x_i) \right| - \inf_{\bar{u} \in \mathbb{S}^{d-1}} \left| \sum_{i=1}^{n} \epsilon_i \phi(\bar{u}^\top x_i) \right| \right] \quad \text{(D.6)}$$

We can bound the first term as

$$
E_{\epsilon_i}\left[\left|\sum_{i=1}^n \epsilon_i \phi(\bar{u}'^\top x_i)\right|\right] \le \sqrt{E_{\epsilon_i}\left[\left(\sum_{i=1}^n \epsilon_i \phi(\bar{u}'^\top x_i)\right)^2\right]}
$$

$$
\le \sqrt{\sum_{i=1}^n \phi(\bar{u}'^\top x_i)^2}
$$

$$
\le M\sqrt{\sum_{i=1}^n (\bar{u}'^\top x_i)^2} \qquad \text{(since $\phi$ is Lipschitz and $\phi(0)=0$)}
$$

$$
\le M\sqrt{\sum_{i=1}^n \|x_i\|_2^2} \tag{D.7}
$$

We note that the second term of equation D.6 can be bounded by

$$
\mathbb{E}_{\epsilon_i}\left[\sup_{\bar{u}\in\mathbb{S}^{d-1}}\left|\sum_{i=1}^n \epsilon_i \phi(\bar{u}^\top x_i)\right| - \inf_{\bar{u}\in\mathbb{S}^{d-1}}\left|\sum_{i=1}^n \epsilon_i \phi(\bar{u}^\top x_i)\right|\right]
$$

$$
\le \mathbb{E}_{\epsilon_i}\left[\sup_{\bar{u}\in\mathbb{S}^{d-1}}\sum_{i=1}^n \epsilon_i \phi(\bar{u}^\top x_i) - \inf_{\bar{u}\in\mathbb{S}^{d-1}}\sum_{i=1}^n \epsilon_i \phi(\bar{u}^\top x_i)\right]
$$

This follows from the general fact that the difference between the supremum and infimum of the absolute value of a quantity is bounded by the difference between the supremum and the infimum. Furthermore, by symmetry of the Rademacher random variables,

$$
\mathbb{E}_{\epsilon_i}\left[\sup_{\bar{u}\in\mathbb{S}^{d-1}}\sum_{i=1}^n \epsilon_i \phi(\bar{u}^\top x_i) - \inf_{\bar{u}\in\mathbb{S}^{d-1}}\sum_{i=1}^n \epsilon_i \phi(\bar{u}^\top x_i)\right] \le 2\mathbb{E}_{\epsilon_i}\left[\sup_{\bar{u}\in\mathbb{S}^{d-1}}\sum_{i=1}^n \epsilon_i \phi(\bar{u}^\top x_i)\right] \tag{D.8}
$$

This simply gives an empirical Rademacher complexity of the hypothesis class $\mathcal{F} \triangleq \{x \mapsto \phi(\bar{u}^\top x) : \bar{u} \in \mathbb{S}^{d-1}\}$ scaled by $n$. By the Lipschitz contraction property of Rademacher complexity, using the fact that $\phi$ is $M$-Lipschitz, we can therefore bound equation D.8 by

$$
2\mathbb{E}_{\epsilon_i}\left[\sup_{\bar{u}\in\mathbb{S}^{d-1}}\sum_{i=1}^n \epsilon_i \phi(\bar{u}^\top x_i)\right] \le 2M\sqrt{\sum_{i=1}^n \|x_i\|_2^2} \tag{D.9}
$$

Plugging equation D.7 and equation D.9 back into equation D.6 gives the desired bound. $\qquad\square$

The following is a generalization bound based on the 1-norm:

**Lemma D.8.** *Let $l(\cdot;y) : \mathbb{R} \to [-c, c]$ be a bounded $M$-Lipschitz loss function. Assume that $\phi$ is a 1-Lipschitz activation with $\phi(0) = 0$. Let $(x_i, y_i)_{i=1}^n$ be drawn i.i.d from $p_{\text{data}}$. Then with probability at least $1 - \delta$ over the dataset, every $\alpha \in \mathcal{L}^1(\mathbb{S}^{d-1})$ satisfies*

$$
\mathbb{E}_{(x,y)\sim p_{\text{data}}}[l(f^\phi(\alpha;x);y)] \le
$$
$$
\frac{1}{n}\sum_{i=1}^n l(f^\phi(\alpha;x_i);y_i) + 12M\frac{\max\{1, \|\alpha\|_1\|X\|_F\}}{n} + c\sqrt{\frac{\log(1/\delta) + \log(\max\{1, 2\|\alpha\|_1\|X\|_F\})}{2n}}
$$

*Proof.* Our starting point is Theorem 1 of Kakade et al. (2009), which states that with probability $1 - \delta$, for any fixed hypothesis class $\mathcal{F}$ and $f \in \mathcal{F}$,

$$
\mathbb{E}_{(x,y)\sim p_{\text{data}}}[l(f(x);y)] \le \frac{1}{n}\sum_{i=1}^n l(f(x_i);y_i) + 2M\hat{\mathfrak{R}}(\mathcal{F}) + c\sqrt{\frac{\log(1/\delta)}{n}} \tag{D.10}
$$

We define $B_j \triangleq \frac{2^j}{\|X\|_F}$ for $j \geq 0$. We note that by Claim D.7, $\hat{\Re}(\mathcal{F}_{B_j}^{1,\phi}) \leq 3\frac{2^j}{n}$. and apply the above on $\mathcal{F}_{B_j}^{1,\phi}$ using $\delta_j \triangleq \frac{\delta}{2^{j+1}}$. Then using a union bound, with probability $1 - \sum_{j=0}^{\infty} \delta_j = 1 - \delta$, for all $j \geq 0$ and $f^\phi(\alpha; \cdot) \in \mathcal{F}_{B_j}^{1,\phi}$

$$\mathbb{E}_{(x,y) \sim p_{\text{data}}}[l(f^\phi(\alpha; x); y)] \leq \frac{1}{n} \sum_{i=1}^{n} l(f^\phi(\alpha; x_i); y_i) + 2M\hat{\Re}(\mathcal{F}_{B_j}^{1,\phi}) + c\sqrt{\frac{\log(1/\delta_j)}{n}}$$

$$\leq \frac{1}{n} \sum_{i=1}^{n} l(f^\phi(\alpha; x_i); y_i) + 6M\frac{2^j}{n} + c\sqrt{\frac{\log(1/\delta) + \log(2^{j+1})}{n}}$$

Now for every $\alpha$ with $\|\alpha\|_1 < \frac{1}{\|X\|_F}$, we use the inequality for $\mathcal{F}_{B_0}^{1,\phi}$, and for every other $\alpha$, we apply the inequality corresponding to $\mathcal{F}_{B_{j+1}}^{1,\phi}$, where $2^j \leq \|\alpha\|_1 \|X\|_F \leq 2^{j+1}$. This gives the desired statement. $\qquad \square$

We can also provide the same generalization error bound for the 2-norm and relu features:

**Lemma D.9.** *In the setting of Lemma D.8, choose $\phi$ to be the* relu *activation. Then with probability $1 - \delta$, every $\alpha \in \mathcal{L}^2(\mathbb{S}^{d-1})$ satisfies*

$$\mathbb{E}_{(x,y) \sim p_{\text{data}}}[l(f^\phi(\alpha; x); y)] \lesssim$$
$$\frac{1}{n} \sum_{i=1}^{n} l(f^\phi(\alpha; x_i); y_i) + M\frac{\sqrt{\kappa}\max\{1, \|\alpha\|_2\|X\|_F\}}{n\sqrt{d}} + c\sqrt{\frac{\log(1/\delta) + \log(\max\{1, \|\alpha\|_2\|X\|_F\})}{2n}}$$

*Proof.* We proceed the same way as in the proof of Lemma D.8. We define $B_j$ as before, and this time have $\hat{\Re}(\mathcal{F}_{B_j}^{2,\phi}) \lesssim \frac{\sqrt{\kappa}2^j}{n\sqrt{d}}$ from Corollary C.4. Thus, again union bounding over all $j$, equation equation D.10 gives with probability $1 - \delta$, for all $j \geq 0$ and $f^\phi(\alpha; \cdot) \in \mathcal{F}_{B_j}^{2,\phi}$

$$\mathbb{E}_{(x,y) \sim p_{\text{data}}}[l(f^\phi(\alpha; x); y)] \lesssim \frac{1}{n} \sum_{i=1}^{n} l(f^\phi(\alpha; x_i); y_i) + M\frac{\sqrt{\kappa}2^j}{n\sqrt{d}} + c\sqrt{\frac{\log(1/\delta) + \log(2^{j+1})}{n}}$$

Now we assign the $\alpha$ to different $j$ as before to obtain the statement in the lemma. $\qquad \square$

Note that if $l$ is some bounded loss such that $l(y; y) = 0$ (for example, truncated squared error), for $\alpha_{\ell_1}$ and $\alpha_{\ell_2}$ the loss terms over the datapoints (in the bounds of Lemmas D.8 and D.9) vanish. For loss $l$, define

$$L_{\ell_1\text{-reg}} \triangleq \mathbb{E}_{x,y \sim p_{\text{data}}}[l(f^\phi(\alpha_{\ell_1}; x); y)]$$
$$L_{\ell_2\text{-reg}} \triangleq \mathbb{E}_{x,y \sim p_{\text{data}}}[l(f^\phi(\alpha_{\ell_2}; x); y)]$$

Next, we will define the kernel matrix $K$ with $K_{ij} = \langle \varphi(x_i), \varphi(x_j) \rangle$. Now we are ready to state and prove the formal theorem describing the gap between the 1-norm solution and 2-norm solution.

**Theorem D.10.** *Recall the definitions of $\alpha_{\ell_1}$ and $\alpha_{\ell_2}$ in equation D.3 and equation D.4. For any activation $\phi$ with the property that $K$ is full rank for any $X$ with no repeated datapoints, there exists a distribution $p_{\text{data}}$ such that with probability 1,*

$$\|\alpha_{\ell_1}\|_1 \leq 1$$

*On the other hand,*

$$\mathbb{E}_{(x_i,y_i)_{i=1}^n \sim_{iid} p_{\text{data}}}[\|\alpha_{\ell_2}\|_2^2] = \frac{n}{\kappa}$$

*For i.i.d samples from this choice of $p_{\text{data}}$, if $l$ is bounded ($l(\cdot; y) : \mathbb{R} \to [-1, 1]$), 0 on correct predictions ($l(y; y) = 0$), and 1-Lipschitz, then with probability $1 - \delta$,*

$$L_{\ell_1\text{-reg}} \lesssim \sqrt{\frac{d}{n}} + \sqrt{\frac{\log(1/\delta) + \log n}{n}}$$

*Meanwhile, in the case that $\|\alpha_{\ell_2}\|_2^2 \geq \frac{n}{\kappa}$, the upper bound on $L_{\ell_2\text{-reg}}$ from Lemma D.9 is $\Omega(1)$ and in particular does not decrease with $n$.*

We will first show that for any dataset $X$, there is a distribution over $Y$ such that the expectation of $\|\alpha_{\ell_2}\|_2$ is large. When it is clear from context, $y$ will denote the vector corresponding to $Y$.

**Lemma D.11.** *There is a distribution $\mathcal{A}$ over $\mathcal{L}^1(\mathbb{S}^{d-1})$ such that for any dataset $X$ with $y_i \triangleq \langle \varphi(x_i), \beta \rangle$ for $\beta \sim \mathcal{A}$,*

$$\mathbb{E}_\beta[\|\alpha_{\ell_2}\|_2^2] \geq \frac{n}{\kappa}$$

*and with probability 1,*

$$\|\alpha_{\ell_1}\|_1 \leq 1$$

We note the order of the quantifiers in Lemma D.11: the distribution $\mathcal{A}$ must not depend on the dataset $X$. We first provide a simple closed-form expression for $\|\alpha_{\ell_2}\|_2^2$.

**Claim D.12.** *If $K$ is full rank, then $\|\alpha_{\ell_2}\|_2^2 = y^\top K^{-1} y$.*

*Proof.* This follows by taking the dual of equation D.4. $\qquad\square$

*Proof of Lemma D.11.* We sample $\beta \sim \mathcal{A}$ as follows: first sample $\bar{u} \sim \mathbb{S}^{d-1}$ uniformly. Then set $\beta$ to have a delta mass of 1 at $\bar{u}$ and be 0 everywhere else. Define the vector $v_{\bar{u}} \triangleq [\phi(\bar{u}^\top x_1) \cdots \phi(\bar{u}^\top x_n)]$; then it follows that we set our labels $y$ to $v_{\bar{u}}$. It is immediately clear that $\|\alpha_{\ell_1}\|_1 \leq \|\beta\|_1 \leq 1$.

To lower bound $\mathbb{E}_\beta[\|\alpha_{\ell_2}\|_2^2]$, from Claim D.12 we get

$$\begin{aligned}
\mathbb{E}_{\beta \sim \mathcal{A}}[\|\alpha_{\ell_2}\|_2^2] &= E_{\bar{u} \sim \mathbb{S}^{d-1}}[v_{\bar{u}}^\top K^{-1} v_{\bar{u}}] \\
&= E_{\bar{u} \sim \mathbb{S}^{d-1}}[\text{trace} K^{-1}(v_{\bar{u}} v_{\bar{u}}^\top)] \\
&= \text{trace}(K^{-1} E_{\bar{u} \sim \mathbb{S}^{d-1}}[v_{\bar{u}} v_{\bar{u}}^\top] \\
&= \frac{1}{\kappa}\text{trace}(K^{-1} K) \qquad\qquad \text{(by definition of } K) \\
&= \frac{n}{\kappa}
\end{aligned}$$

$\square$

*Proof of Theorem D.10.* We note that since the distribution $\mathcal{A}$ of Lemma D.11 does not depend on the dataset $X$, it must follow that

$$\begin{aligned}
\mathbb{E}_{(x_i)_{i=1}^n \sim \text{iid} \mathcal{N}(0, I_{d \times d})} \left[ \mathbb{E}_{\beta \sim \mathcal{A}}[\|\alpha_{\ell_2}\|_2^2] \right] &= \frac{n}{\kappa} \\
\mathbb{E}_{\beta \sim \mathcal{A}} \left[ \mathbb{E}_{(x_i)_{i=1}^n \sim \text{iid} \mathcal{N}(0, I_{d \times d})}[\|\alpha_{\ell_2}\|_2^2] \right] &= \frac{n}{\kappa}
\end{aligned}$$

Thus, there exists $\beta^\star$ such that if we sample $X$ i.i.d. from the standard normal and set $y_i = \langle \varphi(x_i), \beta^\star \rangle$, the expectation of $\|\alpha_{\ell_2}\|_2^2$ is at least $\frac{n}{\kappa}$. We choose $p_{\text{data}}$ corresponding to this $\beta^\star$, with $x$ sampled from the standard normal. Now it is clear that $p_{\text{data}}$ will satisfy the norm conditions of Theorem D.10.

For the generalization bounds, with high probability $\|X\|_F = \Theta(\sqrt{nd})$ as $x$ is sampled from the standard normal distribution. Thus, Lemma D.8 immediately gives the desired generalization error bounds for $L_{\ell_1\text{-reg}}$. On the other hand, if $\|\alpha_{\ell_2}\|_2 \geq \sqrt{\frac{n}{\kappa}}$, then the bound of Lemma D.9 is at least

$$\frac{\sqrt{\kappa}\|\alpha_{\ell_2}\|_2 \|X\|_F}{n\sqrt{d}} \geq \Omega(1)$$

$\square$

# E MISSING PROOFS IN SECTION 5

## E.1 DETAILED SETUP

We first write our regularity assumptions on $\Phi$, $R$, and $V$ in more detail:

**Assumption E.1** (Regularity conditions on $\Phi$, $R$, $V$). *$R$ is convex, nonnegative, Lipschitz, and smooth: $\exists M_R, C_R$ such that $\|\nabla^2 R\|_{op} \leq C_R$, and $\|\nabla R\|_2 \leq M_R$.*

**Assumption E.2.** *$\Phi$ is differentiable, bounded and Lipschitz on the sphere: $\exists B_\Phi, M_\Phi$ such that $\|\Phi(\bar{\theta})\| \leq B_\Phi \; \forall \bar{\theta} \in \mathbb{S}^d$, and $|\Phi_i(\bar{\theta}) - \Phi_i(\bar{\theta}')| \leq M_\Phi \|\bar{\theta} - \bar{\theta}'\|_2 \; \forall \bar{\theta}, \bar{\theta}' \in \mathbb{S}^d$.*

**Assumption E.3.** *$V$ is Lipschitz and upper and lower bounded on the sphere: $\exists b_V, B_V, M_V$ such that $0 < b_V \leq V(\bar{\theta}) \leq B_V \; \forall \bar{\theta} \in \mathbb{S}^d$, and $\|\nabla V(\bar{\theta})\|_2 \leq M_V \; \forall \bar{\theta} \in \mathbb{S}^d$.*

We state the version of Theorem 5.3 that collects these parameters:

**Theorem E.4** (Theorem 5.3 with problem parameters). *Suppose that $\Phi$ and $V$ are 2-homogeneous and Assumptions E.1, E.2, and E.3 hold. Fix a desired error threshold $\epsilon > 0$. Suppose that from a starting distribution $\rho_0$, a solution to the dynamics in equation 5.2 exists. Choose*

$$\sigma \triangleq \exp(-d \log(1/\epsilon)\mathrm{poly}(k, M_V, M_R, M_\Phi, b_V, B_V, C_R, B_\Phi, L[\rho_0] - L^\star))$$

$$t_\epsilon \triangleq \frac{d^2}{\epsilon^4}\mathrm{poly}(\log(1/\epsilon), k, M_V, M_R, M_\Phi, b_V, B_V, C_R, B_\Phi, L[\rho_0] - L^\star)$$

*Then it must hold that $\min_{0 \leq t \leq t_\epsilon} L[\rho_t] - \inf_\rho L[\rho] \leq 2\epsilon$.*

### E.2 PROOF OF THEOREM E.4

Throughout the proof, it will be useful to keep track of $W_t \triangleq \sqrt{\mathbb{E}_{\theta \sim \rho_t}[\|\theta\|_2^2]}$, the second moment of $\rho_t$. We first introduce a general lemma on integrals over vector field divergences.

**Lemma E.5.** *For any $h_1 : \mathbb{R}^{d+1} \to \mathbb{R}$, $h_2 : \mathbb{R}^{d+1} \to \mathbb{R}^{d+1}$ and distribution $\rho$ with $\rho(\theta) \to 0$ as $\|\theta\| \to \infty$,*

$$\int h_1(\theta)\nabla \cdot (h_2(\theta)\rho(\theta))d\theta = -E_{\theta \sim \rho}[\langle \nabla h_1(\theta), h_2(\theta)\rangle]$$

*Proof.* The proof follows from integration by parts. $\square$

We note that $\rho_t$ will satisfy the boundedness condition of Lemma E.5 during the course of our algorithm - $\rho_0$ starts with this property, and Lemma E.9 proves that $\rho_t$ will continue to have this property. We therefore freely apply Lemma E.5 in the remaining proofs. We first bound the absolute value of $L'[\rho_t]$ over the sphere by $B_L \triangleq M_R B_\Phi + B_V$.

**Lemma E.6.** *For any $\bar{\theta} \in \mathbb{S}^{d-1}, t \geq 0$, $|L'[\rho_t](\bar{\theta})| \leq \triangleq B_L$.*

*Proof.* We compute

$$|L'[\rho_t](\bar{\theta})| = \left|\left\langle \nabla R\left(\int \Phi d\rho\right), \Phi(\bar{\theta})\right\rangle + V(\bar{\theta})\right|$$

$$\leq \left\|\nabla R\left(\int \Phi d\rho\right)\right\|_2 \|\Phi(\bar{\theta})\|_2 + V(\bar{\theta}) \leq M_R B_\Phi + B_V$$

$\square$

Now we analyze the decrease in $L[\rho_t]$.

**Lemma E.7.** *Under the perturbed Wasserstein gradient flow*

$$\frac{d}{dt}L[\rho_t] = -\sigma\mathbb{E}_{\theta \sim \rho_t}[L'[\rho_t](\theta)] + \sigma\mathbb{E}_{\bar{\theta} \sim U^d}[L'[\rho_t](\bar{\theta})] - \mathbb{E}_{\theta \sim \rho_t}[\|v[\rho_t](\theta)\|_2^2]$$

*Proof.* Applying the chain rule, we can compute

$$\frac{d}{dt}L[\rho_t] = \left\langle \nabla R\left(\int \Phi d\rho_t\right), \frac{d}{dt}\int \Phi d\rho_t\right\rangle + \frac{d}{dt}\int V d\rho_t$$

$$= \frac{d}{dt}\mathbb{E}_{\theta \sim \rho_t}[L'[\rho_t](\theta)]$$

$$= \int L'[\rho_t](\theta)\rho'_t(\theta)d\theta$$

$$= -\sigma \int L'[\rho_t]d\rho_t + \sigma \int L'[\rho_t]dU^d - \int L'[\rho_t](\theta)\nabla \cdot (v[\rho_t](\theta)\rho_t(\theta))d\theta$$

$$= -\sigma\mathbb{E}_{\theta \sim \rho_t}[L'[\rho_t](\theta)] + \sigma\mathbb{E}_{\bar{\theta} \sim U^d}[L'[\rho_t](\bar{\theta})] - \mathbb{E}_{\theta \sim \rho_t}[\|v[\rho_t](\theta)\|_2^2],$$

where we use Lemma E.5 with $h_1 = L'[\rho_t]$ and $h_2 = v[\rho_t]$. $\qquad \square$

Now we show that the decrease in objective value is approximately the average velocity of all parameters under $\rho_t$ plus some additional noise on the scale of $\sigma$. At the end, we choose $\sigma$ small enough so that the noise terms essentially do not matter.

**Corollary E.8.** *We can bound $\frac{d}{dt}L[\rho_t]$ by*

$$\frac{d}{dt}L[\rho_t] \leq \sigma B_L(W_t^2 + 1) - \mathbb{E}_{\theta \sim \rho_t}[\|v[\rho_t](\theta)\|_2^2] \tag{E.1}$$

*Proof.* By homogeneity, and Lemma E.6, $\mathbb{E}_{\theta \sim \rho_t}[L'[\rho_t](\theta)] = \mathbb{E}_{\theta \sim \rho_t}[L'[\rho_t](\bar{\theta})\|\theta\|_2^2] \leq B_L W_t^2$. We also get $\mathbb{E}_{\bar{\theta} \sim U^d}[L'[\rho_t](\bar{\theta})] \leq B_L$ since $U^d$ is only supported on $\mathbb{S}^d$. Combining these with Lemma E.7 gives the desired statement. $\qquad \square$

Now we show that if we run the dynamics for a short time, the second moment of $\rho_t$ will grow slowly, again at a rate that is roughly the scale of the noise $\sigma$.

**Lemma E.9.** *For all $0 \leq t' \leq t$, $W_{t'}^2 \leq \frac{L[\rho_0]+\sigma t B_L}{b_V - t\sigma B_L}$.*

*Proof.* Let $t^* \triangleq \arg\max_{t' \in [0,t]} W_{t'}^2$. Integrating both sides of equation E.1, and rearranging, we get

$$0 \leq \int_0^{t^*} \mathbb{E}_{\theta \sim \rho_s}[\|v[\rho_s](\theta)\|_2^2]ds \leq L[\rho_0] - L[\rho_t] + \sigma B_L \int_0^{t^*} (W_s^2 + 1)ds$$

$$\leq L[\rho_0] - L[\rho_{t^*}] + t^*\sigma B_L(W_{t^*}^2 + 1)$$

Now since $R$ is nonnegative, we apply $L[\rho_{t^*}] \geq E_{\theta \sim \rho_{t^*}}[V(\theta)] \geq E_{\theta \sim \rho_{t^*}}[V(\bar{\theta})\|\theta\|_2^2] \geq b_V W_{t^*}^2$. We now plug this in and rearrange to get $W_{t'}^2 \leq W_{t^*}^2 \leq \frac{L[\rho_0]+t^*\sigma B_L}{b_V - t^*\sigma B_L} \leq \frac{L[\rho_0]+t\sigma B_L}{b_V - t\sigma B_L} \forall 0 \leq t' \leq t.$ $\qquad \square$

Now let $W_\epsilon^2 \triangleq \frac{L[\rho_0]+\sigma t_\epsilon B_L}{b_v - t_\epsilon \sigma B_L}$. By Lemma E.9, $\forall 0 \leq t \leq t_\epsilon$, $W_t^2 \leq W_\epsilon^2$.

The next statement allows us to argue that our dynamics will never increase the objective by too much.

**Lemma E.10.** *For any $t_1, t_2$ with $0 \leq t_1 \leq t_2 \leq t_\epsilon$, $L[\rho_{t_2}] - L[\rho_{t_1}] \leq \sigma(t_2 - t_1)B_L(W_\epsilon^2 + 1)$.*

*Proof.* From Corollary E.8, $\forall t \in [t_1, t_2]$ we have

$$\frac{d}{dt}L[\rho_t] \leq \sigma B_L(W_\epsilon^2 + 1)$$

Integrating from $t_1$ to $t_2$ gives the desired result. $\qquad \square$

The following lemma bounds the change in expectation of a 2-homogeneous function over $\rho_t$. At a high level, we lower bound the decrease in our loss as a function of the change in this expectation.

**Lemma E.11.** *Let* $h : \mathbb{R}^{d+1} \to \mathbb{R}$ *that is 2-homogeneous, with* $\|\nabla h(\bar\theta)\| \leq M \; \forall \bar\theta \in \mathbb{S}^d$ *and* $|h(\bar\theta)| \leq B \; \forall \bar\theta \in \mathbb{S}^d$. *Then* $\forall 0 \leq t \leq t_\epsilon$, *we have*

$$\left| \frac{d}{dt} \int h d\rho \right| \leq \sigma B(W_\epsilon^2 + 1) + MW \left( -\frac{d}{dt} L[\rho_t] + \sigma B_L(W_\epsilon^2 + 1) \right)^{1/2} \tag{E.2}$$

*Proof.* Let $Q(t) \triangleq \int h d\rho_t$. We can compute:

$$
\begin{aligned}
Q'(t) &= \int h(\theta) \frac{d\rho_t}{dt}(\theta) d\theta \\
&= \int h(\theta)(-\sigma\rho_t(\theta) - \nabla \cdot (v[\rho_t](\theta)\rho_t(\theta))) d\theta + \sigma \int h dU^d \\
&= -\sigma \int h(\bar\theta)\|\theta\|_2^2 \rho_t(\theta) d\theta + \sigma \int h dU^d - \int h(\theta)\nabla \cdot (v[\rho_t](\theta)\rho_t(\theta)) d\theta \tag{E.3}
\end{aligned}
$$

Note that the first two terms are bounded by $\sigma B(W_\epsilon^2 + 1)$ by the assumptions for the lemma. For the third term, we have from Lemma E.5:

$$
\begin{aligned}
\left| \int h(\theta)\nabla \cdot (v[\rho_t](\theta)\rho_t(\theta)) d\theta \right| &= |E_{\theta \sim \rho_t}[\langle \nabla h(\theta), v[\rho_t](\theta) \rangle]| \\
&\leq \sqrt{E_{\theta \sim \rho_t}[\|\nabla h(\theta)\|_2^2] E_{\theta \sim \rho_t}[\|v[\rho_t](\theta)\|_2^2]} && \text{(by Cauchy-Schwarz)} \\
&\leq \sqrt{E_{\theta \sim \rho_t}[\|\nabla h(\bar\theta)\|_2^2 \|\theta\|_2^2] E_{\theta \sim \rho_t}[\|v[\rho_t](\theta)\|_2^2]} && \text{(by homogeneity of } \nabla h) \\
&\leq MW_\epsilon \sqrt{E_{\theta \sim \rho_t}[\|v[\rho_t](\theta)\|_2^2]} && \text{(since } h \text{ is Lipschitz on the sphere)} \\
&\leq MW_\epsilon \left( -\frac{d}{dt} L[\rho_t] + \sigma B_L(W_\epsilon^2 + 1) \right)^{1/2} && \text{(by Corollary E.8)}
\end{aligned}
$$

Plugging this into equation E.3, we get that

$$|Q'(t)| \leq \sigma B(W_\epsilon^2 + 1) + MW_\epsilon \left( -\frac{d}{dt} L[\rho_t] + \sigma B_L(W_\epsilon^2 + 1) \right)^{1/2}$$

$\square$

We apply this result to bound the change in $L'[\rho_t]$ over time in terms of the change of the objective value. For clarity, we write the bound in terms of $c_1$ that is some polynomial in the problem constants.

**Lemma E.12.** *Define* $Q(t) \triangleq \int \Phi d\rho_t$. *For every* $\bar\theta \in \mathbb{S}^d$ *and* $0 \leq t \leq t + l \leq t_\epsilon$, $\exists c_1 \triangleq \text{poly}(k, C_R, B_\Phi, M_\Phi, B_L)$ *such that*

$$|L'[\rho_t](\bar\theta) - L'[\rho_{t+l}](\bar\theta)| \leq C_R B_\Phi \int_t^{t+l} \|Q'(t)\|_1 \tag{E.4}$$

$$\leq \sigma l c_1 (W_\epsilon^2 + 1) + c_1 W_\epsilon \sqrt{l} (L[\rho_t] - L[\rho_{t+l}] + \sigma l c_1 (W_\epsilon^2 + 1))^{1/2} \tag{E.5}$$

*Proof.* Recall that $L'[\rho_t](\bar\theta) = \langle \nabla R(\int \Phi d\rho_t), \Phi(\bar\theta) \rangle + V(\bar\theta)$. Differentiating with respect to $t$,

$$
\begin{aligned}
\frac{d}{dt} L'[\rho_t](\bar\theta) &= \left\langle \frac{d}{dt} \nabla R \left( \int \Phi d\rho_t \right), \Phi(\bar\theta) \right\rangle \\
&= \Phi(\bar\theta)^\top \nabla^2 R(Q(t)) Q'(t) \\
&\leq C_R B_\Phi \|Q'(t)\|_2 \\
&\leq C_R B_\Phi \|Q'(t)\|_1 \tag{E.6}
\end{aligned}
$$

Integrating and applying the same reasoning to $-L'[\rho_t]$ gives us equation E.4. Now we apply Lemma E.11 to get

$$
\begin{aligned}
\|Q'(t)\|_1 &= \sum_{i=1}^{k} \left| \frac{d}{dt} \int \Phi_i d\rho_t \right| \\
&\leq \sum_{i=1}^{k} \left[ \sigma B_\Phi (W_\epsilon^2 + 1) + M_\Phi W_\epsilon \left( -\frac{d}{dt} L[\rho_t] + \sigma B_L (W_\epsilon^2 + 1) \right)^{1/2} \right] \\
&\leq k\sigma B_\Phi (W_\epsilon^2 + 1) + k M_\Phi W_\epsilon \left( -\frac{d}{dt} L[\rho_t] + \sigma B_L (W_\epsilon^2 + 1) \right)^{1/2}
\end{aligned}
$$

We plug this into equation E.6 and then integrate both sides to obtain

$$
\begin{aligned}
C_R B_\Phi \int_t^{t+l} & \|Q'(t)\|_1 \\
&\leq k\sigma l C_R B_\Phi^2 (W_\epsilon^2 + 1) + k C_R B_\Phi M_\Phi W_\epsilon \int_t^{t+l} \left( -\frac{d}{dt} L[\rho_t] + \sigma B_L (W_\epsilon^2 + 1) \right)^{1/2} \\
&\leq k\sigma l C_R B_\Phi^2 (W_\epsilon^2 + 1) + k C_R B_\Phi M_\Phi W_\epsilon \sqrt{l} (L[\rho_t] - L[\rho_{t+l}] + \sigma l B_L (W_\epsilon^2 + 1))^{1/2}
\end{aligned}
$$

Using $c_1 \triangleq \max\{k C_R B_\Phi^2, k C_R B_\Phi M_\Phi, B_L\}$ gives the statement in the lemma. $\qquad\square$

Now we also show that $L'$ is Lipschitz on the unit ball. For clarity, we let $c_2 \triangleq \sqrt{k} M_R M_\Phi + M_V$.

**Lemma E.13.** *For all $\bar{\theta}, \bar{\theta}' \in \mathbb{S}^d$,*

$$
|L'[\rho](\bar{\theta}) - L'[\rho](\bar{\theta}')| \leq c_2 \|\bar{\theta} - \bar{\theta}'\|_2 \tag{E.7}
$$

*Proof.* Using the definition of $L'$ and triangle inequality,

$$
\begin{aligned}
|L'[\rho](\bar{\theta}) - L'[\rho](\bar{\theta}')| &\leq \left\| \nabla R \left( \int \Phi d\rho \right) \right\|_2 \|\Phi(\bar{\theta}) - \Phi(\bar{\theta}')\|_2 + |V(\bar{\theta}) - V(\bar{\theta}')| \\
&\leq (\sqrt{k} M_R M_\Phi + M_V) \|\bar{\theta} - \bar{\theta}'\|_2 \qquad \text{(by definition of } M_\Phi, M_R, M_V)
\end{aligned}
$$

$\qquad\square$

Now the remainder of the proof will proceed as follows: we show that if $\rho_t$ is far from optimality, either the expected velocity of $\theta$ under $\rho_t$ will be large in which case the loss decreases from Corollary E.8, or there will exist $\bar{\theta}$ such that $L'[\rho_t](\bar{\theta}) \ll 0$. We will first show that in the latter case, the $\sigma U^d$ noise term will grow mass exponentially fast in a descent direction until we make progress. Define $K_t^{-\tau} \triangleq \{\bar{\theta} \in \mathbb{S}^d : L'[\rho_t](\bar{\theta}) \leq -\tau\}$, the $-\tau$-sublevel set of $L'[\rho_t]$, and let $m(S) \triangleq \mathbb{E}_{\theta \sim U^d}[\mathbf{1}(\theta \in S)]$ be the normalized spherical area of the set $S$.

**Lemma E.14.** *If $K_t^{-\tau}$ is nonempty, for $0 \leq \delta \leq \tau$, $\log m(K_t^{-\tau+\delta}) \geq -2d \log \frac{c_2}{\delta}$.*

*Proof.* Let $\bar{\theta} \in K_t^{-\tau}$. From Lemma E.13, $L'[\rho](\bar{\theta}') \leq -\tau + \delta$ for all $\bar{\theta}'$ with $\|\bar{\theta}' - \bar{\theta}\|_2 \leq \frac{\delta}{c_2}$. Thus, we have

$$
m(K_t^{-\tau+\delta}) \geq \mathbb{E}_{\bar{\theta}' \sim U^d} \left[ \mathbf{1}[\|\bar{\theta}' - \bar{\theta}\|_2 \leq \frac{\delta}{c_2}] \right]
$$

Now the statement follows by Lemma 2.3 of (Ball, 1997). $\qquad\square$

Now we show that if a descent direction exists, the added noise will find it and our function value will decrease. We start with a general lemma about the magnitude of the gradient of a 2-homogeneous function in the radial direction.

**Lemma E.15.** *Let $h : \mathbb{R}^{d+1} \to \mathbb{R}$ be a 2-homogeneous function. Then for any $\theta \in \mathbb{R}^{d+1}$, $\bar{\theta}^\top \nabla h(\theta) = 2\|\theta\|_2 h(\bar{\theta})$.*

*Proof.* We have $h(\theta + \alpha\bar{\theta}) = (\|\theta\|_2 + \alpha)^2 h(\bar{\theta})$. Differentiating both sides with respect to $\alpha$ and evaluating the derivative at 0, we get $\theta^\top \nabla h(\theta) = 2\|\theta\|_2 h(\theta)$, as desired. $\square$

We state the lemma claiming that our objective will decrease if $L'[\rho_t](\bar{\theta}) \ll 0$ for some $\bar{\theta} \in \mathbb{S}^d$.

**Lemma E.16.** *Choose*

$$l \geq \frac{\log(W_\epsilon^2/\sigma) + 2d\log\frac{2c_2}{\tau}}{\tau - \sigma} + 1$$

*If $K_{t^*}^{-\tau}$ is nonempty for some $t^*$ satisfying $t^* + l \leq t_\epsilon$, then after $l$ steps, we will have*

$$L[\rho_{t^*+l}] \leq L[\rho_{t^*}] - \frac{(\tau/4 - \sigma l c_1(W_\epsilon^2 + 1))^2}{l c_1^2 W_\epsilon^2} + \sigma l c_1(W_\epsilon^2 + 1) \tag{E.8}$$

We will first show that a descent direction in $L'[\rho_t]$ will remain for the next $l$ time steps. In the notation of Lemma E.12, define $z(s) \triangleq C_R B_\Phi \int_{t^*}^{t^*+s} \|Q'(t)\|_1 dt$. Note that from Lemma E.12, for all $\bar{\theta} \in \mathbb{S}^d$ we have $|L'[\rho_{t^*+s}](\bar{\theta}) - L'[\rho_{t^*}](\bar{\theta})| \leq z(s)$. Thus, the following holds:

**Claim E.17.** *For all $s \leq l$, $K_{t^*+s}^{-\tau+z(s)}$ is nonempty.*

*Proof.* By assumption, $\exists \bar{\theta}$ with $\bar{\theta} \in K_{t^*}^{-\tau}$. Then $L'[\rho_{t^*+s}](\bar{\theta}) \leq L'[\rho_{t^*}](\bar{\theta}) + z(s) \leq -\tau + z(s)$, so $K_{t^*+s}^{-\tau+z(s)}$ is nonempty. $\square$

Let $T_s \triangleq K_{t^*+s}^{-\tau/2+z(s)}$ for $0 \leq s \leq l$. We now argue that this set $T_s$ does not shrink as $t$ increases.

**Claim E.18.** *For all $s' > s$, $T_{s'} \supseteq T_s$.*

*Proof.* From equation E.6 and the definition of $z(s)$, $|L'[\rho_{t+s'}](\bar{\theta}) - L'[\rho_{t+s}](\bar{\theta})| \leq z(s') - z(s)$. It follows that for $\bar{\theta} \in T_s$

$$\begin{aligned} L'[\rho_{t+s'}](\bar{\theta}) &\leq L'[\rho_{t+s}](\bar{\theta}) + z(s') - z(s) \\ &\leq -\tau/2 + z(s) - z(s) + z(s') \qquad \text{(by definition of } T_s\text{)} \\ &\leq -\tau/2 + z(s') \end{aligned}$$

which means that $\bar{\theta} \in T_{s'}$. $\square$

Now we show that the weight of the particles in $T_s$ grows very fast if $z(k)$ is small.

**Claim E.19.** *Suppose that $z(l) \leq \tau/4$. Let $\tilde{T}_s = \{\theta \in \mathbb{R}^{d+1} : \bar{\theta} \in T_s\}$. Define $N(s) \triangleq \int_{\tilde{T}_s} \|\theta\|^2 d\rho_{t^*+s}$ and $\beta \triangleq \exp(-2d\log\frac{2c_2}{\tau})$. Then $N'(s) \geq (\tau - \sigma)N(s) + \sigma\beta$.*

*Proof.* From the assumption $z(l) \leq \frac{\tau}{4}$, it holds that $T_s \subseteq K_{t^*+s}^{-\tau/4} \forall s \leq k$. Since $T_s$ is defined as a sublevel set, $v[\rho_{t^*+s}](\bar{\theta})$ points inwards on the boundary of $T_s$ for all $\bar{\theta} \in T_s$, and by 1-homogeneity of the gradient, the same must hold for all $u \in \tilde{T}_s$.

Now consider any particle $\theta \in \tilde{T}_s$. We have that $\theta$ flows to $\theta + v[\rho_{t^*+s}](\theta)ds$ at time $t^* + s + ds$. Furthermore, since the gradient points inwards from the boundary, it also follows that $u + v[\rho_{t^*+s}](\theta)ds \in \tilde{T}_s$. Now we compute

$$\begin{aligned} \int_{\tilde{T}_s} \|\theta\|_2^2 d\rho_{t^*+s+ds} &= (1-\sigma ds)\int_{\tilde{T}_s} \|\theta + v[\rho_{t^*+s}](\theta)ds\|_2^2 d\rho_{t^*+s} + \sigma ds \int_{\tilde{T}_s} 1 dU^d \\ &\geq (1-\sigma ds)\int_{\tilde{T}_s} (\|\theta\|_2^2 + 2\theta^\top v[\rho_{t^*+s}](\theta)ds)d\rho_{t^*+s} + \sigma m(K_{t^*+s}^{-\tau/2+z(s)})ds \end{aligned}$$
$$\tag{E.9}$$

Now we apply Lemma E.15, using the 2-homogeneity of $F'$ and the fact that $L'[\rho_{t^*+s}](\bar{\theta}) \le -\tau/4 \; \forall \theta \in \tilde{T}_s$

$$\|\theta\|_2^2 + 2\theta^\top v[\rho_{t^*+s}](\theta)ds = \|\theta\|_2^2 - 4\|\theta\|_2^2 L'[\rho_{t^*+s}](\bar{\theta})ds$$
$$\ge \|\theta\|_2^2(1 + \tau ds) \tag{E.10}$$

Furthermore, since $K_{t^*+s}^{-\tau+z(s)}$ is nonempty by Claim E.17, we can apply Lemma E.14 and obtain

$$m(K_{t^*+s}^{-\tau/2+z(s)}) \ge \beta \tag{E.11}$$

Plugging equation E.10 and equation E.11 back into equation E.9, we get

$$\int_{\tilde{T}_s} \|u\|_2^2 d\rho_{t^*+s+ds} \ge (1 - \sigma ds)(1 + 2\tau ds)N(s) + \sigma\beta ds$$

Since we also have that $\tilde{T}_{s+ds} \supseteq \tilde{T}_s$, it follows that

$$N(s + ds) = \int_{\tilde{T}_{s+ds}} \|u\|_2^2 d\rho_{t^*+s+ds} \ge (1 - \sigma ds)(1 + \tau ds)N(s) + \sigma\beta ds$$

and so $N'(s) \ge (\tau - \sigma)N(s) + \sigma\beta$. $\qquad\square$

Now we are ready to prove Lemma E.16.

*Proof of Lemma E.16.* If $z(l) = C_R B_\Phi \int_t^{t+l} \|Q'(t)\|_1 \ge \frac{\tau}{4}$, then by rearranging the conclusion of Lemma E.12 we immediately get equation E.8.

Suppose for the sake of contradiction that $z(l) \le \tau/4$. From Claim E.19, it follows that $N(1) \ge \sigma\beta$, and $N(l) \ge \exp((\tau-\sigma)(l-1))N(1)$. Thus, in $\frac{\log(W_\epsilon^2/\sigma)+2d\log\frac{2c_2}{\tau}}{\tau-\sigma}+1$ time, $W_{t^*+l} \ge N(l) \ge W_\epsilon^2$, a contradiction. Therefore, it must be true that $z(l) \ge \tau/4$.

$\qquad\square$

The following lemma will be useful in showing that the objective will decrease fast when $\rho_t$ is very suboptimal.

**Lemma E.20.** *For any time $t$ with $0 \le t \le t_\epsilon$, we have*

$$\frac{d}{dt}L[\rho_t] \le \sigma B_L(W_\epsilon^2 + 1) - \frac{\mathbb{E}_{\theta\sim\rho_t}[L'[\rho_t](\theta)]^2}{W_\epsilon^2} \tag{E.12}$$

*Proof.* We can first compute

$$\mathbb{E}_{\theta\sim\rho_t}[L'[\rho_t](\theta)] = \mathbb{E}_{\theta\sim\rho_t}[L'[\rho_t](\bar{\theta})\|\theta\|_2^2]$$
$$= \frac{1}{2}\mathbb{E}_{\theta\sim\rho_t}[\|\theta\|_2\bar{\theta}^\top v[\rho_t](\theta)] \qquad \text{(via Lemma E.15)}$$
$$\le \frac{1}{2}\sqrt{\mathbb{E}_{\theta\sim\rho_t}[\|\theta\|_2^2]\mathbb{E}_{\theta\sim\rho_t}[\|v[\rho_t](\theta)\|_2^2]} \qquad \text{(by Cauchy-Schwarz)}$$
$$\le \frac{1}{2}W_\epsilon\sqrt{\mathbb{E}_{\theta\sim\rho_t}[\|v[\rho_t](\theta)\|_2^2]}$$

Rearranging gives $\mathbb{E}_{\theta\sim\rho_t}[\|v[\rho_t](\theta)\|_2^2] \ge \frac{\mathbb{E}_{\theta\sim\rho_t}[L'[\rho_t](\theta)]^2}{W_\epsilon^2}$, and plugging this into equation E.1 gives the desired result. $\qquad\square$

*Proof of Theorem E.4.* Let $L^\star$ denote the infimum $\inf_\rho L[\rho]$, and let $\rho^\star$ be an $\epsilon$-approximate global minimizer of $L$: $L[\rho^\star] \le L^\star + \epsilon$. (We define $\rho^\star$ because a true minimizer of $L$ might not exist.) Let $W^\star \triangleq \mathbb{E}_{\theta\sim\rho^\star}[\|\theta\|_2^2]$. We first note that since $b_V W^{\star 2} \le L[\rho^\star] \le L[\rho_0]$, $W^{\star 2} \le L[\rho_0]/b_V \le W_\epsilon^2$.

Now we bound the suboptimality of $\rho_t$: since $L$ is convex in $\rho$,

$$L[\rho^\star] \ge L[\rho_t] + \mathbb{E}_{\theta\sim\rho^\star}[L'[\rho_t](\theta)] - \mathbb{E}_{\theta\sim\rho_t}[L'[\rho_t](\theta)]$$

Rearranging gives

$$L[\rho_t] - L[\rho^\star] \leq \mathbb{E}_{\theta \sim \rho_t}[L'[\rho_t](\theta)] - \mathbb{E}_{\theta \sim \rho^\star}[L'[\rho_t](\theta)]$$

$$\leq \mathbb{E}_{\theta \sim \rho_t}[L'[\rho_t](\theta)] - W^{\star 2} \min \left\{ \min_{\bar{\theta} \in \mathbb{S}^{d-1}} L'[\rho_t](\bar{\theta}), 0 \right\} \tag{E.13}$$

Now let $l \triangleq \frac{W_\epsilon^2}{\epsilon - 2W_\epsilon^2 \sigma}\left(2 \log \frac{W_\epsilon^2}{\sigma} + 2d \log \frac{4W_\epsilon^2 c_2}{\epsilon}\right)$, which satisfies Lemma E.16 with the value of $\tau$ later specified. Suppose that there is a $t$ with $0 \leq t \leq t_\epsilon - 2l$ and $\forall t' \in [t, t + 2l]$, $L[\rho_{t'}] - L^\star \geq 2\epsilon$. Then $L[\rho_{t'}] - L[\rho^\star] \geq \epsilon$. We will argue that the objective decreases when we are $\epsilon$ suboptimal:

$$L[\rho_t] - L[\rho_{t+2l}] \geq \tag{E.14}$$

$$\min \left\{ \frac{(\epsilon/8W_\epsilon^2 - l\sigma c_1(W_\epsilon^2 + 1))^2}{c_1^2 W_\epsilon^2 l} - 3\sigma l c_1(W_\epsilon^2 + 1), l\frac{\epsilon^2}{4W_\epsilon^2} - 2\sigma l B_L(W_\epsilon^2 + 1) \right\} \tag{E.15}$$

Using equation E.13 and $W_\epsilon \geq W^\star$, we first note that

$$\epsilon \leq \mathbb{E}_{\theta \sim \rho_{t'}}[L'[\rho_{t'}](\theta)] - W_\epsilon^2 \min \left\{ \min_{\bar{\theta} \in \mathbb{S}^{d-1}} L'[\rho_{t'}](\bar{\theta}), 0 \right\} \forall t' \in [t, t + l]$$

Thus, either $\min_{\bar{\theta} \in \mathbb{S}^d} L'[\rho_{t'}](\bar{\theta}) \leq -\frac{\epsilon}{2W^{\star 2}} \leq -\frac{\epsilon}{2W_\epsilon^2}$, or $\mathbb{E}_{\theta \sim \rho_{t'}}[L'[\rho_{t'}](\theta)] \geq \frac{\epsilon}{2}$. If $\exists t' \in [t, t + l]$ such that the former holds, then the $\tau \triangleq \frac{\epsilon}{2W_\epsilon^2}$ sub-level set $K_{t'}^{-\tau}$ is non-empty. Applying Lemma E.16 gives

$$L[\rho_{t'}] - L[\rho_{t'+l}] \geq \frac{(\epsilon/8W_\epsilon^2 - l\sigma c_1(W_\epsilon^2 + 1))^2}{c_1^2 W_\epsilon^2 l} - \sigma l c_1(W_\epsilon^2 + 1)$$

Furthermore, from Lemma E.10, $L[\rho_{t+2l}] - L[\rho_{t'+l}] \leq \sigma l c_1(W_\epsilon^2 + 1)$ and $L[\rho_{t'}] - L[\rho_t] \leq \sigma l B_L(W_\epsilon^2 + 1)$, and so combining gives

$$L[\rho_t] - L[\rho_{t+2k}] \geq \frac{(\epsilon/8W_\epsilon^2 - l\sigma c_1(W_\epsilon^2 + 1))^2}{c_1^2 W_\epsilon^2 l} - 3\sigma l c_1(W_\epsilon^2 + 1) \tag{E.16}$$

In the second case $\mathbb{E}_{\theta \sim \rho_{t'}}[L'[\rho_{t'}](\theta)] \geq \frac{\epsilon}{2}$, $\forall t' \in [t, t + l]$. Therefore, we can integrate equation E.12 from $t$ to $t + l$ in order to get

$$L[\rho_t] - L[\rho_{t+l}] \geq l\frac{\epsilon^2}{4W_\epsilon^2} - \sigma l B_L(W_\epsilon^2 + 1)$$

Therefore, applying Lemma E.10 again gives

$$L[\rho_t] - L[\rho_{t+2l}] \geq l\frac{\epsilon^2}{4W_\epsilon^2} - 2\sigma l B_L(W_\epsilon^2 + 1) \tag{E.17}$$

Thus equation E.15 follows.

Now recall that we choose

$$\sigma \triangleq \exp(-d \log(1/\epsilon) \text{poly}(k, M_V, M_R, M_\Phi, b_V, B_V, C_R, B_\Phi, L[\rho_0] - L[\rho^\star]))$$

For the simplicity, in the remaining computation, we will use $O(\cdot)$ notation to hide polynomials in the problem parameters besides $d, \epsilon$. We simply write $\sigma = \exp(-c_3 d \log(1/\epsilon))$. Recall our choice $t_\epsilon \triangleq O(\frac{d^2}{\epsilon^4} \log^2(1/\epsilon))$. It suffices to show that our objective would have sufficiently decreased in $t_\epsilon$ steps. We first note that with $c_3$ sufficiently large, $W_\epsilon^2 = O(L[\rho_0]/b_v) = O(1)$. Simplifying our expression for $l$, we get that $l = O(\frac{d}{\epsilon} \log \frac{1}{\epsilon})$, so long as $\sigma W_\epsilon = o(\epsilon)$, which holds for sufficiently large $c_3$. Now let

$$\delta_1 \triangleq \frac{(\epsilon/8W_\epsilon^2 - l\sigma c_1(W_\epsilon^2 + 1))^2}{c_1^2 W_\epsilon^2 l} - 3\sigma l c_1(W_\epsilon^2 + 1)$$

$$\delta_2 \triangleq l\frac{\epsilon^2}{4W_\epsilon^2} - 2\sigma l B_L(W_\epsilon^2 + 1)$$

Again, for sufficiently large $c_3$, the terms with $\sigma$ become negligible, and $\delta_1 = O(\frac{\epsilon^2}{l}) = O(\frac{\epsilon^3}{d \log(1/\epsilon)})$. Likewise, $\delta_2 = O(d\epsilon \log(1/\epsilon))$.

Thus, if by time $t$ we have not encountered $2\epsilon$-optimal $\rho_t$, then we will decrease the objective by $O(\frac{\epsilon^3}{d \log(1/\epsilon)})$ in $O(\frac{d}{\epsilon} \log \frac{1}{\epsilon})$ time. Therefore, a total of $O(\frac{d^2}{\epsilon^4} \log^2(1/\epsilon))$ time is sufficient to obtain $\epsilon$ accuracy. $\square$

### E.3 DISCRETE-TIME OPTIMIZATION

To circumvent the technical issue of existence of a solution to the continuous-time dynamics, we also note that polynomial time convergence holds for discrete-time updates.

**Theorem E.21.** *Along with Assumptions E.1, E.2, E.3 additionally assume that $\nabla \Phi_i$ and $\nabla V$ are $C_\Phi$ and $C_V$-Lipschitz, respectively. Let $\rho_t$ evolve according to the following discrete-time update:*

$$\rho_{t+1} \triangleq \rho_t + \eta(-\sigma\rho_t + \sigma U^d - \nabla \cdot (v[\rho_t]\rho_t))$$

*There exists a choice of*

$$\sigma \triangleq \exp(-d\log(1/\epsilon)\mathrm{poly}(k, M_V, M_R, b_V, B_V, C_R, B_\Phi, C_\Phi, C_V, L[\rho_0] - L[\rho^\star]))$$

$$\eta \triangleq \mathrm{poly}(k, M_V, M_R, b_V, B_V, C_R, B_\Phi, C_\Phi, C_V, L[\rho_0] - L[\rho^\star])$$

$$t_\epsilon \triangleq \frac{d^2}{\epsilon^4}\mathrm{poly}(k, M_V, M_R, b_V, B_V, C_R, B_\Phi, C_\Phi, C_V, L[\rho_0] - L[\rho^\star])$$

*such that $\min_{0 \leq t \leq t_\epsilon} L[\rho_t] - L^\star \leq \epsilon$.*

The proof follows from a standard conversion of the continuous-time proof of Theorem E.4 to discrete time, and we omit it here for simplicity.

## F ADDITIONAL MATERIAL ON EXPERIMENTS

### F.1 DETAILED SETUP FOR FIGURE 2 EXPERIMENT

Our ground truth comes from a random neural network with 6 hidden units, and during training we use a network with as many hidden units as examples. For classification, we used rejection sampling to obtain datapoints with unnormalized margin of at least 0.1 on the ground truth network. We use a fixed dimension of $d = 20$. For all experiments, we train the network for 20000 steps with $\lambda = 10^{-8}$ and average over 100 trials for each plot point.

### F.2 DETAILED SETUP FOR FIGURE 3 EXPERIMENT

The left side of Figure 3 shows the experimental results for synthetic data generated from a ground truth network with 10 hidden units, input dimension $d = 20$, and a ground truth unnormalized margin of at least 0.01. We train for 80000 steps with learning rate 0.1 and $\lambda = 10^{-5}$, using two-layer networks with $2^i$ hidden units for $i$ ranging from 4 to 10. We perform 20 trials per hidden layer size and plot the average over trials where the training error hit 0. (At a hidden layer size of $2^7$ or greater, all trials fit the training data perfectly.) The right side of Figure 3 demonstrates the same experiment, but performed on MNIST with hidden layer sizes of $2^i$ for $i$ ranging from 6 to 15. We train for 600 epochs using a learning rate of 0.01 and $\lambda = 10^{-6}$ and use a single trial per plot point. For MNIST, all trials fit the training data perfectly. The MNIST experiments are more noisy because we run one trial per plot point for MNIST, but the same trend of decreasing test error and increasing margin still holds.

### F.3 VERIFYING CONVERGENCE TO THE MAX-MARGIN

We verify the normalized margin convergence on a two-layer networks with one-dimensional input. A single hidden unit computes the following: $x \mapsto a_j\mathrm{relu}(w_jx + b_j)$. We add $\|\cdot\|_2^2$-regularization to $a, w,$ and $b$ and compare the resulting normalized margin to that of an approximate solution of the $\ell_1$ SVM problem with features $\mathrm{relu}(wx_i + b)$ for $w^2 + b^2 = 1$. Writing this feature vector is intractable, so we solve an approximate version by choosing 1000 evenly spaced values of $(w, b)$. Our theory predicts that with decreasing regularization, the margin of the neural network converges to the $\ell_1$ SVM objective. In Figure 4, we plot this margin convergence and visualize the final networks and ground truth labels. The network margin approaches the ideal one as $\lambda \to 0$, and the visualization shows that the network and $\ell_1$ SVM functions are extremely similar.

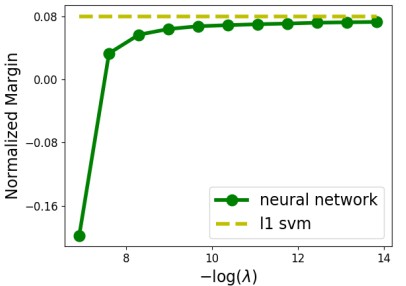 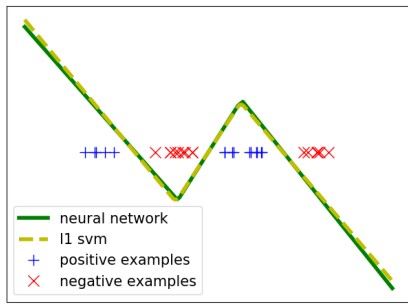

Figure 4: Neural network with input dimension 1. **Left:** Normalized margin as we decrease $\lambda$. **Right:** Visualization of the normalized functions computed by the neural network and $\ell_1$ SVM solution for $\lambda \approx 10^{-14}$.

| Method | CIFAR10 | CIFAR100 |
|---|---|---|
| Weight decay annealing | 5.86 | 26.22 |
| Fixed weight decay | 6.01 | 27.00 |

Table 1: Test error on CIFAR10 and CIFAR100 for initial $\lambda = 0.0005$.

## F.4 EXPERIMENTS ON CIFAR10 AND CIFAR100

We train a modified WideResNet architecture (Zagoruyko & Komodakis, 2016) on CIFAR10 and CIFAR100. Our theory does not entirely apply because the identity mapping prevents ResNet architectures from being homogeneous, but our experiments show that reducing weight decay can still help generalization error in this setting. Because batchnorm can cause the regularizer to have different effects (van Laarhoven, 2017), we remove batchnorm layers and train a 16 layer deep WideResNet. We again compare a network trained with weight decayed annealing to one trained without annealing. We used a fixed learning rate schedule that starts at 0.1 and decreases by a factor of 0.2 at epochs 60, 120, and 160. For CIFAR10, we use an initial weight decay of $0.0002$ and decrease the weight decay by 0.2 at epoch 60, and then by 0.5 at epochs 90, 120, 140, 160. For CIFAR100, we initialize weight decay at $0.0005$ and decrease it by $0.2$ at epochs 60, 120, and 160. We tried different parameters for the initial weight decay and chose the ones that worked best for the model without annealing. We also tried using small weight decays at initialization, but these models failed to generalize well – we believe this is due to an optimization issue where the algorithm fails to find a true global minimum of the regularized loss. We believe that annealing the weight decay directs the optimization algorithm closer towards the global minima for small $\lambda$.

Table 1 shows the test error achieved by models with and without annealing. We see that the simple change of annealing weight decay can decrease the test error for this architecture.

