# OpenReview forum: "On the Margin Theory of Feedforward Neural Networks"
_ICLR.cc/2019/Conference_

### Official Review · AnonReviewer1 · 2018-11-02
**Interesting insights on inductive bias of two-layer ReLU networks**

**Rating:** 7
**Confidence:** 3

**Review:**

This paper studies the implicit bias of minimizers of a regularized cross entropy loss of a two-layer network with ReLU activations. By combining several results, the authors obtain a generalization upper bound which does not increase with the network size. Furthermore, they show that the maximum normalized margin is, up to a scaling factor, the l1 svm margin over the lifted feature space of an infinite-size network. Finally, in a setting of infinite-sized networks, it is proved that perturbed Wasserstein gradient flow finds a global minimum in polynomial time.

I think that the results are interesting and relevant to current efforts of understanding neural networks. The techniques and ideas seem promising and may be applied in more general settings. The paper is mostly clearly written, but there are some issues which I outline below.
1.	It is not clear what is the novelty in sections 2 and 3.1 except the combination of all the results to get a generalization bound which does not increase with network size (which on its own is non-trivial). Specifically,
a.	What is the technical contribution in Theorem 2.1 beyond the results of the two papers of Rosset et al. (journal paper and the NIPS paper which was mentioned in the comment on missing prior work)?
b.	How does Theorem 3.1 compare with previous Rademacher bounds for neural networks which are based on the margin? In Neyshabur et al. (2018), it is shown that margin-based generalization bounds empirically increase with network size. Does this hold for the bound in Theorem 3.1?

2.	In the work of Soudry et al. (2018) section 4.3, they consider deep networks with an unregularized loss and show that gradient descent converges to an l2 max margin solution under various assumptions. What is the connection between this result and the l1 max margin result in section 3.3?

3.	What are the main proof ideas of Theorem 4.3? Why is the perturbation needed?

4.	What is the size of the network that was trained in Section 5 in the experiments of Figure 3? Only the size of the ground truth network is mentioned.


---------Revision------------

I have read the author's response and other reviews. I am not changing the current review.
I have one technical question. In the new generalization bound (Proposition 3.1), the authors claim that the product of Frobenius norms is replaced with a sum. However, I don't see any sum in the proof. Could the authors please clarify this?

---

> ### Author Response · Authors · 2018-11-15
> **response**
>
> We thank the reviewer for the positive feedback. Below we respond to the points raised by the reviewer:
>
> --- “It is not clear what is the novelty in sections 2 and 3.1”
>
> The main contribution in these sections lies in our framework, which disentangles optimization and statistics for analyzing generalization. In prior work, optimization and statistics are necessarily entangled because the implicit regularization of the training algorithm is analyzed. By looking at the global optimizer of the regularized loss, we can avoid this entanglement. As pointed out by the reviewer, this allows us to cleanly obtain generalization bounds that improve with over-parameterization.
>
> --- “What is the technical contribution in Theorem 2.1 beyond the results of the two papers of Rosset et al.”
>
> We believe that the conceptual contribution of the statement of Theorem 2.1 is important, though the proof builds upon Rosset et al’04. It has been not well-understood that existing training algorithms can converge to the max normalized margin solution for deep models. E.g., It has been pointed out in Bartlett et al.’17 that “what is missing is an analysis verifying that SGD applied to standard neural networks returns large margin predictor.”
>
> Our result shows that the max margin solution can be obtained with a weak regularizer, assuming the optimization can succeed. This is particularly relevant for deep learning because cross-entropy + weak regularizer is the method used in practice.
>
> --- “How does Theorem 3.1 compare with previous Rademacher bounds for neural networks which are based on the margin?”
>
> Theorem 3.1 is a direct consequence of the bounds of Neyshabur et. al (2015b) and Golowich et. al (2017). We should have clarified that these generalization error bounds are not considered to be the contribution of the paper, and our revision makes this explicit. Instead, the contribution in our Section 3 comes from 1) the application of our framework to analyze generalization independently from optimization 2) our comparison between neural networks and kernel methods.
>
> --- “it is shown that margin-based generalization bounds empirically increase with network size. Does this hold for the bound in Theorem 3.1?”
>
> Empirically, the generalization error bound of the old Theorem 3.1 (new Proposition 3.1) does decrease with the size of the network, as does the test error. Our revision includes this experiment. We found that the anti-correlation of margin and test error is only apparent when we train the regularized objective for long enough time until convergence, which may be why it’s somewhat surprising.
>
> --- “What is the connection between this result and the l1 max margin result in section 3.3?”
>
> The specific distinction between the two results is as follows: our max-margin results are jointly over optimization of all network parameters, whereas the results in Section 4.2 of https://arxiv.org/abs/1710.10345 (the journal version of Soudry et. al (2018)) pertain to the max-margin problem over a single layer of the network, i.e. the other layers are fixed and only a single layer is optimized. Further, it requires that the activation patterns remain unchanged throughout the dynamics of gradient descent. For two-layer networks, the joint max-margin problem over both layers of the network results in the l1 SVM.
>
> --- “What are the main proof ideas of Theorem 4.3? Why is the perturbation needed?”
>
> The main proof idea is that as long as \rho has mass on some descent direction, the two-homogeneity will result in a decrease in the objective value. (See Lemma E.16). The noise ensures that there will be enough mass in the descent direction to start with.
>
> --- “What is the size of the network that was trained in Section 5 in the experiments of Figure 3?”
>
> To ensure that optimization is not an issue, we use a network size equal to the size of the training set, which varies from 60 to 600 in increments of 60.
>
> To conclude, our contributions are the following: first, our Theorem 2.1 sets up several conceptual contributions in Sections 2 and 3 of our revised paper. We apply Theorem 2.1 to cleanly show 1) optimizing weakly-regularized logistic loss has an implicit bias towards solutions with a maximum possible margin, and therefore best possible generalization error bound (Corollary 3.2 in our revision) 2) this generalization error upper bound is decreasing as the size of the architecture grows. On the technical side, we show that we can still obtain an approximate max-margin using an approximate global minimizer. Next, we show that perturbed gradient descent on an infinite-size network finds global minimizers in polynomial time. Finally, we construct distributions where a neural network enjoys better generalization guarantees than kernel methods.

---

> ### Author Response · Authors · 2018-12-05
> **response to question in revision**
>
> Here is our response to the question in the revision of the review:
>
> --- “I have one technical question. In the new generalization bound (Proposition 3.1), the authors claim that the product of Frobenius norms is replaced with a sum. However, I don't see any sum in the proof. Could the authors please clarify this?”
>
> The sum is implicit in the normalization. To compute the normalized margin, we compute the margin of \Theta/\|Theta\|_F. Note that the sum is obtained by expanding \|\Theta\|_F^2 into \sum_{k = 1}^K \|W_k\|_F^2.

---

### Official Review · AnonReviewer3 · 2018-11-05
**Review of 'On the Margin Theory of Feedforward Neural Networks'**

**Rating:** 6
**Confidence:** 4

**Review:**

UPDATE: after revisions and discussion. There seems to be some interesting results presented in this paper which I think would be good to have discussed at the conference. This is conditional on further revisions of the work by the authors.


This paper studies margin theory for neural nets.

1. First it is shown that margin of the solution to regularized problem approaches max margin solution.
2. Then a bound is given for using approximate solution to above optimization problem instead of exact one. Note that the bound depends on size of the network via parameter a.
3. Then 2-layer relu networks are studied. It shown that max margin is monotonically increasing in size of the network. Note however, it is hard to relate this results to inexact solutions since the bound in that case as was pointed out also depends on the size of the network.
4. Paper also provides comparison with kernel methods, simulations and shows that perturbed wasserstein flows find global optimiziers in poly time.

The paper argues that over-parameterization is good for generalization since margin grows with the number of parameters. However, it should be also noted the radius of data may also grow (and in case of the bounds it seems to be the radius of data in lifted space which increases with the size of the network). I hope authors can clarify this and points 2 and 3 above in their response. In the current form the paper is below the acceptance threshold for me.

---

> ### Author Response · Authors · 2018-11-15
> **response**
>
> We thank the reviewer for the thoughtful reviews. We address the reviewer’s points below:
>
> --- “it should be also noted the radius of data may also grow”
>
> We disagree with the reviewer, and believe that there is a misunderstanding here.  On the contrary, as we increase the size of the network, the radius of the data will not change. (The bound depends on the radius of the raw data, which doesn’t change. It does not depend on the radius of the data in the lifted space which may grow.)
>
> In summary, the bounds depend on the normalized margin of the network and the norm of the raw data. The latter does not change, and the maximum normalized margin increases as the width of the network grows. Furthermore, the degree to which we need to approximate the optimal loss in order to obtain an approximate max-margin also will not change as the width of the network grows. This can explain why over-parameterized models can generalize better.
>
> ---  “the bound depends on size of the network via parameter a”
>
> We consider the parameter a to be fixed as the network size grows. For a multi-layer ReLU network, a would be the depth of the network, so as we increase the width of the hidden layers this quantity will not change. This also addresses the concern in point 3.
>
> In summary, the contributions of our paper are the following: on the conceptual side, our Theorem 2.1 provides a framework for disentangling optimization and statistics when analyzing generalization. We apply this framework to show that 1) the widely-used algorithm of optimizing weakly-regularized logistic loss has an implicit bias towards solutions with a good generalization error upper bound (see Corollary 3.2 in our revision) 2) this implicit bias explains why over-parametrization improves generalization in practice (see the old Theorem 3.2, or Theorem 3.3 in the revision). On the technical side, our Theorem 2.2 first relaxes the requirement of a strict global minimizer and still allows us to obtain an approximate max-margin. We also show that we can find a global minimizer in polynomial time via perturbed gradient descent on an infinite-size neural network. Finally, we also construct distributions where a neural network enjoys better generalization guarantees than kernel methods, which shows the benefit of depth for generalization.

---

> > ### Comment · AnonReviewer3 · 2018-12-04
> > **thanks for clarifications**
> >
> > Thank you for clarifications -- I have better understanding of the results now.
> >
> > After looking through proofs more carefully, it seems that there are a number of other technical issues, which perhaps you could address.
> >
> > 1. For proof of theorem 3.1 and theorem c.1, theorem 2 of Kakade et al is cited. Note that I have not found theorem 2 in that work (https://www.cs.cornell.edu/~sridharan/rad-paper.pdf). In either case, in the bound that you cite, empirircal loss should not be a margin loss, but should be a lipschitz upper bound on that.
> >
> > 2. Also, it seems that Cor 3.2 has an issue. One needs to have Thm 3.1. to hold uniformly over lambdas to take limsup. Detailed proof of Cor 3.2 is missing.
> >
> > Some additional comments:
> >
> > 1. Assumption of separability is rather strong.
> > 2. Result in sec 2.2 requires to approximate best loss to within a constant factor. We do not have algorithms that guarantee that. So this also appers to be strong.
> > 3. After cor 3.2. it is stated that "One consequence of Corollary 3.2 is that optimizing weakly-regularized logistic loss results in the
> > best possible generalization bound out of all models with the given architecture." I did not understand how one has arrived at this conclusion.

---

> > > ### Author Response · Authors · 2018-12-05
> > > **response to comments**
> > >
> > > We thank the reviewer for the comments and the revision of the review. We believe that the responses below address the concerns about correctness in the proofs and we ask that the reviewer reconsider the score given to our paper.
> > >
> > > --- “For proof of theorem 3.1 and theorem c.1, theorem 2 of Kakade et al is cited. Note that I have not found theorem 2 in that work (https://www.cs.cornell.edu/~sridharan/rad-paper.pdf)”
> > >
> > > Here is the link to the paper at NIPS website: http://papers.nips.cc/paper/3510-on-the-complexity-of-linear-prediction-risk-bounds-margin-bounds-and-regularization.pdf. (The link provided by the reviewer seems to be a different version.) Note that the bound that we write in Theorem C.1 matches the bound in Theorem 2 of Kakade et al. We also note that this step is fairly standard in generalization theory and we only include it for formality.
> > >
> > > --- “Also, it seems that Cor 3.2 has an issue. One needs to have Thm 3.1. to hold uniformly over lambdas to take limsup.”
> > >
> > > We strongly disagree with the reviewer and believe that there is a misunderstanding about uniform convergence.  Proposition 3.1 is a uniform convergence result (as is standard for proving generalization bounds) and therefore holds uniformly over all possible parameters theta that separate the data, including \Theta_{\lambda, m} for every \lambda, and m. (Note that Proposition 3.1 doesn’t even involve lambda.) Therefore, this allows us to take the limsup.
> > >
> > > --- “Assumption of separability is rather strong.”
> > >
> > > We disagree and would argue that this is a very mild assumption when the networks are over-parameterized. As long as the network has width more than n, then generically the data can be separable. Moreover, the networks used in practice can often perfectly classify the training data. In fact, one strength of this paper is that it requires very few assumptions on the data.
> > >
> > > --- “Result in sec 2.2 requires to approximate best loss to within a constant factor. We do not have algorithms that guarantee that.”
> > >
> > > Our Section 4 provides an polynomial time algorithm that guarantees this for an infinite-size neural net. A finite-size neural net guarantee is likely to be impossible because of the intractability of the optimization problems.
> > >
> > > --- “it is stated that "One consequence of Corollary 3.2 is that optimizing weakly-regularized logistic loss results in the best possible generalization bound out of all models with the given architecture." I did not understand how one has arrived at this conclusion.”
> > >
> > > As the bound in Corollary 3.2 depends on the inverse of \gamma^{\star,M}, and \gamma^{\star, M} is the largest possible value of the margin, Corollary 3.2 achieves the best possible version of the generalization bound in Proposition 3.1.
> > >
> > > --- Regarding the reviewer’s original comments about the radius of the data: we would like to check whether this has been clarified by our response to the original review. To reiterate, the generalization bounds depend on the normalized margin of the network and the norm of the raw data. The latter does not change as the size of the network grows.

---

### Official Review · AnonReviewer2 · 2018-11-08
**Theorem 2.1 -2.2 are interesting**

**Rating:** 5
**Confidence:** 4

**Review:**

Overall I found that the paper does not clearly compare the results to existing work. There are some new results, but some of the results stated as theorems are immediate consequence of existing work and a more detailed discussion and comparison is warranted. I will first give detailed comments on the establishing the relationship to existing work and then summarize my evaluation.

————
Detailed comments on contributions and relationships to existing work.

A. Theorem 2.1 establishes the limit of the regularized solutions as the maximum margin separator.
This result is a generalization the analogous results for linear models Theorem 3 in Rosset et al. (2004) “Boosting as a regularized path to maximum margin separator” and Thm 2.1 in Rosset Zhu Hastie “margin maximizing loss functions”  (the later paper missing from references, and that paper generalizes the earlier result for multi-class cross entropy loss).
Main difference from earlier work:
1. extends the results for linear models to any homogeneous function
2. (minor) the previous results by Rosset et al. were stated only for lp norms, but this is a minor generalization since the earlier work didn’t at any point use the lp-ness of the norm and immediately extends for any norms.

Secondly, Theorem 2.2 also gives a bound on deviation of margin when the regularization is not driven all the way to 0. I do think this theorem would be differently stated by making the explicitly showing dependence of suboptimal margin \gamma’ on lambda and the sub optimality constant of loss. This way, one can derive 2.1 as a special case and also reason about what level of sub-optimality of loss can be tolerated.

B. Theorem 3.1 derives generalization bounds of learned parameters in terms of l2 margin.
—this and many similar results connecting generalization to margins have already been studied in the literature (Neyshabur et al. 2015b for example covers a larger family of norms than just l2 norm). Specially an analogous bound for l1 margin can also be found in these work which can be used in the discussions that follow.

C. Theorem 3.2: This result to my knowledge is new, but also pretty immediate from definition of margin. The proof essentially follows by showing that having more hidden units can only increase the margin since the margin is maximized over a larger set of parameters.

D. Comparison to kernel machines: Theorem 3.3 seems to be the paraphrasing of corollary 1 in Neyshabur et al (2014). But the authors claim that the Theorem 3.3 also holds when “the regularizer is small”. I do not understand what the authors are referring to here or how the result is different form existing work. Please clarify

-----------
In summary, The 2.1-2.2 on extension of the connection between regularized solution and maximum margin solution to general homogeneous models and to non-asymptotic regimes
-- this is in my opinion key contribution of the paper and an important result. But there is not much new technique in terms of proof here

---

> ### Author Response · Authors · 2018-11-15
> **response**
>
> We thank the reviewer for the comments. Our revision addresses many of the reviewer’s concerns regarding citations of prior work in our old Section 3. We have restructured our old Sections 3.1-3.3 into a new section which more explicitly discusses relationships with prior work.
>
> In response to the reviewer’s discussion regarding the contributions of our paper, we argue that our paper makes the following key conceptual and technical contributions.
>
> Conceptual:
>
> 1. The framework exhibited in Sections 2 and 3 allows us to disentangle optimization from statistical analysis. This is important because prior work on “implicit bias” requires restrictive assumptions about convergence of the iterates of the training algorithm (See for example Theorem 1 of Gunesekar et. al 2018b, which assumes that the loss goes to zero and the difference between the iterates of GD converges.) By considering global minimizers of the weakly-regularized loss, we can analyze generalization directly while avoiding these assumptions. Our analysis also applies to a much broader class of networks than analyzed by any prior work in this area.
>
> 2. In our revised Section 3, we apply this framework to study generalization. We show that for general depth-K networks, simple l2 weight decay + logistic loss optimizes a generalization bound that depends on the network only through the normalized margin and depth. This insight holds for a very broad class of neural nets.
>
> 3. In our new Theorem 3.3, we observe that the normalized margin is non-decreasing as we increase the width of the architecture. This explains why over-parameterization has been observed to help generalization performance in practice.
>
> Technical:
> 1. Our Theorem 2.2 shows that approximating the optimal loss within a constant factor is sufficient for obtaining a constant factor approximation of the maximum margin.
>
> 2. Our old Theorem 3.5 (new Theorem 4.3) constructs a distribution where neural nets will generalize well, but the generalization guarantees of the kernel method will be poor. This suggests that depth can be beneficial for generalization.
>
> 3. Our old Theorem 4.3 (new Theorem 5.3) shows that perturbed gradient descent can find a global minimizer for infinite-size neural networks in polynomial time. Prior work (Chizat and Bach (2018)) did not specify a convergence time and assumed convergence to some solution. Our result helps to fill in the picture for our Theorem 2.1, which assumes that we obtain a global minimizer. Our analysis for this result is technically involved.
>
> --- “this theorem would be differently stated … explicitly showing dependence of suboptimal margin \gamma’ on lambda and the sub optimality constant of loss.”
>
> We believe our Theorem 2.1 stated as it is now cleanly expresses the main contribution of Section 2 and puts our work in context with prior works (which show algorithmic bias towards some “minimum norm” solution). However, we have revised Theorem 2.2 to allow for general sub-optimality constants of the loss.
>
> --- “Theorem 3.1 derives generalization bounds … many similar results connecting generalization to margins have already been studied in the literature”
>
> We agree with this statement and have replaced Theorem 3.1 with Proposition 3.1, which states a generalization bound for arbitrary depth networks. It is a straightforward consequence of the bounds of Golowich et. al (2017), which depend on the product of Frobenius norms of the weight matrices. We observe that this product can be replaced by a sum, resulting in parameter free bounds in terms of only the inverse normalized margin. This is important conceptually as our Corollary 3.2 then shows that simple l2 weight decay + logistic loss optimizes the normalized margin and therefore generalization bound.
>
> --- “This result to my knowledge is new, but also pretty immediate from definition of margin.”
>
> We would argue that it is an important contribution of our framework in Sections 2 and 3 that we can produce clean proofs for such results.
>
> --- “The 2.1-2.2 … there is not much new technique in terms of proof here.”
>
> We’d like to reiterate that the conceptual contributions of Theorem 2.1 are valuable: it gives us a framework for analyzing generalization separately from optimization. In our new Section 3, we leverage this framework to cleanly show that 1) by optimizing weakly-regularized cross entropy loss, we are also optimizing the generalization bound in terms of normalized margin and 2) the previous point explains why increasing the width of the network can improve generalization. Both points give valuable insights into the generalization properties of neural nets trained in practice.
>
> On the topic of novel technical contributions, our Theorem 4.3 (new Theorem 5.3) shows that noisy gradient descent can find the global minimizer of the regularized loss for infinite-size networks in polynomial time, and the proof is fairly technically involved.

---

### Official Review · AnonReviewer4 · 2018-11-13
**Review for "On the Margin Theory of Feedforward Neural Networks"**

**Rating:** 5
**Confidence:** 4

**Review:**

The authors claim to prove three things: (1) Under logistic loss (with a vanishing regularization), the normalized margin (of the solution) converges to the max normalized margin, for positive homogenous functions. This is an asymptotic result: the amount of regularization vanishes. (2) For one hidden layer NN, the max margin under l_2 norm constraint on weights in the limit, is equivalent to the l_1 constraint (total variation) on the sign measure (specified by infinite neurons) for the one hidden layer NN. (3) Show some convergence rate for the mean-field view of one hidden layer NN, i.e., the Wasserstein gradient flow on the measure (of the neurons). The author show some positive result for a perturbed version.

The problem is certainly interesting. However, my main concerns are: (1) the novelty of the main theorems given the literature, and (2) the carefulness of stating what is known in the literature review.

In summary:
1. Theorem 2.1, Theorem 3.1, and Theorem 3.3 are anticipated, or not as critical, given the literature (detailed reasons in major comments).

2. The construction in Theorem 3.5 is nice, but, it is only able to say an upper bound of the generalization of kernel is not good (comparing upper bounds is not enough). In addition, For Theorem 4.3. [Mei Montanari and Nguyen 2018] also considers similar perturbed Wasserstein gradient flow, with many convergence results. One needs to be more careful in stating what is new.


Major comments:
1. Theorem 3.3 (and Theorem 3.2) seems to be the most interesting/innovative one.
However, I would like to argue that it might be natural in one line proof, with the following alternative view:

--
l_2 norm constraint normalized margin, one hidden layer NN, with infinite neurons

gamma^star, infty :=
\max \min_i y_i int_{neuron} w || u || ReLU( x_i \bar{u}) dS^{d-1} -- integral over normalized neurons over sphere

under the constraint
int_{neuron} (w^2 + ||u||^2) dS^{d-1} \leq 1

This is equivalent to the l_1 constraint margin (variation norm), one hidden layer NN,
gamma_l_1 :=
\max \min_i y_i int_{neuron} rho(u) ReLU( x_i \bar{u}) dS^{d-1} -- integral over normalized neurons over sphere

under the constraint
int_{neuron} |rho(u)| dS^{d-1} \leq 1/2

here rho(u) is the sign measure represented by neurons. Simply because at the optimum
w || u || = 1/2 ( w^2 + || u ||^2) := rho(u)
therefore
gamma^star, infty =  gamma_l_1

So one see the factor 1/2 exactly.
--

In addition, [Bach 18, JMLR:v18:14-546] discuss more in depth the l_1 type constraint
(TV of sign measure) rather then l_2 type constraint (RKHS) for one hidden layer NN with infinite neurons. The authors should cite this work.

It is clear that l_1(neuron) < l_2(neuron) therefore
l_2 constraint margin is always smaller than l_1 constraint margin.

2. Theorem 2.1. I think the proof is almost a standard exercise given [Rosset, Zhu, and Hastie 04].
The observation for it generalizes to positive homogenous function beyond linear is a nice addition, but not crucial enough to stand out as an innovation.

Much of the difficulty in related paper lies in achieving non asymptotic convergence rate to max margin solution, for logistic loss [Soudry, Hoffer and Srebro 18], or what happens when data is not perfectly separable [Ji and Telgarsky 18].

3. Generalization result Theorem 3.1. Maybe it is better to state as a corollary, given the known results in the literature, in my opinion. This generalization is standard result from margin-based bounds available
[Koltchinskii and Panchenko 02, Bartlett and Mendelson 02].

In addition, the authors remark that the limit for (3.3) may not exist. You can change to limsup, your footnote[4]
is essentially the limsup definition.

4. Theorem 3.5. This construction of the data distribution is the part I like. However, you should remind the reader that
having a small margin for the kernel only implies the the upper bound for generalization is bad.
Comparing the upper bound doesn't mean kernel method is performing bad for the instance.

From a logic view, it is unclear the benefit of Theorem 3.5.
I do agree one can try to see in simulation if kernel/RKHS approach (l_2) is performing worse for generalization, for one hidden layer NN. But this is separate from the theory.

5. Theorem 4.3. This result should be put in the context of the literature. Specifically
[Mei Montanari and Nguyen 2018], Eqn 11-12. The perturbed wasserstein flow the authors considered
looks very close to [Mei Montanari and Nguyen 2018], Eqn 11-12, admittedly with the logistic loss instead of the square loss.

Right now, as stated in the current paper, it is very hard for the general audience to understand the contribution. A better job in comparing the literature will help.

For the technical crowd, maybe emphasize on why the "simga" can help you achieve a positive result.

Minor Comments:

6. One additional suggestion: seems to me Section 4 is a bit away from the central topic of the current paper.

I can understand that the optimization/convergence result will help complete the whole picture. However, to contribute to the "margin theme", it would be better to state with the "small vanishing regularization", how it affects the convergence of Theorem 4.3.
Even with this, it is unclear as one don't know how to connect different part of the paper: with what choice of vanishing regularization will generate a solution with a good margin, using the Wasserstein gradient flow.

---

> ### Author Response · Authors · 2018-11-15
> **response**
>
> We thank the reviewer for the comments. We highlight our following novel conceptual and technical contributions.
>
> Conceptual:
>
> 1. First, we point out the conceptual novelty of the overall framework presented in Sections 2 and 3: unlike prior works related to implicit regularization, we disentangle analysis of statistics and optimization by looking at the global minimizer of the weakly-regularized loss.
>
> Bartlett et al.’17 point out that “what is missing is an analysis verifying that SGD applied to standard neural networks returns large margin predictor.” We show that the currently used objective function can encourage a max-margin solution, if optimization is successful.
>
> 2. In our revised Section 3, we apply this framework to study the generalization properties of the solution. The revised Proposition 3.1 bounds generalization error for arbitrary depth networks in terms of the inverse normalized margin, and follows from replacing the product of Frobenius norms in the bound of Golowich et. al (2017) with a sum. It then follows that simple l2 weight decay + logistic loss optimizes the normalized margin and therefore generalization bound.
>
> 3. Our new Corollary 3.2 combined with our observation that the max-margin is nondecreasing in the width of the architecture (old Theorem 3.2, new Theorem 3.3) can explain why over-parameterized networks can generalize better in practice.
>
> Technical:
>
> 1. We show an exact global minimum is unnecessary; to obtain a constant factor approximation of the max margin, we only need to optimize the loss within a constant factor.
>
> 2. We construct a distribution on which neural networks can generalize much better than kernel methods, which highlights the value of depth in generalization.
>
> 3. We prove that noisy gradient descent on infinite neural nets converges to global minimizers in polynomial time. We emphasize the polynomial time nature of our result; prior work (such as [Mei Montanari and Nguyen 2018]) does not specify a convergence rate. The analysis for this result is technically involved.
>
> In our revision of the paper, we have incorporated the reviewer’s feedback. The revision also addresses the concern about novelty in Sections 3.1-3.3 by stating explicitly the relationships with prior work.
>
> Responses to specific comments:
>
> --- “observation … is a nice addition, but not crucial enough to stand out as an innovation”
>
> As argued earlier, we believe our application of Theorem 2.1 to disentangle statistics and optimization is innovative and a key conceptual contribution of our work.
>
> --- “difficulty in related paper lies in achieving non asymptotic convergence rate to max margin solution”
>
> We’d argue that with the current techniques, it’s not clear implicit regularization results can be rigorously achieved for non-linear models. (Related work often assumes convergence in direction.)
>
> In contrast, we show polynomial time convergence to the max-margin solution for infinite size neural networks. This doesn’t solve the finite neuron case either, but the analysis works for a broad class of networks and is technically involved.
>
> ---  “Comparing the upper bound doesn't mean kernel method is performing bad for the instance.”
>
> Our experiments do show that kernel methods indeed perform worse than neural nets.
>
> We believe that it is reasonable and insightful to compare upper bounds, as constructing generalization lower bounds for fixed feature spaces is difficult. In the literature, lower bounds are commonly constructed for some choice of the feature map. Our comparison is challenging because we construct the gap for the fixed ReLU feature map.
>
> --- “From a logic view, it is unclear the benefit of Theorem 3.5.”
>
> We argue that the comparison in Theorem 3.5 is valuable because it highlights the importance of depth in generalization (as kernel methods correspond to fixing random hidden layer weights and only training the last layer.)
>
> ---  “The perturbed wasserstein flow the authors considered looks very close to [Mei Montanari and Nguyen 2018], Eqn 11-12”
>
> We prove a polynomial time convergence result, whereas the result of [Mei, Montanari, and Nguyen 2018] does not characterize convergence rates. Our algorithms are also different: our noise corresponds to randomly re-initializing a small fraction of the neurons, whereas their noise corresponds to Langevin dynamics in parameter space.
>
> --- “emphasize on why the "simga" can help you achieve a positive result”
>
> The small noise ensures that there will be some mass in a descent direction, which allows the algorithm to decrease the objective.
>
> ---  “it would be better to state with the "small vanishing regularization", how it affects the convergence of Theorem 4.3.”
>
> For the weakly regularized loss, the convergence rate will be polynomial in problem parameters and 1/\lambda. Choosing \lambda = 1/poly(n) is sufficient to obtain a constant factor approximation to the max margin and also gives polynomial time convergence.

---

> > ### Comment · AnonReviewer3 · 2018-12-04
> > **corollary 3.2**
> >
> > I actually have concerns about correctness of Cor. 3.2
> >
> > It seems to me that to take limsup in the proof of corollary you need the bound of 3.1 to hold simultaneously for all lambda with high probability (i.e. uniform bound over lambda).
> >
> > One probably can give such a bound (akin to uniform bound for a margin parameter) but you may have to pay a price of sqrt( log log 1/lambda) / sqrt(n), which does not allow you to take limsup.
> >
> > In either case, I think the proof of Cor. 3.2 needs to be written out in detail.

---

> > > ### Author Response · Authors · 2018-12-05
> > > **response to comment**
> > >
> > > --- “It seems to me that to take limsup in the proof of corollary you need the bound of 3.1 to hold simultaneously for all lambda with high probability (i.e. uniform bound over lambda).”
> > >
> > > As mentioned in our response to the other comment, the generalization bound of Proposition 3.1 holds uniformly for all lambda, allowing us to take the limsup.

---

### Public Comment · (anonymous) · 2018-10-09
**Missing prior work**

 The results in Theorem 2.1 are not new. These results have already been shown by Rosset et. al. (see Theorem 2.1 of https://papers.nips.cc/paper/2433-margin-maximizing-loss-functions.pdf). Although the work of Rosset et. al. only focuses on the linear classifiers, their results can be easily extended to homogenous functions classes, which is the primary focus in the current paper. So, it'd be good if the authors cite and discuss this work in the paper.

---

> ### Author Response · Authors · 2018-10-10
> **Thank you for the feedback.**
>
> Thank you for expressing this concern in your comment. We have already acknowledged the connection of our work with that of Rosset et. al in Section 1.1 (see page 3) and Section A.1 (see page 13); however, the next revision of our paper will highlight this connection more prominently.
> Although our proof techniques are based on those of Rosset et. al, we believe our Theorem 2.1 is interesting on its own because of its applicability to deep ReLU networks. We would also like to emphasize that our paper does not only focus on Theorem 2.1, but also presents other main results in Sections 3 and 4. Section 3 discusses properties specific to neural net margins, and Section 4 discusses the optimization of regularized infinite-sized neural networks.

---

### Public Comment · (anonymous) · 2018-11-06
**Technical Details and other claims**


1) On over-parametrization leading to generalization: The authors claim that their results can explain why over-parametrization leads to better generalization. But the results in Thm 3.1 only provide upper bounds on generalization. So the authors should explain why these bounds are not loose (how do these bounds compare with other existing bounds; example Neyshabur et al?). If the bounds are loose, then these results need not explain the over-parametrization puzzle.

2) Some of the claims made in the paper need better justification. For example, on page 2, line 4 the authors say that "Our work explains why optimizing the training loss can lead to parameters with a large margin and thus, better generalization error". It is not clear how the results in Thm 2.1 can explain this. To be more precise, Thm 2.1 only talks about global minimizers when the regularization parameter gets close to 0. Why should this global minimizer be related to the solutions found by GD/SGD on the training loss?

3) Theorem C.1, Equation C.2. \gamma_{\alpha} is a random quantity. So care needs to taken when replacing \gamma in Equation C.1 with \gamma_{\alpha}. In Equation C.1, \gamma is a fixed quantity.

Same comment holds for Thm 3.1, Equation 3.3.

---

> ### Author Response · Authors · 2018-11-15
> **Thank you for the interest in our work.**
>
> Thank you for your interest in our work. Our responses to your comments:
>
> --- “So the authors should explain why these bounds are not loose”
>
> In general, generalization error bounds are probably not meant to be tight: existing bounds for even linear models can be loose by constant factors or in many cases more than constant factors empirically. Arguably, their purpose is instead to provide intuition for comparing models and designing regularization schemes. The significance of the bound in Theorem 3.1 is that it highlights normalized margin as an important factor in generalization, and our Theorem 2.1 explains why one could hope to obtain a maximum normalized margin. This combination can explain why the neural networks that we train can generalize well in practice. Empirically, we observe that these bounds do decrease with the size of the hidden layer, as does the test error.
>
> --- “how do these bounds compare with other existing bounds; example Neyshabur et al?”
>
> The generalization bound in Theorem 3.1 is a direct consequence of the bounds of Neyshabur et. al (2015b) and Golowich et. al (2017).
> Our latest revision has made this point clear (see Proposition 3.1 in the revision).
>
> We would like to emphasize that we do not consider the bounds to be a contribution of our paper; rather, we consider the key contributions of our old Section 3.1-3.3 to be conceptual.
>
> First, the bound in the new Proposition 3.1 depends only on the depth and normalized margin of the network (and we also note that it applies to arbitrary-depth networks). By applying our Theorem 2.1 with this bound, we make the important observation that simple weight decay with cross entropy optimizes a very natural generalization bound for deep networks (that only depends on the parameters through the normalized margin). This is a priori not obvious and is an important conceptual explanation for why deep nets can generalize in practice.
>
> Second, in our new Theorem 3.3, we also note that the margin is non-decreasing in the width of the architecture for arbitrary networks. Combined with our Theorem 2.1, which explains why one could hope to achieve a maximum margin in the first place, this explains why over-parameterization has been observed to improve generalization in practice.
>
> --- “Why should this global minimizer be related to the solutions found by GD/SGD on the training loss?”
>
> First, our Theorem 4.3 (new Theorem 5.3) shows that the global minimum is in fact attainable in polynomial time via noisy GD for extremely over-parameterized networks. This is a key technical contribution of our paper and requires a fairly involved technical analysis.
>
> Second, the behavior of GD/SGD is generally challenging to analyze, and our assumption of convergence to a global minimizer fits in context with assumptions made in prior work about the iterates of GD/SGD (see Theorem 1 of Gunesekar et. al 2018b, for example, which assumes that the loss goes to zero and the difference between the iterates of GD converges). Furthermore, our Theorem 2.2 relaxes the assumption about attaining a global minimizer: we only require a constant factor approximation of the global minimum loss value in order to obtain a constant factor approximation of the maximum margin, and the generalization bound of our old Theorem 3.1/new Proposition 3.1 will essentially be just as good.
>
> Finally, we would like to emphasize the conceptual contribution here: by considering global minimizers, we can disentangle the statistical properties of these solutions from how they are obtained by the optimization algorithm. We believe this disentanglement will be useful for future analyses.
>
> --- “So care needs to taken when replacing \gamma in Equation C.1 with \gamma_{\alpha}.”
>
> Thank you for pointing this out. Our proof relies on Theorem 2 of Kakade et. al (2009). Theorem 2 of Kakade et. al (2009) holds for all choices of margin \gamma, and therefore Equation C.1 in our old Theorem C.1 also holds for all choices of margin \gamma. Our new revision states Theorem 2 of Kakade et. al (2009) explicitly.

---

### Author Response · Authors · 2018-11-15
**Overview of our Revisions**

We have revised the paper to incorporate feedback from the reviewers and address a common criticism from the reviewers, which is the lack of references to prior work in our old Section 3.

Besides adding more explicit references to prior work and addressing specific feedback from reviewers, we have made the following additional major changes:

--- We have separated our former Section 3 into two sections numbered 3 and 4. Our new Section 3 is on generalization properties which follow as a direct consequence of pre-existing bounds and our Theorem 2.1, and our Section 4 emphasizes the comparison between the margins of two-layer neural networks and kernel methods.

--- In our new Section 3, we state our results in terms of general depth-K neural networks, as they also hold in this setting. This emphasizes that our Theorem 2.1 can be applied to analyze generalization for a very broad class of networks.

In this new section, we have separated our old Theorem 3.1 (the
two-layer generalization bound) into Proposition 3.1 and Corollary 3.2. Proposition 3.1 states essentially the depth-K version of the bound of the old theorem. Corollary 3.2 combines Theorem 2.1 and Proposition 3.1. We have also replaced our old Theorem 3.2 (which states that the maximum margin is non-decreasing in the width of a two-layer network) with the new Theorem 3.3, which generalizes this observation to deep networks.

--- In our new Section 4, we highlight the comparison between neural networks and kernel methods as a main contribution.

--- We have added an experiment that verifies that the test error does decrease and margin does increase as the size of the hidden layer increases. We have also moved an experiment verifying the convergence to the max margin solution as lambda goes to 0 to the appendix.

We hope that our restructuring of our old Section 3 better highlights the contributions of our old Sections 3.1-3.3, which is in the application of our Theorem 2.1 to look at generalization properties of the neural network obtained from training without worrying about the training algorithm itself. Specifically, this allows us to provide a very simple analysis for why over-parameterization can improve generalization (see our new Theorem 3.3).

We also hope that this restructuring will place more emphasis on our comparison between neural networks and kernel methods. We believe this comparison is valuable because it sheds more light on the generalization properties of neural nets and provides evidence that optimizing all layers is beneficial for generalization.

---

### Meta-Review · Area_Chair1 · 2018-12-16
**robust reviewing process; rewrite required**

**Confidence:** 4
**Recommendation:** Reject

**Metareview:**

This paper has received reviews from multiple experts who raise a litany of issues. These have been addressed quite convincingly by the authors, but I believe that ultimately this work needs to go through another round of reviewing, and this cannot be achieved in the context of ICLR's reviewing setup. I look forward to reading the final version of the paper in the near future.